Resource

# Dynamic changes in RNA–protein interactions and RNA secondary structure in mammalian erythropoiesis

Mengge Shan[1,2] , Xinjun Ji[3], Kevin Janssen[5] , Ian M Silverman[3] , Jesse Humenik[3], Ben A Garcia[5], Stephen A Liebhaber[3,4], Brian D Gregory[1,2]

**Two features of eukaryotic RNA molecules that regulate their post-transcriptional fates are RNA secondary structure and RNA-binding protein (RBP) interaction sites. However, a comprehensive global overview of the dynamic nature of these sequence features during erythropoiesis has never been obtained. Here, we use our ribonuclease-mediated structure and RBP-binding site mapping approach to reveal the global landscape of RNA secondary structure and RBP–RNA interaction sites and the dynamics of these features during this important developmental process. We identify dynamic patterns of RNA secondary structure and RBP binding throughout the process and determine a set of corresponding protein-bound sequence motifs along with their dynamic structural and RBP-binding contexts. Finally, using these dynamically bound sequences, we identify a number of RBPs that have known and putative key functions in post-transcriptional regulation during mammalian erythropoiesis. In total, this global analysis reveals new post-transcriptional regulators of mammalian blood cell development.**

## Introduction

In eukaryotic systems, RNA-binding proteins (RBPs) interact with RNAs from synthesis to decay, thereby adding complexity to the transcriptome. A recent whole mRNA RBPome study identified several hundred RBPs in HeLa cells that have the capacity to play critical roles in determining RNA functions (1). These functions encompass an array of post-transcriptional processes including splicing, polyadenylation, nuclear export, localization, transport, translation, and degradation (2).

RBPs have been shown to interact with their binding targets in a sequence and RNA secondary structure specific manner (3). Several high-throughput methods have emerged in the last decade to address the interplay between RNA-binding proteins, their targets,

and RNA secondary structure. These techniques generally either use chemical probing agents or structure-specific RNases (single-stranded RNases (ssRNases) and double-stranded RNases [dsRNases]) to provide site-specific evidence for a region being in single- or double-stranded configurations (4, 5).

To date, the known repertoire of RBP–RNA interaction sites has been built on a protein-by-protein basis, with studies identifying the targets of a single protein of interest, often through the use of techniques such as crosslinking and immunoprecipitation sequencing (CLIP-seq) (6). In CLIP-seq, samples are irradiated with UV to induce the cross-linking of proteins to their RNA targets. Subsequent immunoprecipitation with an antibody pulls down the protein and bound RNA fragments, and these fragments are then sequenced and mapped back to the transcriptome. For instance, this technique was used to identify PABPC1-binding sites throughout the transcriptome, and this study was further able to demonstrate that 5′ UTR PABPC1 interaction sites mediated translational control of bound target transcripts (7). In contrast to this targeted approach, we have recently reported the approach of protein interaction profile sequencing (PIP-seq) (8) that allows for a global and unbiased analysis of RBP–RNA interaction sites in a sample of interest. In PIP-seq, RNA-protein interactions are stabilized by formaldehyde cross-linking followed by interrogation using a combination of proteinase, ssRNase, and dsRNase treatments. This approach creates a set of orthogonal libraries that are capable of concurrently elucidating RNA secondary structure and protein bound sequences throughout the transcriptome of interest (8).

Studies on mammalian erythropoiesis have revealed that this process involves a series of complex and stage-specific changes in gene expression. Mature erythrocytes are derived from hematopoietic stem and progenitor cells, which undergo a series of increasingly restrictive lineage commitment events. Importantly, this process includes a significant dependence on post-transcriptional regulatory processes, especially during its terminal steps (9). Terminal erythropoiesis involves a decrease in cell size, an increase in the production of hemoglobin, membrane reorganization, chromatin condensation, and finally enucleation (10, 11). Whereas each

[1]Department of Biology, University of Pennsylvania, Philadelphia, PA, USA   [2]Genomics and Computational Biology Graduate Group, Perelman School of Medicine, Epigenetics Institute, University of Pennsylvania, PA, USA   [3]Department of Genetics, Perelman School of Medicine, Epigenetics Institute, University of Pennsylvania, PA, USA   [4]Department of Medicine, Perelman School of Medicine, Epigenetics Institute, University of Pennsylvania, PA, USA   [5]Department of Biochemistry and Biophysics, Perelman School of Medicine, Epigenetics Institute, University of Pennsylvania, PA, USA

Correspondence: bdgregor@sas.upenn.edu; Correspondence: liebhabe@pennmedicine.upenn.edu

stage of differentiation also exhibits stage-specific transcriptomes, the contributions of RNA secondary structure and RBP-binding site dynamics to this process have not been previously explored on a global scale.

Here, by performing PIP-seq on mouse erythroid leukemia (MEL) cells, we identify RBP-binding sites, provide a transcriptome-wide look at RNA secondary structure, and profile these RNA features and their interactions throughout the terminal stages of red blood cell development. Our results produce an unbiased view of RBP-binding events and RNA secondary structure changes that occur during the terminal stages in mammalian red blood cell development. In addition, the datasets provide a resource for future investigations of the functional importance of bound RNA regions and RNA secondary structures to this critical process of cellular differentiation.

## Results

### MEL cells as a model for red blood cell differentiation

Murine erythroleukemia (MEL) cell lines are arrested at the pro-erythroblast stage of development, show very low level of spontaneous differentiation, can be maintained indefinitely in tissue culture, and can be induced to differentiate along the erythroid lineage through treatment with a variety of chemical compounds such as DMSO. Studies show that by day 3 of growth on DMSO, there is a 20-fold increase in heme uptake, and a 12-fold increase in hemoglobin synthesis compared to unstimulated cells and by day 4 the cells have mostly matured to normoblasts that stain positive for benzidine reflecting high levels of hemoglobin content (12). Thus, since the 1970s, this cell line has served as an attractive model for the study of the terminal events in erythroid differentiation in vitro (13).

MEL cells undergoing terminal erythroid differentiation become smaller in size and show marked nuclear compaction (14, 15). At this late stage of differentiation, transcription is dramatically curtailed and the cellular events are almost entirely post-transcriptionally regulated (16), making this system an attractive model for studying mammalian post-transcriptional regulation in the context of an important developmental process. For our analysis, we generated two biological replicates of cultured MEL cells at each of three time points post DMSO treatment: 0, 2, and 4 d. mRNA-seq on these samples confirmed the biological relevance of the selected time points by revealing large-scale increases of globin gene transcripts as would be expected in a setting of terminal erythroid differentiation (Fig S1A). Specifically, NM_001278161 (Hbb-b1), NM_016956 (Hbb-b2), NM_008218 (Hba-a1), NM_001083955 (Hba-a2), NM_001201391 (Hbb-bs), and NM_010405 (Hba-x) are among the genes whose abundance continuously increases throughout the three time points of differentiation used in this study (false discovery rate [FDR] < 0.05, DESeq2 analysis). Previous work has also characterized a set of genes whose expression is repressed as MEL cells progressively commit to the erythroid lineage. We find that our data recapitulates several of these repressed marker genes (13). Specifically, by day 4 of DMSO induction, we observed that Cdk4, GAPDH, Myb, and Myc are all significantly (FDR < 0.05, DESeq2 analysis) down-regulated (Fig S1B).

As expected, a Gene Ontology (GO) enrichment analysis of the differentially abundant transcripts showed an enrichment for erythrocyte developmental genes in transcripts that are continuously increasing ($P$-value < 8.04 × 10$^{-5}$, hypergeometric test). On the other hand, continuously decreasing transcripts showed an enrichment for those associated with nucleosome assembly ($P$-value < 8.97 × 10$^{-12}$, hypergeometric test) and other terms associated with DNA binding and gene regulation (Fig S1C). Furthermore, when we retrieved the 133 transcripts annotated to be relevant to erythrocyte differentiation from the Mouse Genome Institute (17, 18) and queried how many of them were statistically differentially abundant in our mRNA-seq dataset, we found that of the 123 genes with retrievable RefSeq accession identifiers, 28 were differentially abundant in cells 2 d after DMSO treatment (day 2) as compared to before treatment (day 0), 9 were differentially abundant after 4 d of treatment with DMSO (day 4) compared to 2, and a total of 33 transcripts were differentially abundant when comparing data from day 4 cells to those from before DMSO treatment. In total, these mRNA-seq data suggest that these MEL cells are indeed undergoing erythroid differentiation and serve as an appropriate gateway to studying post-transcriptional regulation using our PIP-seq pipeline.

### PIP-seq analysis of MEL cell development

PIP-seq libraries were prepared from two biological replicates of MEL cells collected at 0-, 2-, and 4-d post cell differentiation induction with DMSO (same samples as used for mRNA-seq) (19). PIP-seq analysis allows global identification of RNA-protein interaction sites as well as mapping of RNA secondary structure (8, 20, 21). Briefly, total cellular extract was divided into footprinting and structure only samples (four total libraries per replicate). To globally identify RBP-bound RNA sequences, footprinting samples were treated with an RNase specific to either ssRNA or dsRNA (ssRNase or dsRNase, respectively), followed by protein denaturation and sequencing library preparation. Conversely, the "structure only" samples had proteins denatured in SDS and degraded with Proteinase K prior to RNase digestion. The denaturation of proteins before RNase treatment makes sequences that were RBP-bound in the footprinting sample accessible to RNases in these "structure only" reactions. Thus, sequences that are enriched in footprinting relative to "structure only" samples are identified as protein-protected sites (PPSs) (8, 20, 21). The "structure only" libraries allowed us to determine the native (protein-bound) RNA base-pairing probabilities for the total transcriptomes of the three MEL cell developmental time points (Fig S2).

The 24 resulting libraries (three time-points, two replicates per time-point, four libraries per replicate) produced between 36 and 58 million raw reads per library. We first assessed the reproducibility of our technology by using a 1,000-nt sliding window approach to correlate the read abundance between the biological replicates of the footprinting or structure-only samples at each time point. Overall, we observed high Pearson correlation values between 0.985 and 0.989 (Fig S3A), indicating the high reproducibility and quality of these PIP-seq libraries.

To identify PPSs in the three different MEL developmental stages, we used a Poisson distribution model to identify regions enriched in the footprinting samples as compared to the structure-only libraries for the three different time points at a FDR of 5% as

previously described (8, 20, 21). Because ribosomes are also capable of interacting with RNA and PIP-seq is unable to distinguish between ribosome-protected sites and RBP-protected sites, we filtered out PPSs that were between 20 and 40 nts in length as the majority of ribosome occupancy sites are ~30 nt long (22). We first analyzed the size distribution of our entire collection of PPSs (Fig 1A) and determined that the most PPSs were >40 nt. In fact, only about 17% of PPSs fell within the 20–40 nt region, and these PPSs were then excluded from further analysis to minimize the effect of potential ribosomal interaction.

To further assess the ability of our PIP-seq data to reproducibly characterize the global landscape of RBP-binding and RNA secondary structure in the MEL cell samples, we calculated PPS density and RNA secondary structure scores at each nucleotide (termed RBP-binding and RNA secondary structure, respectively) for all detectable protein-coding transcripts across the three developmental time points as previously described (8, 20, 21, 23, 24). Using the structure-only samples, RNA secondary structure is represented by a structure score, which is a generalized log ratio (glog) of dsRNA-seq and ssRNA-seq reads at a particular nucleotide, with positive and negative scores indicating nucleotides that are more likely to be paired (dsRNA) or unpaired (ssRNA), respectively. The raw structure scores were then normalized to the average structure score of the entire spliced transcript, resulting in structure scores in which the positive or negative values indicate the likelihood of a nucleotide being double-stranded (more structured) or single-stranded (less structured), respectively. To ensure reproducibility of the calculated structure scores, structure scores for each biological replicate of the three developmental time points were calculated separately. This analysis revealed a significant level of similarity in overall structure scores for total transcripts (Spearman's $\rho$ > 0.93; $P$-value < 2.2 × 10$^{-16}$; asymptotic t approximation) (Fig S3B and C) between biological replicates of all three time points, further confirming the high reproducibility of the PIP-seq experiments' ability to assess the global landscape or RNA secondary structure.

To ensure reproducibility of the RBP-binding levels, average PPS density for each biological replicate of the three developmental time points was calculated separately. This revealed a significant level of similarity in the patterns of RBP binding (PPS density) surrounding the start and stop codons of mRNAs (Spearman's $\rho$ > 0.82; $P$-value < 2.2 × 10$^{-16}$; asymptotic t approximation) (Fig S4A–C) between biological replicates of all three time points, providing even more evidence of the high reproducibility of the PIP-seq experiments. Given this significant level of correlation between replicate datasets for both RBP-binding and RNA secondary structure analysis, all further analyses were performed using structure scores and RBP-binding values calculated from merged biological replicates.

### The RNA–protein interaction landscape of developing red blood cells

Even after the previously described filtering step, we identified 245,466 total PPSs (total PPSs) across the three developmental time points (Fig 1B). On average, 41% of the PPSs were detected across both biological replicates at each of the three time points (high confidence PPSs), with the lowest reproducibility found in the 3′ UTRs (Fig S5A). This reproducibility compares favorably with CLIP-

seq experiments, which generally produce <35% overlap between biological replicates, especially when considering the complexity of the RBP bound transcriptome in mammals (25). As further confirmation of bona fide RBP interaction site detection by PIP-seq, we overlapped our PPSs and a set of randomly generated mock PPSs with binding site calls for PABPC1 or PABPC4 from a high-quality CLIP-seq dataset (7). For both no treatment (day 0) and DMSO treatment for 2 d (day 2), PIP-seq detected PPSs were significantly ($P$-value < 2.2 × 10$^{-16}$; $\chi^2$ test) more enriched for CLIP-detected PABPC1 and PABPC4-binding sites than a set of randomly selected background control set of sites (see the Materials and Methods section). Specifically, 9.53% of day 0 PABPC1 CLIP sites overlapped with the day 0 PPSs by at least one nucleotide compared to 0.24% of the background control set. Similarly, 7.41% of day 2 PABPC1 CLIP sites overlapped a high confidence PPS site, while only 0.29% of the same sites overlapped the background sites (Fig S5B). The same pattern of higher overlap between CLIP sites and actual PPSs as compared to background control sites was observed for PABPC4 CLIP-seq data for both day 0 and day 2 treatments. This recapitulation of known PABPC1 and PABPC4-binding sites in our de novo PPS detection confirms that PIP-seq is capable of detecting verifiable RBP–RNA interaction sites. Also, these data were remarkably consistent with PABPC1's known interaction with the poly(A) tail (26, 27); PPSs that overlapped PABPC1 CLIP-sites showed a positive enrichment in the 3′ UTR identified PPSs (Fig S5C).

As outlined above, PIP-seq identified a total of 245,466 PPSs across the three developmental time points. These PPSs could be further classified into "control PPSs," representing the PPSs that were found in undifferentiated cells (n = 49,663), or "developmental PPSs," which are the PPSs found only in differentiated cells (n = 121,705). The rest of the PPSs were distributed among the possible combinations of time points, with 36,510 PPSs being shared between all three measured timepoints (Fig 1B). The PPSs were then grouped based on their genomic classification (i.e., intron, CDS, 3′ UTR, or 5′ UTR) within mRNAs. This revealed that the majority of PPSs are found within the intron (47–55%), followed by the coding sequence (CDS) (21–27%), and the 5′ and -3′ UTRs (8–12% total) (Fig 1C). Overall, these patterns showed no significant differences across the three time points examined in our data.

### Potential RBP–RNA interacting sites are evolutionarily conserved

Because RBP–RNA interaction sites can be sequence dependent, we hypothesized that functionally important interaction sites would be less prone to random genetic mutation. First, we characterized the nucleotide frequency of the PPSs and found that the nucleotides contained in PPSs seem to be equally distributed among the four bases with the exception of Cs, which are the least common nucleotide present in collection of identified PPSs (Fig 1D). We hypothesized that these PPSs, being potential sites of interaction between RBPs and RNAs, would be more conserved than sites that are not interacting with RBPs. Using a comparison of the 60-way PhastCon scores calculated using mammalian species (including guinea pig, kangaroo rat, and rabbit), we also found that PPSs have significantly (all $P$-values < 0.001, Kolmogorov-Smirnov test) higher PhastCon scores than their equally sized flanking regions that occur directly up- and downstream of the PPS (see the Materials and

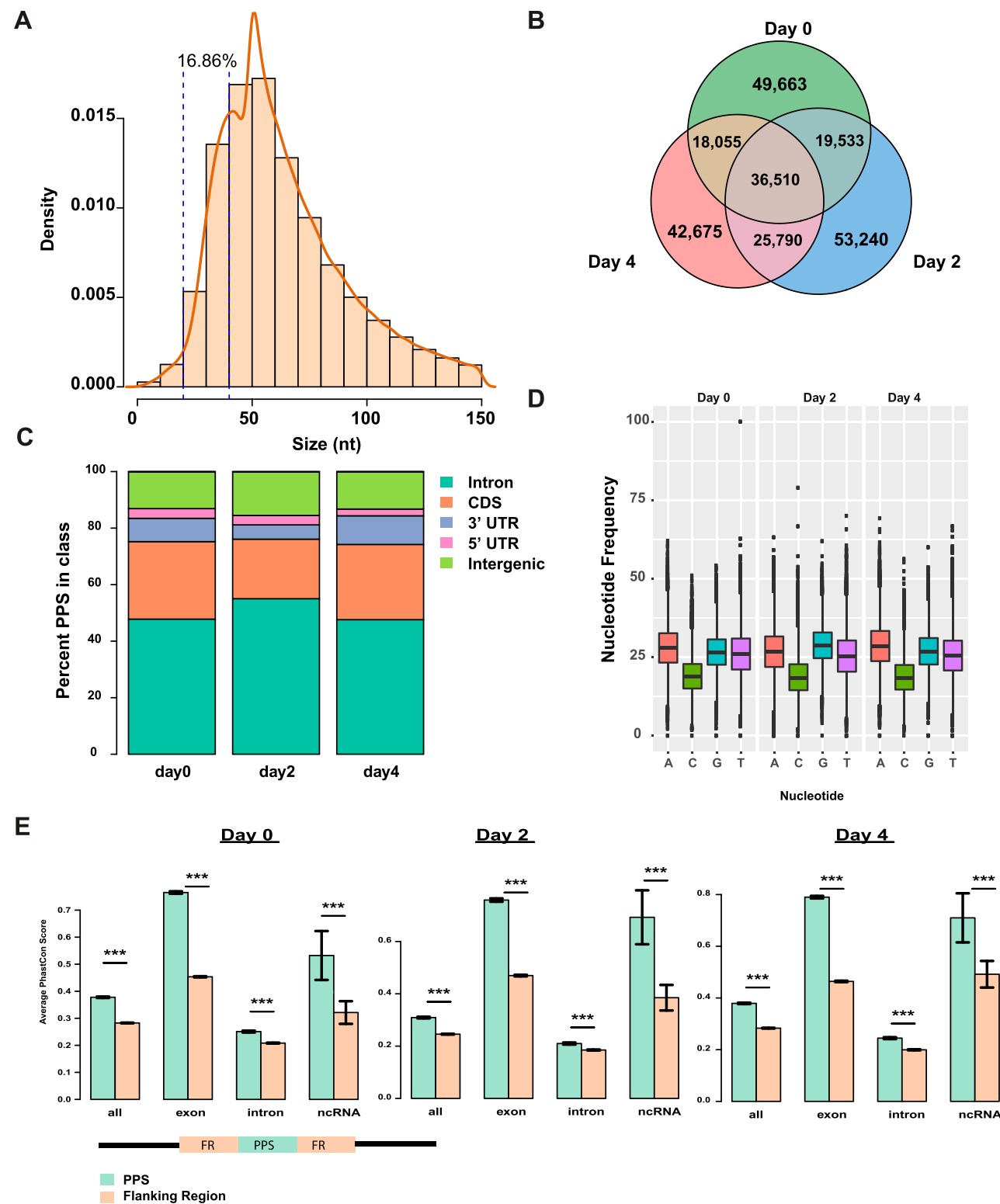

**Figure 1. Characterization of protein-protected sites (PPSs) during MEL differentiation.**
**(A)** Density plot showing the distribution of the entire collection of PPSs before filtering out those within 20–40 nt size range (region bounded by blue dotted line; percentage indicating the percentage of total PPSs that were filtered out). **(B)** Overlap of PPSs identified at day 0 (green), day 2 (blue), and day 4 (red). **(C)** The genomic classes represented by PPSs before, 2, or 4 d after induction of MEL cell differentiation. **(D)** Boxplots of distribution of each nucleotide in PPSs found at 0, 2, or 4 d post induction with DMSO. **(E)** Comparison of average PhastCon scores between PPSs (green bars) and equal-sized flanking regions (orange bars) for various genomic regions at each time point. ***denotes $P$-value < $1 \times 10^{-10}$, Kolmogorov–Smirnov test. Error bars ± SEM.

Methods section) (Fig 1E), indicating that PPSs are even more evolutionarily conserved than their neighboring regions. Consistent with our model, we observed that PPSs within exons had the highest PhastCon score, indicating that these PPSs are the most conserved throughout evolution. These results lead us to propose that PPSs, as potential interaction sites between RBPs and their target RNA, are less prone to the effects of genetic drift in mice, likely as a result of maintaining a sequence-specific interaction with a RBP(s) (20).

## RBP–RNA interactions are dynamic during terminal erythropoiesis

The current understanding of RBP–RNA interactions is that of a dynamic relationship where RBPs bind to specific locations along an RNA molecule depending on the RBP's role in the RNA's lifecycle. Thus, specific RBP–RNA interactions can occur in a cell and developmental specific context to affect a specific aspect of the target RNA molecule. In support of this, we witnessed the dynamics of these relationships in our PIP-seq data, as there were PPSs that are only detected after MEL cells have been induced to differentiate (Fig 1B), and we were able to subset the MEL transcriptome into transcripts that only had day 0 PPSs and those with PPSs only after differentiation induction (Fig 2A). An analysis of the mammalian phenotypes (MPs) associated with these two subsets of transcripts showed that both sets are associated with abnormal hematopoiesis phenotypes, such as abnormal blood physiology (Bonferroni: 0.015; day 0 PPSs), abnormal hematopoietic system physiology (Bonferroni: 0.025; day 0 PPSs), and hematopoietic system phenotype (Bonferroni: 0.028; day 2 and day 4 PPSs). As expected, these two subsets of transcripts also had differing GO enrichment patterns, with transcripts that contain just day 2 and day 4 PPSs being enriched for those encoding proteins required for erythrocyte development, whereas those with day 0 PPSs were enriched for mRNAs encoding proteins involved in DNA and RNA processing (Fig 2C).

When we measured the percentage of a transcript that was covered by a PPS at a given timepoint, we observed that some transcripts become increasingly bound throughout development, just as there were transcripts who become less covered as cells develop (Fig 2D). After identifying the transcripts with the largest increases or decreases in PPS coverage, we noted that they were enriched for transcripts related to abnormal hematopoietic phenotypes (Fig 2E). This led us to examine the relationship between the percentage of a transcript that is covered by PPSs and its RNA abundance. While we detected a visible positive trend between PPS coverage and RNA abundance as measured by median transcripts per million (TPM), we also noted that the range between TPMs in each of the deciles measured was very similar (Fig 2F). This suggests that although there is a minor correlation between PPS coverage and RNA abundance, there are definitely additional factors that regulate RNA-RBP interactions on mRNAs that are independent of RNA abundance.

## RNA-binding proteins are differentially regulated during terminal erythropoiesis

It has long been established that RNA-binding proteins play an essential role in regulating translation during erythropoiesis (28).

Thus, we were interested in studying the regulation of their transcript abundance levels during mammalian erythropoiesis. The RBPDB catalog encompasses 515 murine RBPs (29). Using DAVID (30, 31) we were able to retrieve the appropriate RefSeq annotation for 472 of these RBPs and we examined whether the levels of their encoding mRNAs changed in abundance throughout erythropoiesis as modeled by our MEL cells. Out of the 472 RBPs, 127 of them were identified as being differentially abundant in at least one comparison with FDR < 0.05. Functionally, these differentially expressed RBPs were involved in splicing (Bonferroni: $6.67 \times 10^{-35}$, hypergeometric test), mRNA transport (Bonferroni: 0.009, hypergeometric test), and positive regulation of translation (Bonferroni: 0.013, hypergeometric test). Furthermore, of the 58 RNA-binding proteins that have established roles in the erythropoietic pathway according to their GO annotations, 13 are differentially abundant in our MEL model, with the majority of them showing decreased abundance during the course of this developmental process (Fig S6A). The change in their abundance suggests that these RBPs are either responding to, or acting in, the erythropoietic process.

We have previously demonstrated that PPSs identified in the PIP-seq analysis are sites of RBP–RNA interactions (8, 19, 20, 21). Thus, we leveraged databases such as ATtRACT (32) and RBPDB (29), which contain experimentally determined binding sequences of several murine RBPs, to interrogate whether our PPSs contained the known interaction sequences of any characterized RBPs. To minimize the influence of ribosomal binding, we focused on high confidence PPSs found in the 3′ UTR. Using ATtRACT, we first scanned these PPSs for the binding sequences of potential RBP partners and saw that suggested RBPs were, for the most part, known to play roles in regulating hematopoiesis (Fig S6B). For example, proteins that seem to bind to enriched sequences found in PPSs at 2- and 4-d post induction include PUM2, which is known to regulate hematopoiesis (33), and ADAR1/ADAR2. The protein ADAR1 is essential for erythropoiesis in mice by providing adenosine-to-inosine RNA editing (34), whereas ADAR2 is suspected to be a marker for myeloid blast cell differentiation (35). In addition, the search also identified potential binding sites for the RNA-binding proteins DAZL and TLR3. Whereas DAZL has no currently known roles in erythropoiesis, TRL3 mutant mice do exhibit abnormal hematopoietic phenotypes that include abnormal B and NK cell physiology (36), suggesting TLR3 and potentially DAZL as candidates for future validation. The ability to use our collection of PPSs to identify prevalent motifs for known RBPs creates an attractive method for identifying RBPs that may have, as of yet, unidentified roles in erythropoiesis and provides a valuable resource for developing future research directions.

When we queried our mRNA-seq data for the abundance pattern of transcripts encoding RBPs, we noted that mRNAs encoding nine of those RBPs showed patterns of differential abundance in at least one pairwise comparison, suggesting that they are themselves under regulation in mammalian erythropoiesis development. Interestingly, Tlr3 and Nfkb1, which were the two RBPs with the most potential sites in the sequences that were scanned by ATtRACT, were not among the transcripts with differential mRNA abundance. This lack of correlation may reflect a need for the cell to maintain a steady state of these mRNAs throughout the erythropoiesis process (Fig S6C). However, we did note that Nova2 showed a significant increase in abundance in day 4 cells compared with uninduced

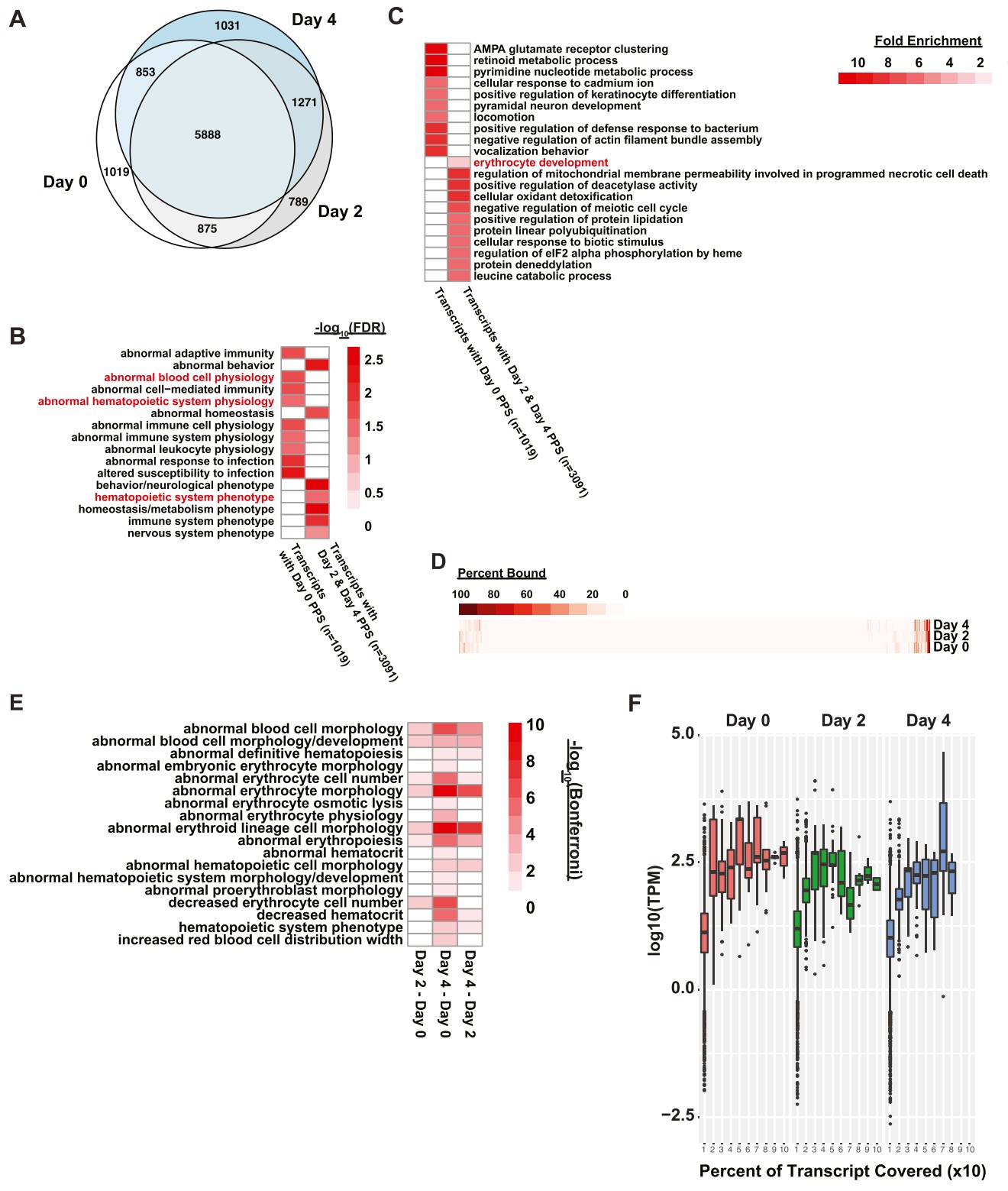

**Figure 2. Dynamics of RBP–RNA interactions in mammalian erythropoiesis.**
**(A)** Euler diagram showing overlap of transcripts with high confidence protein-protected sites (PPSs) found at day 0 (white), day 2 (gray), or day 4 (light blue). **(B)** –log$_{10}$ (false discovery rate) values of blood and hematopoiesis related mammalian phenotype terms associated with either transcripts that only have day 0 PPSs and genes that only have day 2 and day 4 PPSs. **(C)** Fold enrichment of top 10 top significant Gene Ontology (GO) terms and erythropoiesis related GO terms in transcripts with just day 0 PPSs or transcripts with day 2 or day 4 PPSs. Erythropoiesis relevant GO terms are highlighted in red. **(D)** Heat map of transcripts colored by the percentage covered by a PPS at 0, 2, or 4 d after induction. **(E)** Heat map of significant mammalian phenotypes related to erythropoiesis associated with the top 10% of transcripts that increase or decrease in PPS coverage in pairwise comparisons between MEL cells after 0, 2, or 4 d of differentiation. **(F)** Boxplot of transcripts per million values for transcripts at 0 (red), 2 (green), or 4 (blue) d after differentiation, grouped by the percentage of the transcript covered split into deciles.

cells and our high confidence PPSs showed that a large portion of them contained a potential binding site for the NOVA2 protein. This observation supports NOVA2 as an attractive candidate for the regulation of mammalian erythropoiesis. Taken together, our results demonstrate that the transcripts encoding several of the known RBPs are differentially regulated in erythropoiesis and that we can use our PPSs to look for potential regions of RBP–RNA interactions for these known RBPs. Furthermore, we were able to identify several RBPs whose abundances change throughout erythropoiesis and could also potentially bind to high confidence PPSs based on their validated binding sequence.

## RNA secondary structure and RBP-binding landscapes are dynamic in erythropoiesis

To compare the patterns of RNA secondary structure and RBP binding, we focused on the region 400 nt up- and downstream of the start and stop codon of detectable mRNAs expressed in the three developmental time points, as these regions have important regulatory functions in mRNA fate and function. We calculated the RNA secondary structure of 22,605 transcripts for all three time points and uncovered progressive increases in the level of RNA secondary structure in both of these regions during the later stages of mammalian red blood cell development (all $P$-values < 0.001; Wilcoxon test) (Fig 3A). In the 3′ UTR, our RNA secondary structure analysis revealed an overall increase in RNA secondary structure during mammalian erythropoiesis, suggesting that 3′ UTRs are collectively becoming more double-stranded during this developmental process. The pattern is less clear when it comes to the 5′ UTR. In the 5′ UTR, we observed the interesting pattern of a decrease in RNA secondary structure going from day 0 to day 2 cells and then an increase in RNA secondary structure in the same region at day 4 such that, on average, the region becomes more double-stranded in day 4 cells than in day 0 cells. Of note is that the mean RNA secondary structure immediately surrounding the annotated start and stop codons increased throughout development. This general increase in RNA secondary structure is likely to result in RNAs acquiring a more energetically favorable state (more paired) during these later stages of development. In total, these findings reveal large-scale changes in RNA secondary structure during a mammalian cell developmental process, which could be one of the ways that these transcripts are post-transcriptionally regulated in erythropoiesis. Because RNA secondary structure is an important part of post-transcriptional regulation, the transcripts that show marked difference in RNA secondary structure throughout development could be those that are functionally important to mammalian erythropoiesis.

We then measured the PPS density of the same 22,605 transcripts across the three time points and found that PPS density of transcripts is higher in the CDS as than the UTRs, with a slight peak of protein binding directly over the start and stop codons (Fig 3B). In contrast, RNA secondary structure presented a dramatic decrease at both UTR-CDS junctions over the translation start and stop codons and then rose again throughout the CDS and UTR regions (Fig 3A). In the 5′ UTR, day 2 and day 0 PPS density seem to be at similar levels whereas day 4 PPS density is markedly decreased until the start codon. In the 400-nt window after the start codon, we

see that PPS density levels between day 0 and day 4 are overlapping, whereas day 2 levels drop noticeably. The greatest separation between the three time points is observed in the 400-nt window after the stop codon, where we observed that day 2 PPS density levels drop significantly when compared with day 0 levels, whereas day 4 PPS density values were significantly (all $P$-values < 0.001; Wilcoxon test) higher than those at day 0 (Fig 3). Our data revealed an increase in RNA secondary structure in the 5′ UTR of day 4 cells in comparison to uninduced cells and, in the same region, we observed a decrease in PPS density, suggesting that the increase in RNA secondary structure and decrease in RBP–RNA interaction are related. Overall, the structured 5′ UTRs observed at day 4 could serve to impede the binding of RBPs that could regulate the translation or other functionalities of those particular transcripts (37). RBPs have been shown to bind to the 3′ UTR to control mRNA stability and also translation in erythropoiesis (28), and the increase in PPS density in the region at day 4 could be a result of the cell stabilizing the transcripts that are still present in the later stages of development as transcription is decreased. In total, our results demonstrate wide-spread rearrangements of RBP binding near the translation start and stop codons during the process of erythroid terminal differentiation.

As a large portion of the overall transcriptome in late-stage erythroid differentiation is taken up by globin (e.g., Hbb) transcripts, we wondered how much, if any, Hbb contributed to the overall patterns of RNA secondary structure and PPS density changes observed in our PIP-seq analysis. To answer that question, we re-analyzed our data after removing Hbb transcripts and found that neither the RNA secondary structure (Fig S7A) nor the PPS density patterns (Fig S7B) demonstrated appreciable changes, and thus we retained Hbb in all subsequent analyses. We also further investigated the relationship between PPSs and RNA secondary structure by calculating the change in RNA secondary structure for PPSs that are found in the 3′ UTR or the 5′ UTR. For PPSs that were detected in the UTRs at any time point, the average RNA secondary structure showed a very minor increase when we compared induced cells to uninduced cells (Fig S8A), although the values ranged from -3 to 2.6. We observed a similar spread of change in RNA secondary structure scores when we specifically examined PPSs that were only found in induced cells (Fig S8B). These findings suggest that the presence or absence of a PPS is not uniformly correlated with an increase or decrease in RNA secondary structure in the untranslated regions.

Although the addition of an RBP–RNA interaction site in the UTRs did not lead to a consistent increase or decrease in RNA secondary structure, we do observe an anti-correlation between RBP–RNA interaction sites and RNA secondary structure on a global scale when we examine the window around the start and stop codons in entire transcriptome instead of focusing on specific transcripts. Consistent with previous studies carried out in Arabidopsis (20, 21), our combined analyses of RBP binding and RNA secondary structure revealed that these two features tend to be generally anti-correlated features across all detectable mRNAs (Spearman's $\rho \leq -0.25$; $P$-values < 3.97 × 10$^{-4}$; asymptotic $t$ approximation) in all three developmental time points (Fig S9A–C). In addition to this transcriptome-wide pattern for all three time points, we found on average that the strongest anti-correlations occurred in the last 100 nt of the 5′ UTR and into the first 100 nt of the CDS (Spearman's $\rho \leq -0.46$; $P$-values < 1.03 × 10$^{-11}$;

**A**

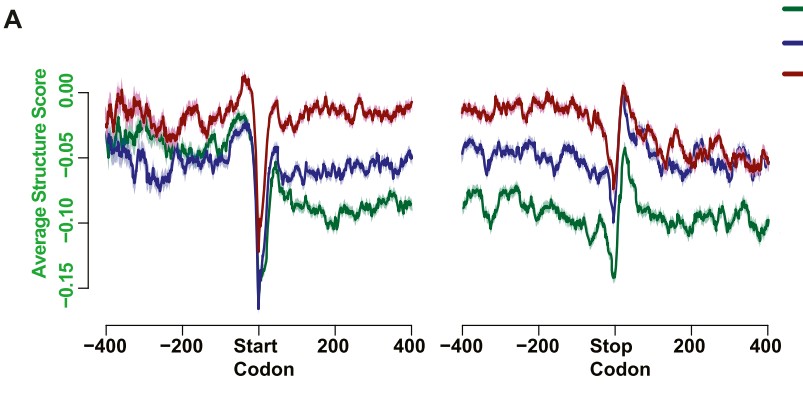

| Region | Day 2 vs Day 0 | Day 4 vs Day 0 | Day 4 vs Day 2 |
|---|---|---|---|
| -400 - Start | < 2.2e-16 | < 2.2e-16 | < 2.2e-16 |
| Start - 100 | < 2.2e-16 | < 2.2e-16 | < 2.2e-16 |
| -400 - Stop | < 2.2e-16 | < 2.2e-16 | < 2.2e-16 |
| Stop - 400 | < 2.2e-16 | < 2.2e-16 | < 2.2e-16 |

**Figure 3. Distinct RNA-protein and RNA secondary structure profiles in differentiating MEL cells.**
Scaled RNA secondary structure score (top) or average protein-protected site density profiles (bottom) at each nucleotide ± 400 nt from the annotated start or stop codons in detectable mRNAs expressed in MEL cells before (green lines) as well as 2 (blue lines) or 4 (dark red lines) d after MEL cell differentiation. Table below each chart lists the calculated *P*-value for the ± 100 nt surrounding the start and stop codon per the Wilcoxon rank sum test. Solid lines indicate the average value at the position and shading around the lines represent ± SEM.

**B**

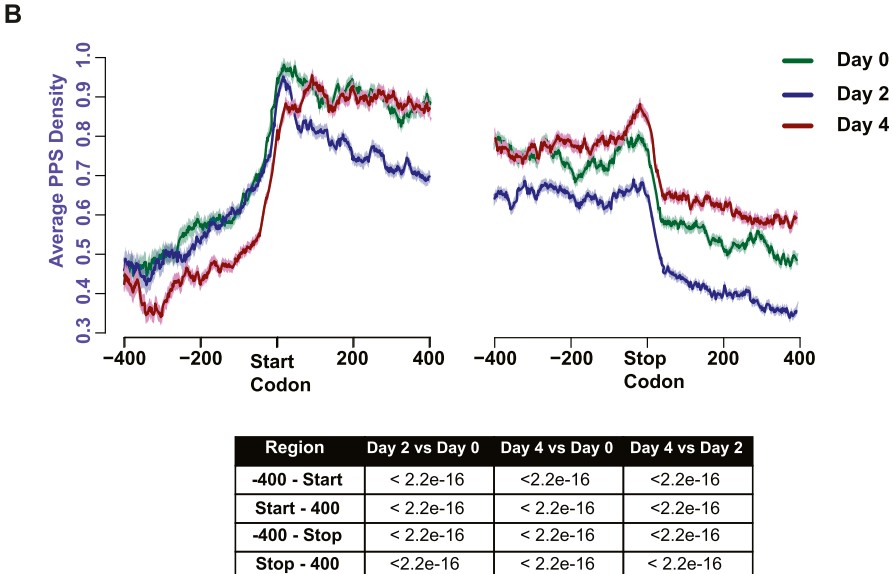

| Region | Day 2 vs Day 0 | Day 4 vs Day 0 | Day 4 vs Day 2 |
|---|---|---|---|
| -400 - Start | < 2.2e-16 | <2.2e-16 | <2.2e-16 |
| Start - 400 | < 2.2e-16 | < 2.2e-16 | <2.2e-16 |
| -400 - Stop | < 2.2e-16 | < 2.2e-16 | <2.2e-16 |
| Stop - 400 | <2.2e-16 | < 2.2e-16 | < 2.2e-16 |

asymptotic *t* approximation) as compared with the 200 nt surrounding the stop codon, which demonstrated less, but still significant, anti-correlation in the two developmental time points (Fig S9A–C). In general, these findings reveal that RNA secondary structure and RBP binding tend to be anti-correlated features in mammalian red blood cell progenitors.

We then separated the transcripts into percentiles based on the changes in their RNA secondary structure and analyzed the enriched GO terms and MPs associated with those that show the most increase in RNA secondary structure, those that show the most decrease in RNA secondary structure, and those that show little change (45–55th percentiles of change) in RNA secondary structure as controls. When we looked at the change in differentiated cells from their undifferentiated state, we found that most of the enriched GO terms are involved in general biological pathways, including primary metabolic and nucleic acid metabolic process

(Fig 4A). In terms of mammalian physiology, transcripts that increase or decrease in RNA secondary structure are both associated with general organismal survival. We observed that transcripts which fall within the top 10% of transcripts that increase or decrease in structure after 4 d of DMSO-induction display enrichment for phenotypes such as abnormal erythropoiesis (FDR: 0.01), abnormal definitive hematopoiesis (FDR: 0.002), and abnormal blood cell morphology/development (FDR: 3.18 × $10^{-5}$) (Fig 4B). Overall, our findings revealed an enrichment for transcripts encoding proteins associated with hematopoietic processes and phenotypes in those that exhibit larger increases in RNA secondary structure around the start codon, particularly when comparing cells in the later stages of red blood cell development with undifferentiated control samples.

Next, we used the change in the average RNA secondary structure score in the 200-nt window around the start codon to partition the transcripts into six distinct clusters. Each of the six clusters showed

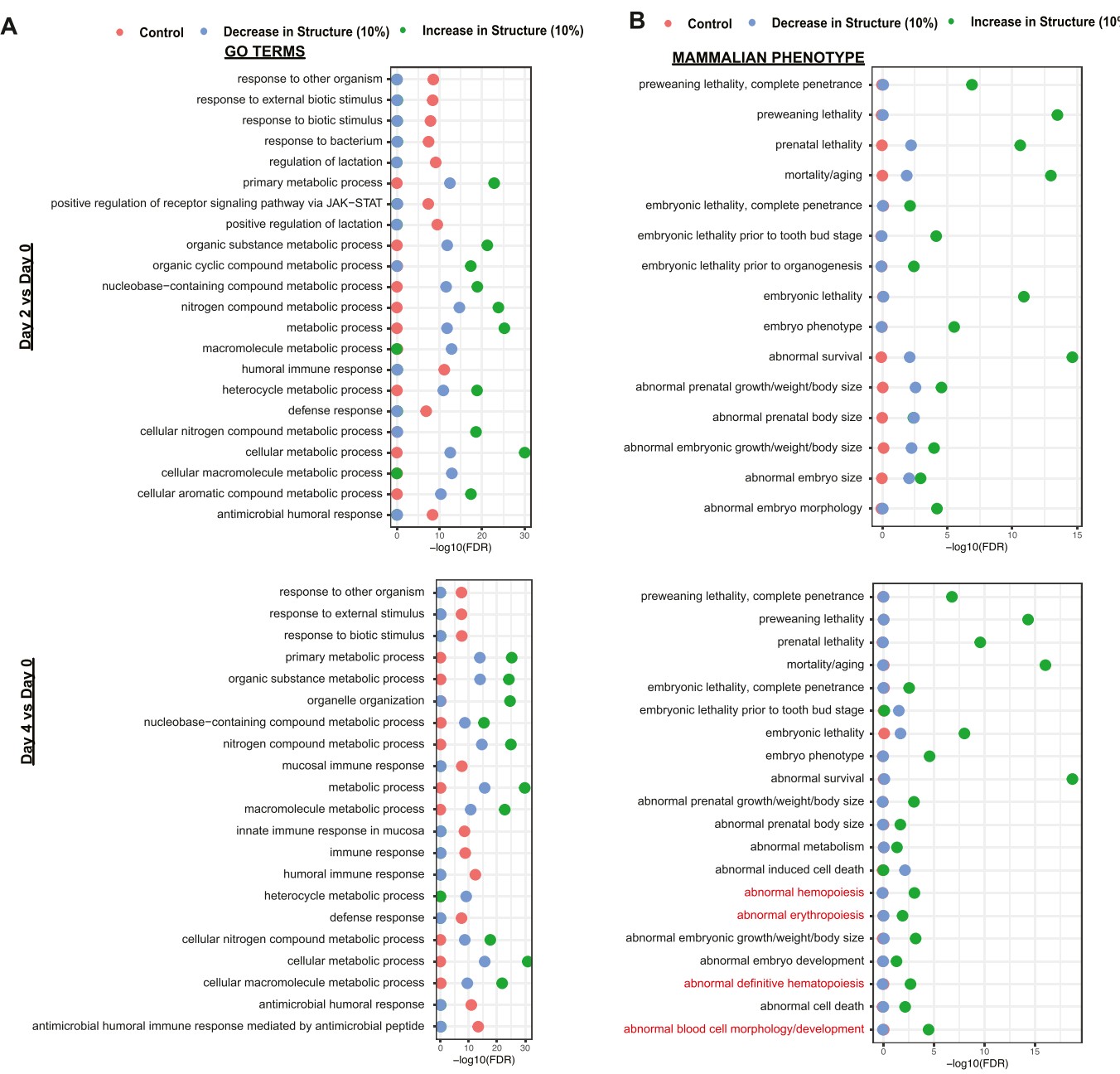

**Figure 4. Transcripts with the most change in RNA secondary structure are associated with hematopoietic phenotypes.**
**(A)** $-\log_{10}$(false discovery rate) of Gene Ontology enrichment associated with three sets of transcripts: control set of transcripts (45–55th percentile of change in RNA secondary structure score; red dots), transcripts with the highest 10% decrease in RNA secondary structure score (green dots), and transcripts with the highest 10% of increase in RNA secondary structure score (blue dots). Change is calculated between day 2 and day 0 cells (top) and day 4 and day 0 cells (bottom). **(B)** Mammalian phenotypes associated with three sets of transcripts: control set of transcripts (45–55th percentile of change in RNA secondary structure score; red dots), transcripts with the highest 10% decrease in RNA secondary structure score (green dots), and transcripts with the highest 10% increase in RNA secondary structure score (blue dots). Change is calculated between day 2 and day 0 cells (top) and day 4 and day 0 cells (bottom). Hematopoietic related terms are highlighted in red.

different dynamics in the pattern of changes in RNA secondary structure around the start codon. However, the changes in RNA secondary structure did not appear to correlate with changes in mRNA abundance or in PPS coverage. In fact, when comparing the mRNA abundance patterns for all six clusters, we noticed that they all exhibited a similar pattern despite how their RNA secondary structure changed throughout development (Fig 5A). In addition,

most of the transcripts showed no change in how much of the transcript is covered by RBP binding events, irrespective of the change in RNA secondary structure. This lack of correlation among RNA secondary structure conformation, RBP–RNA interaction, and mRNA abundance suggests that these parameters do not have a cause-and-effect relationship and appear to be largely independent of each other when interrogated on a global scale.

When we examined what biological processes the proteins encoded by the transcripts in each of the clusters were involved in, cluster 1 showed no significant enrichment for terms despite being the cluster with the highest number of transcripts. Cluster 2, the cluster in which transcripts increased in RNA secondary structure in terminal differentiation, was enriched in various metabolic processes as metabolic process (FDR: $5.46 \times 10^{-33}$) and cellular macromolecule metabolic process (FDR: $2.82 \times 10^{-26}$). The next cluster with the most significant terms was cluster 3, which contained transcripts that increase and then decrease in RNA secondary structure in the developmental window that we probed. Cluster 3 showed an enrichment in the same terms as cluster 2, however to a lesser degree. Cluster 4, which has transcripts going from unpaired to a more paired state, and cluster 5, its converse, have no significant terms to note, potentially because of the small number of transcripts that fall within either cluster (69 for cluster 4 and 208 for cluster 5). Transcripts in cluster 6 started off being paired, changed to a more unpaired state upon differentiation, and then somewhat increased their overall RNA secondary structure score in the latest time point. These transcripts showed a very minor enrichment for the same terms that are also found in clusters 2 and 3 (Fig 5B). Overall, changes in secondary structure surrounding the mRNA start codon do not appear to be a good indicator of encoded protein function in the process of terminal erythropoiesis.

However, once we interrogated the clusters for any associated MPs, we found that cluster 2 was enriched for transcripts that, when mutated, resulted in mice with abnormal blood phenotypes such as the more general hematopoietic system phenotype (cluster 2 FDR: 0.001) or the more specific abnormal erythropoiesis (FDR: 0.037) and abnormal blood cell morphology/development (FDR: $1.88 \times 10^{-5}$) (Fig 5C). This observation suggests that although the transcripts undergoing structural rearrangement, particularly those that increase in RNA secondary structure throughout development, are not enriched in those that function specifically in erythropoiesis, they are the ones that would likely lead to abnormal phenotypes if mutated. In total, our findings reveal that RNA secondary structure and RBP-binding events are dynamic throughout the development of mammalian red blood cells. In addition, we observe an anti-correlation of mRNA secondary structure and RBP-binding events around the translation start and stop codons. This anti-correlation parallels what we have observed in other eukaryotic transcriptomes (20, 21).

### Identifying RBPs as potential post-transcriptional regulators of erythropoiesis

As we've confirmed the presence of known RBP-binding sequences in our PPSs, we next sought to leverage our collection of PPSs to discover additional overrepresented sequences, which could be new binding sequences for RBPs that are key regulators of terminal erythropoiesis in our model. To do so, we first isolated protein interaction sites that overlapped the 50–100-nt window immediately downstream of annotated stop codons. We decided on this region because the large majority (~90%) of annotated 3′ UTRs are at least 50 nts long and that window encompassed a region of change in RNA secondary structure where structure increases in induced cells when compared with uninduced cells (Fig 3). We then used the de novo motif finding algorithm HOMER (38) to identify

several significantly enriched motifs in these sequences. The online database RBPDB (29) provided a resource for matching de novo identified motifs against the RNA recognition sequences of known RBPs. Among the motifs enriched in our collection of PPSs (Figs S10–Figs S12), we identified the RNA recognition sequences for the RBPs ELAVL1, PABPC1, FUS, and KHDRBS3 (Fig 6A), suggesting that these proteins may be involved in the post-transcriptional regulation of red blood cell development. In fact, knockdown of ELAVL1 induces a variety of hematopoietic abnormalities, including abnormal definitive hematopoiesis and decreased erythroid progenitor cell number (25, 28, 39, 40, 41). PABPC1 can bind to the 3′ poly(A) tail of mRNAs as well as interact with the cap-binding complex subunit eIF4G to facilitate mRNA translation and PABPC4, a related protein, has been shown to play a critical role in erythroid differentiation (7, 16, 42). FUS has been identified to contribute to the maintenance of hematopoietic stem cells and FUS-deficient mice also exhibit abnormal hematopoietic phenotypes such as decreased B cell numbers, although no erythropoietic specific phenotypes have been reported (43, 44). These findings suggest that the enriched sequence motifs in our PPS datasets are bound by a collection of RBPs, a number of which are known to function in mammalian erythropoiesis. Thus, the other RBPs with enriched binding sequences in our PPS datasets in the terminal stages of MEL development may be good candidates for further testing of functionality in this important developmental process.

### Identifying novel RBP-bound RNA motifs

As only a small subset of our PPSs contained binding sequences of known RBPs, we leveraged the entire collection of PPSs to identify additional RBP interaction sequences that have not been previously identified. For this analysis, we selected the 400–500-nt window downstream of the stop codon because this region showed dramatic changes in RNA secondary structure and RBP-binding density and was away from the CDS and possible influences of ribosomal complexes. In that window, the average RNA secondary structure increased (i.e., becomes more double-stranded) in differentiated cells and the PPS density decreases at day 2 and then increases to a level higher than day 0 by day 4 (Fig 3). These changes in RNA secondary structure and PPS density suggest the presence of RBP–RNA interactions in the region that are independent of ribosome binding, which makes this area a region of interest for identifying enriched motifs. Our HOMER motif enrichment analysis of high confidence (found in both biological replicates) PPSs detected 10 statistically enriched motifs for day 2 PPSs (n = 122) (Fig S13) and 12 statistically enriched motifs in day 4 PPSs (n = 253) (Fig S14). From these sets, we selected two motifs (one from each time point) for further study (Fig 6B and C). Each selected motif ranked in the top three based on the percentage of bound mRNAs that contained the RBP interaction sequence and the *P*-value associated with the motif. We also took into account the information content of each nucleotide in the motif and tried to maximize the number of positions with high information content in each motif. Both selected motifs showed an increase in RNA secondary structure around the beginning and throughout the motif site at day 2, followed by a decrease in structure in day 4, although the

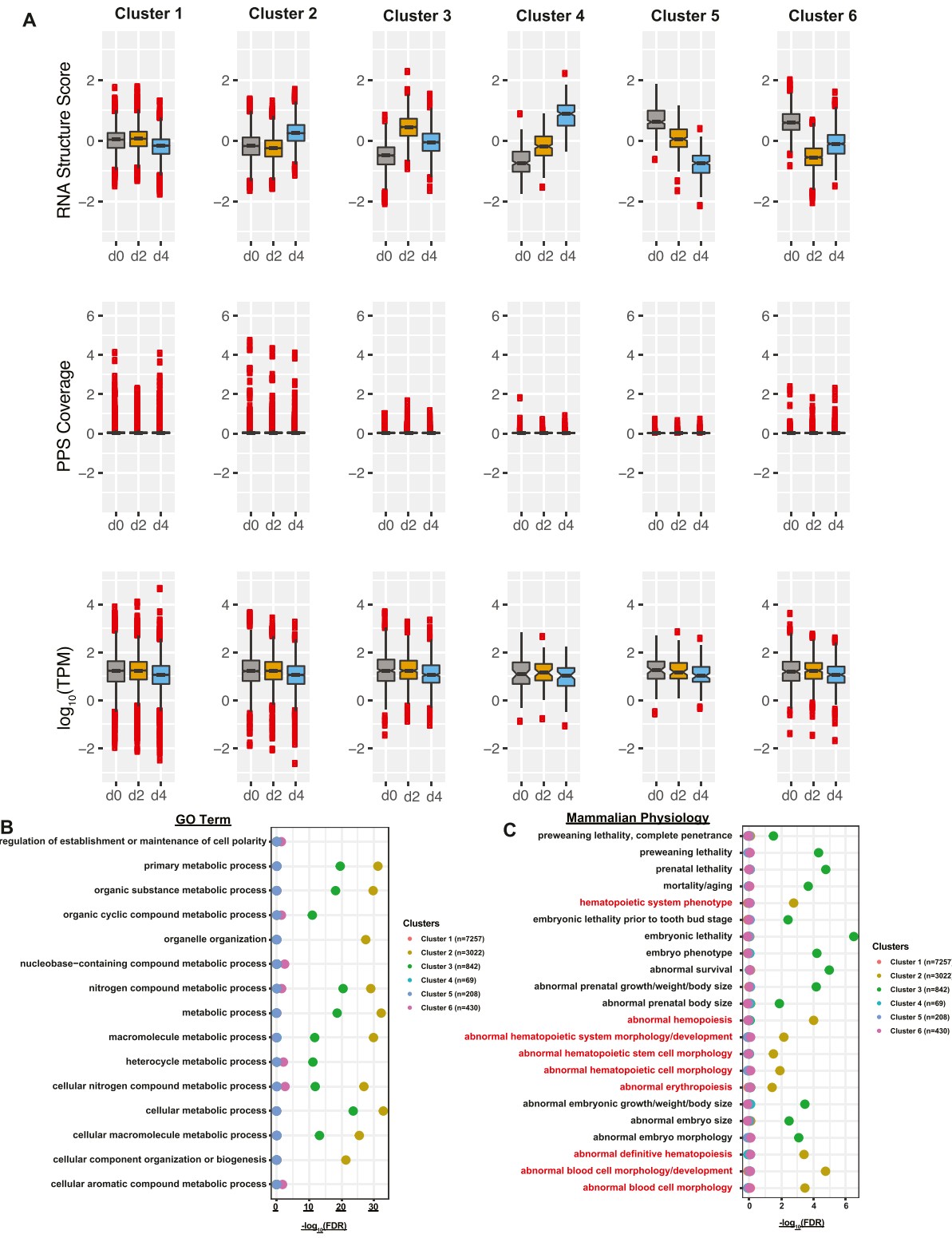

**Figure 5. Hierarchical clustering of transcripts based on their patterns of RNA secondary structure score during MEL cell differentiation.**
**(A)** The RNA secondary structure score (top), protein-protected site coverage (middle), and normalized mRNA abundance (bottom) distribution of transcripts in each of the six clusters identified by hierarchical clustering at day 0 (gray box), day 2 (yellow box), and day 4 (blue box). **(B)** –log$_{10}$(false discovery rate) of Gene Ontology enrichment of the transcripts identified in each cluster. Erythropoiesis related terms are highlighted in red. **(C)** –log$_{10}$(false discovery rate) of any mammalian phenotypes associated with transcripts in each of the clusters. Erythropoiesis related terms are highlighted in red.

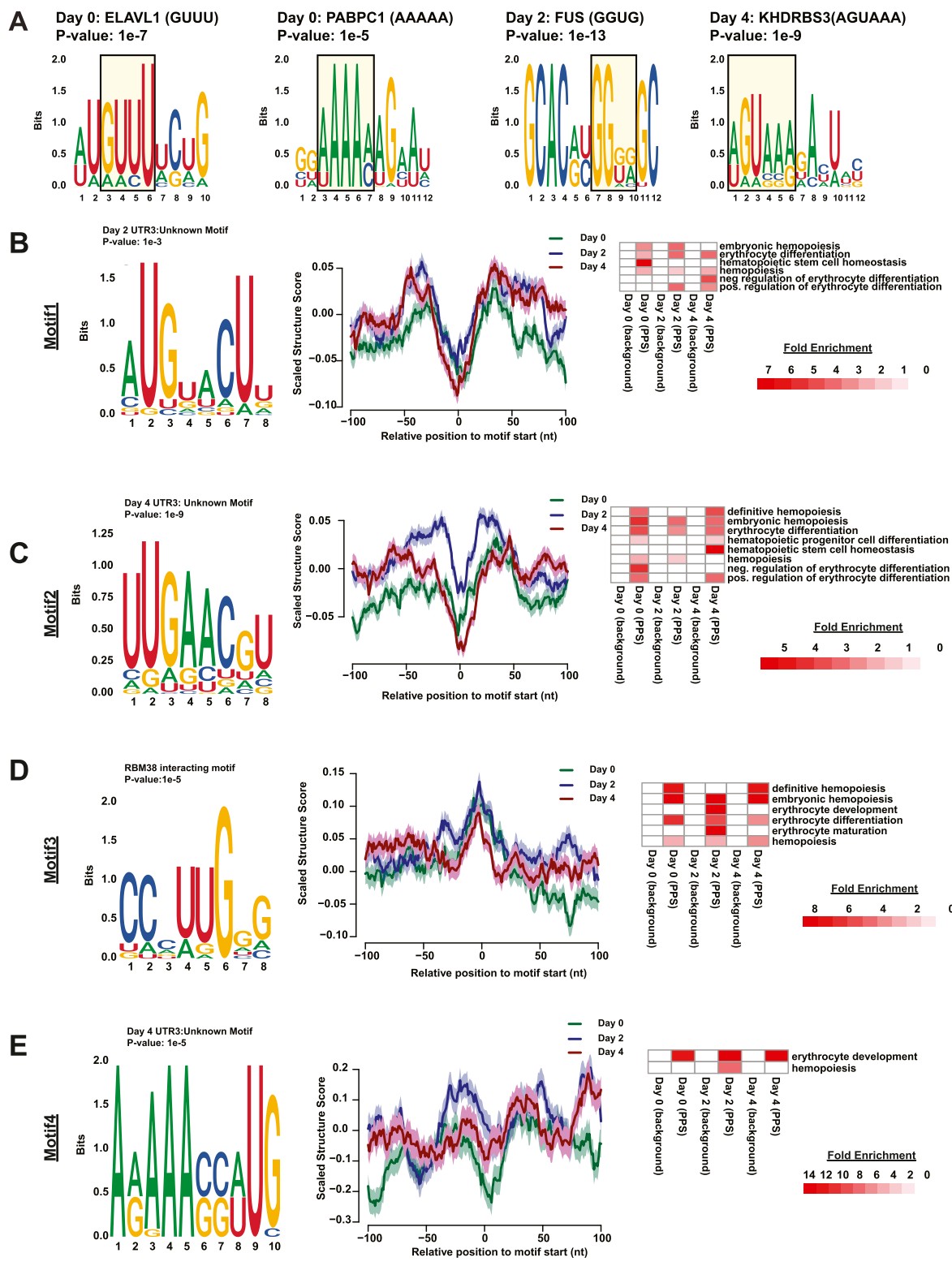

**Figure 6.  RNA-binding proteins as post-transcriptional regulators of erythropoiesis.**
**(A)** Selected motifs identified from time point specific protein-protected sites (PPSs) found 50–100 nt downstream of the stop codon in 3′ UTRs. Motifs containing the RNA recognition motifs of known RNA-binding proteins are annotated with the protein name. **(B, C, D, E)** Selected motif (left) enriched in (B) PPSs found in the 3′ UTR of day 2 post induction PPSs, (C) PPSs found in 3′ UTR of day 4 post induction PPSs, (D) PPSs in MEL cells overlapping known RBM38-binding sites (46), and (E) PPSs found in both biological replicates located in the 3′ UTR of erythropoiesis annotated genes as well as RNA secondary structure profile (middle) at each nucleotide ± 100 nt from the predicted start of the motif in MEL PPSs before (green line), 2 d (blue line), and 4 d (red line) after induction of MEL differentiation. Fold enrichment value erythropoiesis related Gene Ontology terms as compared to a controlled background on the left.

trend was more obvious in the Motif 2 (day 4 3′ UTR motif) than Motif 1 (day 2 3′ UTR motif) (Fig 6B and C). An additional motif was selected do to its close proximity to known binding sites of RBM38, which is an RBP that has a well-documented role in regulating alternative splicing during terminal erythropoiesis (45) and has recently been shown to regulate translation during terminal erythropoiesis (46). To identify this enriched sequence motif, we used a set of 3,359 3′ UTR high confidence PPSs that all contained the known RBM38-binding motif and identified the 92 PPSs that directly overlapped previously identified RBM38-binding sites (38) by at least one nucleotide. These 92 PPSs were then analyzed using HOMER for enriched motifs, yielding a total of 11 significant motifs (Fig S15). From this group of 11 sequences, we selected the most significantly enriched motif that was also found in the most RBP bound target sequences as determined by our PIP-seq analysis (Fig 6D). Interestingly, this selected motif (motif 3) was identified as being bound by TAL1, which is known to be involved in erythrocyte differentiation (47); TRIM10, a protein needed for globin gene transcription (48); and other proteins of importance to erythropoiesis. A further analysis of the PIP-seq data revealed that the motif occurs in a region where RNA secondary structure increased during mammalian red blood cell development, suggesting it as a potential binding sequence for an uncharacterized RBP that prefers to bind RNA in a paired conformation (double-stranded RBP [dsRBP]) (Fig 6D). Although the overall analysis of RNA secondary structure profiles and RBP-binding densities suggests a general anti-correlation between these two features, there are RBPs that preferentially bind to double-stranded RNAs and perform a variety of post-transcriptional regulatory actions (49). Notably, our mRNA-seq data detected the significant increase in the levels of 11 mRNAs encoding dsRBPs when comparing day 4 RNA abundance to undifferentiated (day 0) MEL cells (Table S1). Given the existence of dsRBPs in the mammalian genome and the up-regulation of several of them throughout erythropoiesis, we included this motif on the basis of trying to identify RBPs with the potential for interacting with areas of increased structure.

An additional approach to identify RBP-binding motifs that are functionally important for the post-transcriptional regulation of erythropoiesis is to start with PPSs found specifically in transcripts identified relevant to red blood cell development. Using the Ensembl database and the R biomaRt tool (version 2.40.3), we identified 248 mRNAs encoding proteins that make up the categories labeled erythropoiesis or hematopoiesis in the current mouse GO annotation. To test the accuracy of the annotation, we analyzed the GO annotation of these identified transcripts using DAVID (31) and verified that they are indeed related to erythropoiesis by subjecting the list to GO analysis (Fig S16A). We then performed motif enrichment analysis on 34 high confidence day 4 PPSs found in the 3′ UTRs of these transcripts. Of the 11 enriched motifs (Fig S16B), we were particularly interested in the poly adenine (polyA) motif identified through this process (motif 4) (Fig 6E), as it ranked in the top three in terms of abundance in the target sequences and was also statistically significant. RNA secondary structure analysis around the motif start site revealed an increase in structure in differentiated cells when compared with day 0, which matched the overall trend observed in the conglomeration of all transcripts (Fig 6E).

After identifying these four motifs of interest (all motifs listed in Table S2), we then used reverse scanning on our entire collection of PPSs because we could not reliably restrict downstream RNA affinity tests to look at a specific genic region, to identify the number of total PPSs that contained each motif. This analysis revealed that motif 1 was found in a subset of 53,649 PPSs, motif 2 in 77,740 PPSs, motif 3 in 22,252 PPSs, and motif 4 in 2,296 PPSs. Although most of the PPSs were found within the intron region of transcripts, once the frequency was normalized to the overall distribution in the genome, PPSs were found to be positively enriched for these protein-bound motifs within the CDSs and both UTR regions of their respective transcripts (Fig S17A–D), similar to the overall observations made on the entire collection of PPSs. As with the broader set of PPSs, we see an under-enrichment of PPSs in the intronic region, which could be attributed to the splicing machinery removing most of the annotated introns during post-transcriptional transcript processing. As further evidence of their relevance in red blood cell development, when we examine transcripts containing the top 10% of matches as ranked by their respective log-odds score, we saw an enrichment for hematopoiesis relevant GO terms in comparison to the control set of transcripts that was matched for strand and chromosome bias (Fig 6B–E). In total, our findings revealed that identifying new protein-bound motifs using PIP-seq allows us to uncover protein-bound sequences in sets of transcripts that encode proteins important for the process of mammalian blood cell development. Thus, future studies can be designed to assess the roles of these putative protein-binding sites as novel posttranscriptional regulatory sites of erythroid differentiation.

## Identifying potential posttranscriptional regulators of erythropoiesis

We selected a small subset of motifs (Motifs 1–4) for downstream identification of their corresponding RBPs. These motifs were selected on the basis of being novel, being located in RNA regions with dynamic RNA secondary structure(s), and being enriched in hematopoiesis relevant transcripts. RNA affinity chromatography was followed by mass spectrometry analysis. In this technique, we covalently attached a synthetic RNA motif to agarose beads. We then incubated these RNA baits, as well as a bead-only control, with whole MEL cell protein lysates from the three developmental time points. The beads were then stringently washed and tightly bound proteins were pulled down and then identified via mass spectrometry (MS). Using this approach, we detected the presence of ~647 proteins (Table S3) that potentially interacted with at least one of the four different motifs.

To identify proteins that showed a stronger binding to a certain probe, we performed a fold change analysis on data normalized to the sample mean. To do this, we calculated the fold change in the normalized values for the protein of interest from that particular motif compared to the other three motifs and selected only proteins that were >10-fold enriched in at least one motif as compared with other probes. This analysis identified a smaller subset of proteins (n = 315) as being >10-fold enriched in at least one motif as compared with other probes (Table S4 and Fig S18A). Specifically,

motif 1 had 156 proteins bound by proteins in lysates from the post-differentiation time points, motif 2 had 132 proteins bound by proteins in the 2- and 4-d post-differentiation lysates that were not identified in lysates from day 0, motif 3 had 18 proteins bound by proteins in lysates from day 4 but not day 0, and motif 4 had 71 proteins bound with protein lysates from day 2 or day 4. When we further required the proteins to be bound ≥ 10-fold at all tested time points (including day 0), the number significantly dropped so that neither motif 1 nor motif 2 had proteins matching that criterion, whereas motif 3 had 4 such proteins and motif 4 had seven proteins that were enriched at both day 2 and day 4.

As confirmation of the biological relevance of our approach, we observed a nice enrichment of RNA-binding proteins in the 315 proteins that were identified as interacting with at least one motif (Fig 7A). Interestingly, the RNA abundance of the transcripts encoding the proteins identified in the mass spectrometry showed an overall decrease in abundance, which was significant in the terminal stages of erythropoiesis (*P*-values < 0.05) (Fig S18B). A protein domain enrichment analysis revealed that the most prevalent protein domain is the RNA recognition motif domain (Fig S18C). GO analysis of these proteins showed that 54 are annotated as being RBPs (Bonferroni *P*-value: $5.26 \times 10^{-22}$, hypergeometric test compared with default background) and 95 have the more general description of being nucleotide-binding (Bonferroni *P*-value: $4.47 \times 10^{-46}$, hypergeometric test compared with default background). Although many terms relevant to RNA binding were significantly enriched with Bonferroni values < 0.05, the most significant terms were RNA binding and poly(A) RNA binding (Fig 7B). GO analysis of the RNA-binding proteins showed that a large number of them were associated with regulation of splicing (Bonferroni: $8.86 \times 10^{-13}$, hypergeometric test) or translation (Bonferroni: $3.54 \times 10^{-4}$, hypergeometric test). Specifically, proteins pulled down by motifs 3 and 4 were highly enriched for those that are part of the mouse spliceosome, whereas those enriched in motifs 1 and 2 are involved to a smaller degree with RNA transport (Fig 7B). This enrichment for proteins involved in alternative splicing in our mass spectrometry data highlights the potential of alternative splicing as a key post-transcriptional regulation mechanism in mammalian erythropoiesis.

To further narrow the list of interacting proteins to those that are relevant to red blood cell development, we focused on RBPs that were enriched only after differentiation (i.e., 2 or 4 d after DMSO induction) but not in day 0. Motif 1 failed to pull down any proteins matching this criterion while DKC1 was pulled down using motif 2 in both day 2 and day 4 samples. Motif 3, which is a potential binding motif for an RBM38 interacting protein, had five proteins (EIF4G1, PTBP1, DDX17, EIF4A3, and FXR3) that matched the requirement and motif 4, the poly A motif discovered in erythropoiesis relevant transcripts, had another five such proteins (HNRNPF, HNRNPH1, HNRNPH2, LARP1, and PURA) (Fig 7C). A Kyoto Encyclopedia of Genes and Genomes (KEGG) pathway enrichment analysis showed an enrichment for proteins associated with RNA transport and the splicing machinery in the set of proteins that were associated with the motifs. The most significant enrichment for the spliceosome observed in motif 4 (Fig 7D).

More interestingly, when we surveyed the Mouse Genome Database (17, 18, 50) for phenotypic information on mice models with the 11 proteins that were enriched at the later developmental time

points as compared with day 0, we found that abnormal levels of DKC1, HNRNPH1, and PURA result in abnormalities in the hematopoietic system. Specifically, PURA is associated with reticulocytopenia (51) and *Pura* deletion is detected in patients with acute myelogenous leukemia, linking this protein to a role in mammalian hematopoiesis (52). Linkages were observed with two additional RNBPs; HNRNPH1 knockdown mice see a decreased mean corpuscular volume and decreased average hemoglobin concentration (53) and DKC1 deficient mice experience abnormal erythrocyte morphology along with decreased hemoglobin levels (54).

Although LARP1, which was identified as being enriched in pulldowns using the poly(A) motif 4, showed no identified abnormal hematopoietic phenotype in mouse models, it is a RBP recently noted to be lost in the 5q⁻ syndrome, a type of macrocytic anemia caused by monoallelic deletion of a region that encompasses the *Larp1* gene in humans (55). Furthermore, depleting LARP1 levels in CD34[+] bone marrow precursor cells led to a reduction in 5′ TOP mRNA levels, p53 induction, and most interestingly, an anemic phenotype (56).

HNRNPH1, HNRNPF, HNRNPA3, and HNRNPH2 are all members of the heterogenous nuclear ribonucleoprotein family. As a family, hnRNPs are RBPs that interact with heterogenous nuclear RNA and function in various steps of RNA processing. HNRNPH1, HNRNPF, and HNRNPH2 are very similar to each other with amino acid sequence similarity ranging from 76% to 95%, which makes it challenging to identify which of the proteins is specifically enriched. The Human Protein Atlas detects high levels of antibody staining for all three proteins in hematopoietic cells and RNA abundance levels, measured at 102.6 TPM (HNRNPF), 175.2 TPM (HNRNPH1), and 70.1 TPM (HNRNPH2), respectively, were also correspondingly high. Mice heterozygous for *Hnrnph1* have phenotypic features such as decreased mean corpuscular hemoglobin concentration and increased mean corpuscular volume according to the Mouse Genome Database (50). Using Z-scores as a measurement of membership by association to co-expressed genes associated with the phenotype, the ARCHS[4] database predicted HNRNPF to be associated with abnormal hematopoietic system (Z-score: 2.982) in mice and reticulocytopenia (Z-score: 3.9), abnormality of cells of the erythroid lineage (Z-score: 3.497), and abnormal number of erythroid precursors (Z-score: 3.359) (57, 58). These findings suggest that we have identified bona fide RNA-bound sequence motifs relevant to mammalian red blood cell development. By identifying the proteins interacting with these important regulatory sequences, we have identified known and potentially new posttranscriptional regulators of mammalian hematopoiesis.

### DKC1 interacts with Appl2 and Dido1 in terminal erythropoiesis

We identified DKC1 as a protein of interest based upon its enrichment in the RNA affinity pulldown experiment using motif 2 (Fig 7C) that is found in the 3′ UTR of MEL cells 4 d after DMSO induction. Furthermore, mice deficient in DKC1 develop hematopoietic symptoms including bone marrow failure, anemia, decreased hemoglobin content, etc. (54, 59). More recently, it was identified as a target of GATA1 and is noted to be an important player in the increase of telomerase activity in human erythropoiesis (60). In humans, DKC1 is associated with X-linked dyskeratosis congenita

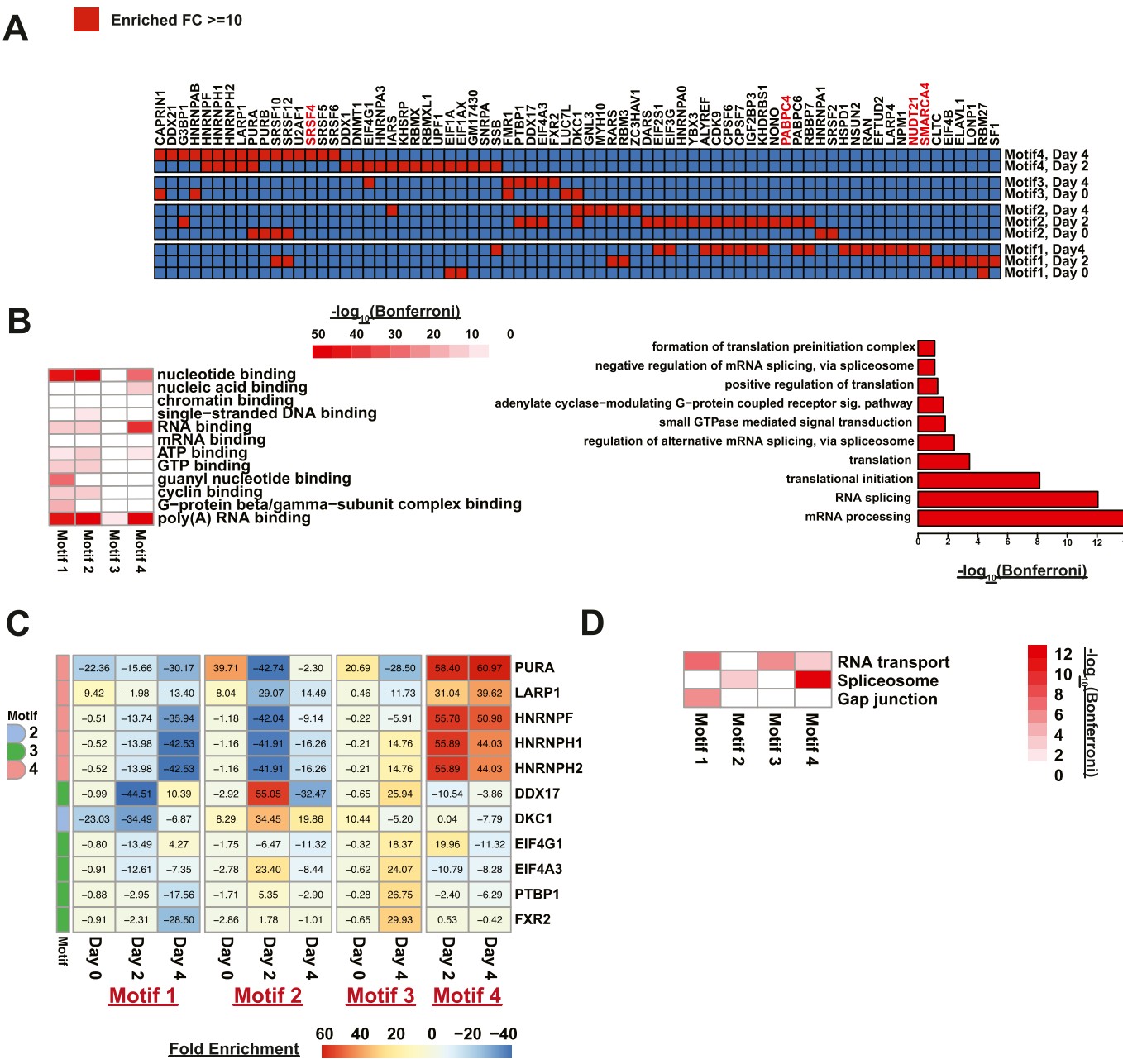

**Figure 7. Identification of RNA-binding proteins (RBPs) that interact with motifs enriched in areas of large-scale secondary structure and RBP binding during mammalian erythropoiesis.**
**(A)** Heat map of RBPs identified as being enriched with ≥ 10-fold (red cell) in each comparison when compared with other probes in at the same time point. Proteins with functions relevant to erythropoiesis are written in red. **(B)** −log₁₀(Bonferroni) values of RNA binding related Gene Ontology terms enriched in proteins detected using motifs 1–4 as bait in RNA affinity pulldowns followed by mass spectrometry (left). Bar plot showing the fold enrichment of pulled down RBPs' annotated biological processes (left). **(C)** Heat map of RBPs enriched in day 2 or day 4 pulldowns. Text inside cell is the fold enrichment against other probes (i.e., probe iBAQ value/average [other probes' iBAQ value]). Colored boxes on the left indicate what motif the protein is most strongly associated with. **(D)** KEGG pathway analysis of proteins identified as being enriched for binding to Motifs 1, 2, 3, and 4 (as noted).

and patients exhibit abnormal blood phenotypes such as anemia, leukopenia, acute myeloid leukemia, and thrombocytopenia (61).

Based on these phenotypes and our RNA-binding observations, we selected this protein for further study. Specifically, we sought to validate some of the transcripts predicted to interact with this RBP based on sequence similarity to the motif with which we found it

interacts (motif 2). We selected two transcripts (Dido1 and Appl2) that contained at least one matching motif and were annotated to exhibit abnormal erythrocyte phenotypes when mutated in mouse models (62). We performed RNA immunoprecipitation (RIP) using DKC1 as the protein of interest and ACTIN as the control protein to collect the pool of mRNAs that interact with each of the proteins. We

then used these bound RNA fractions in quantitative reverse transcriptase PCR (qPCR) experiments to validate our hypothesis that these target transcripts would demonstrate an increased enrichment in the DKC1 pulldown when compared with the ACTIN pulldown as the case would be if DKC1 is indeed interacting with these two transcripts. We first performed Western Blots on the RIP lysates to confirm that that we pulled down the DKC1 and ACTIN proteins (Fig S19A). We then used qPCR, with probes designed against Dido1, Appl2, Aff1, and Adar to test the binding of DKC1 and ACTIN to each of these four transcripts. Whereas Dido1 and Appl2 were predicted targets of DKC1, DKC1 was not predicted to interact with either Aff1 or Adar, so they served as negative control transcripts in this experiment, whereas ACTIN serves as the negative protein control. In short, our hypothesis predicted that we would see an enrichment in the amount of interaction between DKC1 and Dido1 and Appl2 but not Aff1 or Adar. Based on our hypothesis, the interaction between DKC1 and its predicted targets should also be greater than the interaction observed between ACTIN and the same target RNAs. We also performed a pulldown with IgG and used that pulldown as a baseline to detect the level of interaction between our protein(s) of interest and our target transcript(s) of interest. In concordance with our predictions, the qPCR data revealed significant enrichment in the interaction between $\alpha$-DKC1 and its target transcripts when compared with the $\alpha$-ACTIN negative control pulldown (Fig S19B) (P-values < 0.05; Wilcoxon rank sum test). We did not observe the same level of enrichment in the interaction between DKC1 and the negative control transcripts Aff1 and Adar, validating our predictions.

When we examined our mRNA-seq data, we found that Appl2 shows a continual and significant increase in RNA abundance throughout MEL development, whereas Dido1 mRNA steady-state levels remain relatively constant throughout this developmental time period (Fig S19C). Thus, the binding of DKC1 to these transcripts could be involved in maintaining their levels throughout this developmental process by protecting them from degradation. In total, our RIP-qPCR findings validated our approach of first identifying enriched motifs, then using RNA affinity to identify associated proteins, followed by using RIP-qPCR to validate predicted mRNAs that interact with the proteins. Specifically, we identify with this approach the potential binding sequence of DKC1, a protein that could be a regulator of erythropoiesis, and identify Appl2 and Dido1 as two of its potential targets, laying the foundation for future studies on the function of this RBP in mammalian erythropoiesis.

## Discussion

In this study, we have applied the high-throughput sequencing technology PIP-seq to a model of erythropoiesis and characterized the transcriptome-wide landscape and dynamics of RNA secondary structures and RBP occupancy. Our data suggest that PPSs, which are potential sites of RBP occupancy, are more conserved throughout evolution, possibly because of their biological function. This conservation also aligns well with the idea that RNA sequence is one of the factors that determines RBP binding (3). The

importance of RBPs in erythropoiesis is further borne out by the observations that transcripts with PPSs detected post-differentiation are functionally enriched for those that are involved in erythrocyte development (Fig 2C) and the transcripts that show dynamics in RBP occupancy, as measured by changes in the percentage of a transcript covered by PPSs, are enriched for those connected with abnormal hematopoietic phenotypes (Fig 2E).

We further note that throughout development, we observe an increase in the average RNA secondary structure (Fig 3A) in the 400-bp window surrounding the start and stop codon, suggesting that subsets of RNAs are transitioning from being single-stranded to being more double-stranded in nature. Hypothetically, this could then influence the interaction between RBPs and a particular subset of RNAs as binding sites are rendered inaccessible, which could be a mechanism by which the cell regulates the fate of those RNA molecules. In support of that hypothesis, we further observe that RNAs with the most dramatic changes in RNA secondary structure are, indeed, enriched for those that function in erythropoiesis (Fig 4). In the same 400-bp window around the start and stop codon, we observe the general trend that RBP occupancy and RNA secondary structure seem to be anti-correlated, perhaps suggesting that the RBPs active in erythropoiesis tend to largely prefer single-stranded RNAs (Fig S9). However, when we examine specific clusters of RNAs, the anti-correlation between RNA secondary structure and RBP occupancy vanishes (Fig 5), which could potentially suggest that the anti-correlation is on the global scale, but individual subsets of transcripts are exceptions to this general trend.

One application of PIP-seq data is using motif enrichment analysis followed by RNA affinity pulldown and mass spectrometry to identify potential post-transcriptional regulators of erythropoiesis. Using this process, we were able to identify several proteins that could potentially be regulators of erythropoiesis. Specifically, the KEGG pathway enrichment analysis revealed an enrichment for proteins involved in splicing and RNA transport, which tracts well with the idea that RNA splicing plays a large role in terminal erythropoiesis (63), and suggests that alternative splicing would be an important regulatory process to understand for erythropoiesis. However, we were interested in identifying specific proteins that may act as regulators of this process. By using RIP-qPCR, we validated that DKC1 could be a potential regulator of Appl2 and Dido1, both of which are found at significant levels throughout red blood cell development (Fig S19C).

Taken together, these data establish a comprehensive database of in vivo RNA secondary structure and RBP–RNA interactions for the important process of mammalian erythropoiesis. In total, our findings reveal the power of using a global genomic screen of RNA secondary structure and RNA-protein interaction site dynamics using PIP-seq to identify potential new post-transcriptional regulators of an important developmental process. In addition, these data provide a resource for future studies that can focus on identifying corresponding functions and novel pathways of post-transcriptional control during terminal erythroid differentiation. In the future, these newly described proteins and corresponding collections of target RNAs will be further studied to better understand the mechanisms by which they regulate this important mammalian developmental process.

# Materials and Methods

## Cell culture and differentiation

MEL cells were grown under standard conditions in minimal essential medium (MEM) and supplemented with 10% (vol/vol) FBS and 1× antibiotic antimycotic (Invitrogen). MEL cells in suspension culture at the log phase of growth at a density of $2 \times 10^5$/ml were supplemented with 2% DMSO (Sigma-Aldrich) to induce differentiation, and cells were collected at various time points for further analysis.

## PIP-seq library construction and read mapping

PIP-sequencing libraries were constructed as outlined in Silverman and Gregory (2015) ([19]). Briefly, MEL cells were induced to differentiate and then collected 0-, 2-, and 4-d post-DMSO induction. Whole cell samples were treated to 1% formaldehyde solution under vacuum to cross-link RBP–RNA interactions. The reaction was then quenched by vacuum infiltrating 125 mM glycine into the sample, followed by washing with ddH$_2$O. Then each sample was split into four libraries: two for structure only libraries and two for footprinting libraries. Footprinting libraries were treated with either 100 U/ml of a ssRNAse (RNaseONE [Promega]) in 1× RNaseONE buffer for 1 h at room temperature, or 2.5 U/ml of a dsRNase (RNAse V1 [Ambion]) in 1× RNase buffer for 1 h at 37°C as previously described ([8]). Protein denaturation and digestion was carried out by treating the samples with 1% SDS and 0.1 mg/ml Proteinase K (Roche) for 15 min at room temperature, followed by 2-h incubation at 65°C to reverse the cross-linking. The structure libraries were also constructed in a similar fashion, except that cross-linked lysates were treated with 1% SDS and 0.1 mg/ml Proteinase K (Roche) and then subjected to ethanol precipitation first. Then the samples underwent their respective RNAse treatments. RNA from the four samples (two footprinting libraries and two structure libraries) were then isolated using the QIAGEN miRNeasy RNA isolation kit per manufacturer protocol (QIAGEN). Then the RNA underwent strand-specific sequence library preparation as previously described and the resulting four libraries for each sample (footprinting-dsRNase, footprinting-ssRNase, structure-dsRNase, and structure-ssRNASE) were sequenced using Illumina HiSeq2000 following the standard protocol for 50-bp single read sequencing.

PIP-seq reads were trimmed using cutadapt to remove 3′ sequencing adapters (cutadapt version 1.2.1 with parameters –e 0.006 –O 6 –m 14). Resulting trimmed reads were then collapsed into unique reads and aligned to the mm10 mouse genome sequence using TopHat (version 2.0.10 with parameters–library-type fr-secondstrand–read-mismatches 2 –read-edit-dist 2 –max-multihits 10 –b2-very-sensitive–transcriptome-max-hits 10 –no-coverage-search–no-novel juncs). Any PCR duplicates were collapsed to single reads for all downstream analysis.

## mRNA-seq library construction, processing, and alignment

mRNA-seq libraries were constructed in biological replicates, following standard library construction protocol. Briefly total RNA was extracted using Trizol Reagent as per the manufacturer's protocol and then sequenced using TruSeq stranded mRNA-seq. Adapters were trimmed using cutadapt with the following parameters (-f fastq -a [ADAPTER] -e 0.06 -O 6 -m 14) and then aligned against the mm10 genome using TopHat (v2.1.0) with the following parameters (–library-type fr-secondstrand –read-mismatches 2 –read-edit-dist 2 –max-multihits 10 –b2-very-sensitive –no-coverage-search -p 15).

## mRNA-seq differential gene analysis

Unadjusted read counts from the TopHat alignment were inputted into the R package DESeq2 ([64]) and log$_2$ fold changes of MEL cells after 0, 2, or 4 d of DMSO induction were calculated. Normalized read count values are displayed using TPM values.

## GO enrichment and MP prediction

Various lists of transcripts were analyzed using DAVID ([30], [31]), using the entire list of transcripts with >1 FPM in at least one sample as the background for GO enrichment. Significant terms were those with a Bonferroni value <0.05. MPs and disease enrichment were predicted by submitting the same list of genes to the MouseMine ([62]) tool and then downloading the resulting lists of phenotypes or diseases using the same background as above, and then further using R to identify those with Bonferroni values <0.05 for subsequent visualization.

## Identification of PPSs

PPSs were identified using a modified version of the R package CSAR ([65]). Read coverage values were calculated for each base in the genome and a Poisson test was used to determine an enrichment score for footprint as compared with structure only libraries. PPSs were then called with a FDR of 5% as previously described ([8], [20], [21]).

## Random background PPSs

Background mock PPSs were generated by taking the high confidence PPSs and then randomly shuffling them in the transcriptome (i.e., regions annotated as being either UTR, CDS, exon, or intron) such that chromosome, feature size, and strandedness were preserved while averting any region that is called as a PPS in any sample. This was accomplished by using the bedtools shuffle feature with the following parameters (-incl -[annotated_gene_file] -excl [PPS_file_for_timepoint_anyrep] -chrom -noOverlapping) using the browser extensible data (BED) file of high-confidence PPSs for a particular time point as the input.

## Analysis of PPSs

PPS annotation was performed using the mm10 genome annotations in a "greedy" fashion such that all annotations overlapping a PPS by one nucleotide was counted equally (i.e., if a PPS overlapped both an miRNA and a CDS, both categories increased their count by 1). Conservation of PPSs was determined by comparing PhastCons scores and the number of SNPs within PPSs relative to equally sized flanking regions. PhastCon scores for PPSs compared with the

same-sized flanking regions were calculated as previously de-scribed (8, 21).

## Calculating the structure score statistic

For every base in our set of detectable transcripts, we calculated the ratio of dsRNA-seq and ssRNA-seq coverages from the structure only samples as previously described (60). Briefly, for every coverage value of dsRNA-seq ($n_{ds}$) and ssRNA-seq ($n_{ss}$) of a given base $I$, the structure score is calculated as follows:

$$S_i = glog(ds_i) - glog(ss_i)$$
$$= log_2\left(ds_i + \sqrt{1 + ds_i^2}\right) - log_2\left(ss_i + \sqrt{1 + ss_i^2}\right),$$

$$ds_i = n_{ds}\frac{max(L_{ds}, L_{ss})}{L_{ds}},$$
$$ss_i = n_{ss}\frac{max(L_{ds}, L_{ss})}{L_{ss}},$$

where $S_i$ is the structure score, $ds_i$ and $ss_i$ are the normalized read coverages, and $L_{ds}$ and $L_{ss}$ are the total covered length by mapped dsRNA-seq and ssRNA-seq reads, respectively. In this analysis, we used the total coverage length because we believe it is a more reasonable assumption for the transcriptome to have comparable levels of paired/unpaired regions. We used a generalized log ratio (glog) instead of the normal log-odds because of its ability to tolerate values of 0 (i.e., regions where there are neither ds- nor ssRNA reads). In addition, the glog function is still asymptotically equivalent to the standard log ratio when coverage values are large. Because we are only interested in the intra-molecular interactions that contribute to the self-folding secondary structure, we used only sense-mapping reads. Near the CDS boundaries, we calculated the structure score for up/downstream of the CDS start and end sites, aligned those to detectable mRNAs, and then averaged the score to produce the profile.

Identifying changes in RNA secondary structure was carried out by calculating the structure scores as outlined above and then filtering for the transcripts with a structure score at all time points. Hierarchical clustering was performed by first calculating the average RNA secondary structure score in the 200-nt window around the start codon for each transcript and then using the built-in hierarchical clustering function in R with default parameters to identify the six clusters. Transcripts from each of the cluster was then submitted to MouseMine for GO and MP analysis (62).

## RBP bound sequence motif identification and profiling secondary structure at these sites

HOMER (38) was used to identify any enriched motifs with parameters –rna –size gven–p 2, respectively. Using HOMER, we also mapped detected motifs back to the genome to identify every occurrence of the motif in our bound mRNAs using scanMotifsGenomeWide.pl with -bed option to format output into a BED6 format. Then we extracted a window of ± 100 nt up- and downstream of the predicted motif start position and plotted the structure score of nucleotides within that region as previously described.

## Searching for motifs in PPSs

The collection of PPSs was scanned for each of the motifs in order to identify which PPSs contained those sequences, and subsequently which encompassing transcripts, contained binding sites for putative RBPs using HOMER's scanMotifGenomeWide.pl command with the -bed option to format output into a BED file.

The list of sites containing each motif was then mapped back to their corresponding transcripts using genomic coordinates. The transcripts were then subsequently analyzed for GO enrichment using DAVID's default options against the mm10 background (30, 31).

## RNA affinity pulldown

MEL cells 0, 2, and 4 d post differentiation were collected and washed 2× with 1× PBS and lysed in 500 $\mu$L–1 ml of Lysis buffer with the addition of complete EDTA-free protease inhibitor (1 mini tablet per 10 ml; Roche). Cells were sheared with a 26-gauge needle and the supernatant was collected after spinning at 13,000$g$ for 10 min at 4°C. Samples were stored at –80°C. We then used motifs identified within PPSs of interest (Figs S9S–Figs S12) as "bait" to pull down any interacting proteins. The protocol used was identical to the one previously used in Foley 2017 (21). Briefly, the motifs were converted to RNA "baits" (IDT) and covalently linked to agarose beads before being incubated in a binding reaction (3.2 mM MgCl₂, 20 mM creatine phosphate, 1 mM ATP, 1.3% polyvinyl alcohol, 25 ng of yeast tRNA, 70 mM KCl, 10 mM Tris, pH 7.5, and 0.1 mM EDTA) with ~ 50 $\mu$g of protein lysate for 30 min at RT. Beads were then washed four times with GFB-200 (20 mM TE, 200 mM KCl, and 6 mM MgCl₂) and once with 20 mM Tris–HCL (pH 7.4). The proteins were then directly trypsinized on the beads.

## Preparing samples for mass spectrometry analysis

RNA-bound proteins were treated with 100 mM NH₄HCO₃ containing ~6 ng/$\mu$l of MS-grade trypsin (Promega) and incubated at 37°C overnight for 12–18 h. The samples were then vacuum dried before being submitted to MS analysis.

## Mass spectrometry analysis

Mass spectrometry analysis was performed similarly to previous reports (66, 67). Samples were desalted on in-house StageTips which were created by pushing a small punch of 3 M Empore C18 paper into a P200 pipette tip. StageTips were prepared by flushing with acetonitrile (ACN) and then 0.1% TFA. Samples were loaded in 0.1% TFA, washed with 0.1% TFA, and eluted with 0.1% TFA in 70% ACN. Samples were placed in a Savant SpeedVac SC100 to be dried and then resuspended in 0.1% TFA, the peptides were separated on a Dionex UltiMate 3000 with a C18 trap column and an in-house C18 analytical nanoflow column using 0.1% formic acid as buffer A and 0.1% formic acid in 80% ACN as buffer B. Peptides were analyzed by a Thermo Fisher Scientific Q Exactive HF Hybrid Quadrupole-Orbitrap Mass Spectrometer using higher-energy collisional dissociation (HCD)

fragmentation. Data were processed in MaxQuant using iBAQ quantitation and a FDR of 1%.

### Identifying enriched proteins

Any proteins without an iBAQ value were assigned a default value of 0 before the analysis. Possible contaminants and confounding proteins such as keratin and ribosomal proteins were filtered out from the analysis. The iBAQ values were then log transformed and centered around the mean of each sample. Enriched proteins were identified by calculating the $\log_2$ fold change of a test probe versus the average of all other probes tested in the same sample. The list of enriched proteins was then manually checked in the MGI database and proteins with a known hematopoietic phenotype was noted.

### RIP-qPCR

MEL cells were induced with DMSO and collected after 4 d. These cells were then cross-linked by adding 37% formaldehyde dropwise to confluent cell culture dishes for a final concentration of 1%. Then the cell cultures were incubated for 10 min at room temperature before being treated with 1 M glycine for a final concentration of 125 mM. Cell lysis and the subsequent RIP protocol was performed using the Magna RIP Kit (17-701; Millipore) according to manufacturer instructions with a rabbit polyclonal antibody $\alpha$–DKC1 (ab93777; Abcam) and the rabbit polyclonal antibody against $\beta$-ACTIN (ab8227; Abcam) as control. The resulting RNA was reverse transcribed using the iScript Reverse Transcription Supermix for RT-qPCR (1708840; Bio-Rad) and then pre-amplified with SsoAdvanced PreAmp Supermix (172-5160; Bio-Rad), following the manufacturer's instructions. Standard qPCR was performed for Adar (F: TGAGCATAGCAAGTGGA-GATACC; R: GCCGCCCTTTGAGAAACTCT), Aff1 (F: GAAGGAAAGACGCAACC AAGA; R: TAGCTCATCGCCTTTTGCAGT), Appl2 (F: CACCCTCACAGATTACACC AAC; R: GGAGAACCATAGTGTCTGCCAG), Dido1 (F: GAAGCACCCAAGGC-TATCAAA; R: GTCCACCTCCGTGTCTCCT) and GAPDH (F: CGTCCCGTAGA-CAAAATGGT; R: TTGATGGCAACAATCTCCAC). The primers for Appl2, Dido1, Aff1, and Adar were identified using PrimerBank ([68] and [69]), whereas GAPDH primers were the same as used in reference [70].

### Western blot

Protein lysates were prepared from cross-linked 4-d post-DMSO induction cells using RIPA buffer and run on 4–12% Bis–Tris Gels (NP0322; Invitrogen) in MES at 100 V for 2 h. Then the gel was transferred to polyvinylidene fluoride (PVDF) membranes at 200 mA at 4°C for 2 h. After 2 h of blocking in 5% BSA at room temperature, the membrane was incubated in primary $\alpha$–DKC1 (6 $\mu$l antibody in/ 10 ml of 5% BSA; ab93777; Abcam) or $\alpha$-ACTIN (9 $\mu$l antibody/10 ml of 5% BSA; ab8227; Abcam) antibody overnight in 4°C. Membranes were then washed three times in tris-buffered saline (TBST) for 10 min each then incubated for an hour in goat anti-Rabbit secondary (1:3,000 in 3% BSA; ab6721; Abcam). After three 10-min washes with TBST, the membrane was then removed from the liquid and ECL Prime Western Blotting Detection Reagent (GE Healthcare) was applied to the membrane for 1 min. Images were then taken in 10 s increments until saturation.

## Data Availability

Raw FASTQ files for the PIP-seq libraries as well as processed BED files containing PPSs and calculated RNA secondary structure scores are available on GEO under the accession number GSE142242 (https://www.ncbi.nlm.nih.gov/geo/query/acc.cgi?acc=GSE142242). The FASTQ files for accompanying mRNA-seq are available on GEO under the accession number GSE148421 (https://www.ncbi.nlm.nih.gov/geo/query/acc.cgi?acc=GSE148421).

## Supplementary Information

## Acknolwedgements

We thank the members of both the Gregory and Liebhaber labs, past and present, for their thoughtful feedback and comments throughout the project. This work was supported by the NIH through grant R01HL065449-18.

### Author Contributions

M Shan: data curation, software, formal analysis, validation, investigation, methodology, and writing—original draft, review, and editing.
X Ji: data curation, formal analysis, validation, investigation, methodology, and writing—original draft, review, and editing.
K Janssen: resources, formal analysis, investigation, and methodology.
IM Silverman: resources, data curation, formal analysis, investigation, and methodology.
J Humenik: resources, data curation, formal analysis, investigation, and methodology.
BA Garcia: resources, methodology, and project administration.
SA Liebhaber: conceptualization, data curation, formal analysis, supervision, funding acquisition, investigation, project administration, and writing—original draft, review, and editing.
BD Gregory: conceptualization, resources, data curation, software, formal analysis, supervision, funding acquisition, validation, investigation, visualization, methodology, project administration, and writing—original draft, review, and editing.

### Conflict of Interest Statement

The authors declare that they have no conflict of interest.

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
