## [Reviewer comments · Life Science Alliance]

Life Science Alliance

Dynamic Changes in RNA-Protein Interactions and RNA Secondary Structure in Mammalian Erythropoiesis

Mengge Shan, Xinjun Ji, Kevin Janssen, Ian Silverman, Jesse Humenik, Benjamin Garcia, Stephen Liebhaber, and Brian Gregory

DOI: <https://doi.org/10.26508/lsa.202100659>

Corresponding author(s): *Brian Gregory, University of Pennsylvania*

Review Timeline:

Submission Date:	2020-01-24
Editorial Decision:	2020-02-26
Revision Received:	2020-12-16
Editorial Decision:	2021-02-07
Revision Received:	2021-04-02
Editorial Decision:	2021-05-21
Revision Received:	2021-07-06
Accepted:	2021-07-07

Transaction Report:

February 26, 2020

Re: Life Science Alliance manuscript #LSA-2020-00659-T

Brian D Gregory
University of Pennsylvania

Dear Dr. Gregory,

Thank you for submitting your manuscript entitled "Global Analysis of the RNA-Protein Interaction and RNA Secondary Structure Landscapes Identifies Dynamic Changes During Mammalian Erythropoiesis" to Life Science Alliance. The manuscript has now been seen by expert reviewers, whose reports are appended below.

As you will see, the three reviewers raise similar concerns and think that the approach taken may lead to artifacts. They further think that your conclusions are currently not supported by the data provided and that alternative explanations exist. Finally, they point to lacking controls.

Given these concerns and the work that would need to go into revising the manuscript to address the major issues, we are afraid we concluded that we cannot proceed further with the paper. We are thus returning your manuscript to you with the message that we cannot publish it here.

We are sorry our decision is not more positive, but hope that you find the reviews constructive. Of course, this decision does not imply any lack of interest in your work and we look forward to future submissions from your lab.

Thank you for your interest in Life Science Alliance.

Sincerely,

Reviewer #1 (Comments to the Authors (Required)):

Shan et al present a manuscript describing the dynamics of the RNA-binding proteins occupancy and RNA structure remodeling in the course of mammalian erythropoiesis. Authors build on a method that was developed and used initially with plant extracts and use it in a mammalian cell line.

Most of the claims are justified by the presented analysis, authors seemed to be knowledgeable with in silico and statistical analysis as far as I can judge, but did a very little homework while writing and compiling figures for this manuscript. Albeit there is a lot of work to be still done, should the authors invest the time in (mostly in silico) experimentation and very careful formatting of the manuscript to resolve the queries below, I would support the publication of this manuscript in LSA.

Major points

1. Authors describe how MEL cells can be used as a model for differentiation into the erythrocytes and extract the samples for PIP-seq from different stages of the differentiation process. Recognizing that the focus of authors lies in the large data side, they should nevertheless show quality controls for the cellular aspect of the study, specifically in the course of differentiation (e.g. cytospin images with HE or benzidine staining, measurement of haemoglobin levels in the differentiated cells, cell size parameters etc.).
2. Authors excluded the CDS from the analysis due to the ribosomal footprints on the mRNAs. Yet, pre-initiation complexes including 40S subunits can leave footprints masking as PPS on the 5'UTR as well (2018 paper from Thomas Preiss lab). Can authors show an analysis that would exclude significant contribution of such scenario?
3. Some miRNAs can have high abundance; especially the miR-451 is massively expressed in the terminal erythropoiesis. Do authors control for dsRNA PIP sites masked as miRNAs target sites?
4. Page 8 lane 3 refers to a SNP analysis in Figure S1C. There is a labelling mistake or the respective figure was not included in submitted manuscript, as the S1C is showing overlap of PIPs with CLIP for PABC1.
5. On page 10-11 authors discuss increase in PPS and RNA structures on mRNAs in the course of erythroid differentiation. Yet, authors do not discuss here - or in any other part of the paper - the contribution of single mRNA, Hbb encoding for haemoglobin, to the signals in their seq datasets. In theory, pretty much all the translation in reticulocytes is devoted to this mRNA and authors should showcase this example in their PPS and RNA structure profiles, and evaluate the extent of this mRNA contribution to the aggregated results in the plots in Figures 2 and 3.
6. Authors stress out how protein occupancy and RNA structure are anti-correlated in their analysis. Yet, in Figure 5 authors show examples of significantly enriched motif (I assume by analysis of PPS sites) and a high structural score. Though they devote one sentence to this, it will be important to analyse and interpret this more thoroughly.
7. In the analysis on pages 18-19 authors selected a GO set with erythropoiesis, identified motifs in these mRNAs, then overlapped with PPS and then analysed which genes have such PPS and score with GO in erythropoiesis-related topics. This seems to me as "self-fulfilling prophecy" analysis. Authors should elaborate on what exactly was the aim and include a figure panel describing the analysis steps clearly.
8. The mass spectrometry data set seems to be of a questionable quality. No column in Supp Table 1 is devoted to the beads only control - therefore I assume that proteins coming up in this control were subtracted. In that case, the top 70 most enriched proteins for any motif of any developmental day contain essentially histones, tubulins, ATPase subunits or glycolytic enzymes, strongly suggesting that biochemical approach was not appropriate. Authors should compare their datasets to the "crapome" dataset that is publicly available and carefully re-analyse the hits.
9. Motif 3 binds RBM38, but this protein is not in the MS dataset. It may be that the peptides do not fly well, albeit this is unlikely as RBM38 recurrently appears in RNA-binding proteomics datasets. In that case, how do the authors contemplate the motif 3 and RBM38 related results?

Minor points

1. First Results section titled "MEL cells as a model for red blood cell differentiation" does not provide any results in the form of figures or data. Unless controls from Major point 1 are included, authors should split and integrate this part into Introduction and Methods section.
2. Does the overlap of PIP sites and PABC1 CLIP site overlap primarily in 3'UTR, as expected?

3. Some Figures have unreadable axis or other labeling. It should be re-done or, in case needed, XLS files should be provided.
4. Page 8 lane 3, provide what "SNP" abbreviation stands for.
5. Figures 2 and 3 were swapped and Figure 2 lacks panel B annotation. This made it very hard to read and review.
6. Figure 1 panel D, labelling of X axis is cut at Day 0.
7. Results in the section titled "Potential RBP-RNA interacting sites reveal a conservation of structure" don't show the conservation of RNA structure; the title is therefore misleading. In addition, there is no appreciable difference in the 0,2,4 days plots in Figure 1D, therefore I recommend to show an overlap of these.
8. Page 8 bottom, authors repeat 3x times the argument of anti-correlation between PPS and RNA structure, when it only applies to translation start sites and stop codons. This section should be re-written accordingly.
9. Page 11, last sentence, dot missing.
10. When plotting Venn diagrams, circles should be scaled to the quantitative data they represent.
11. Figure 5 legend, "(C)" annotation missing.
12. What is the sequence of the motifs (motif 1, 2, 3, 4) discussed in the second paragraph on the page 19? The ones in Figure 5? Then they should be labeled as such and their sequences should be included in the Supplementary Figure 3B-E.
13. Lysis buffer composition for motif-beads missing.

Reviewer #2 (Comments to the Authors (Required)):

In this manuscript, Shan and colleagues applied PIP-seq technique and analytic pipeline to monitor change in RNA-binding protein (RBP) interactions and RNA secondary structures during the last stages of erythropoiesis. The authors use murine erythroleukemia (MEL) cells, which are arrested at the proerythroblast stage of development, for which terminal erythroid differentiation can be initiated by dimethyl sulfoxide treatment. The authors performed PIP-seq experiment pre DMSO treatment and at 2- and 4-day post-treatment. PIP-seq results provide information on RBP interaction sites and RNA secondary structure in cell lysate following RBP crosslinking in vivo. The authors extracted two metrics from their PIP-seq results: a protein protected sites (PPS) and a structure scores that provide information on RBP binding and RNA secondary structure, respectively. Next, they analyzed the distribution of PPS throughout the transcriptome and found an enrichment in UTRs and CDSs compared to introns. Moreover, the PPS sequences tend to be more conserved than their flanking sequences. A meta-analysis of PPS and Structure score surrounding the start and stop codon of protein coding transcripts reveals an anti-correlation between RBP binding and RNA structure and dynamics of RBP binding and RNA structure at those sites during erythropoiesis. Then, the authors identified motifs enriched in PPS regions found in UTRs and speculate on the potential function of RBPs known to bind these motifs during mammalian erythropoiesis. Next, they analyze the relationship between four of those motifs and changes in RNA structure in those regions. Finally, for the same set of four motifs, the authors attempted to identify the RBPs specific to those sequences. Using these motifs as baits, the authors pulled down proteins from cell lysates and analyze them mass spectrometry. This yielded a list of 569 proteins binding at least one motif and the authors end up speculating at length about the potential roles of few of these proteins during terminal erythroid differentiation.

In brief, the authors used the PIP-seq technique to analyze changes in RBP binding and RNA structure during terminal erythroid in a high throughput manner. They identified specific dynamics of

RBP binding and RNA structure at the start and stop codon of protein coding transcripts. Finally, they tried to identify new regulators of gene expression during erythroid differentiation using their data. While a map of the dynamic landscape of RBP binding and RNA structure during terminal erythroid differentiation is of high interest and importance, the poor writing of the manuscript and very shallow, yet all over the place, analysis of their dataset shed serious doubts on the usefulness of their resource. Moreover, their conclusions are often poorly supported by their results or there is not enough information provided to attest the strength of their conclusions.

There is a list of major issues:

1. The poor writing of the manuscript makes it extremely hard to assess the quality of the data and the authors' conclusions. For instance, figures 2 and 3 seems to have been switched somehow. Certain paragraphs are just incomprehensible (see major point 23). There is a major lack of information when presenting the results. I will try to address several of these points along with my scientific arguments in my review.

2. The authors mentioned that upon DMSO treatment, MEL cells start differentiation at a rate of approximately 30-100%. It is therefore primordial to analyze and report what proportion of cells undergo differentiation following their treatment. This control is essential to assess the quality and accuracy of their datasets.

3. The authors never justified the choice of their time points. It is unclear why they chose 2- and 4-days post-treatment (in addition to the pre-treatment time point). It is important to know and mention what developmental stages these time points correspond to and justify how they are relevant in the process of terminal erythroid differentiation. For example, is there a drastic change in mRNA level or translation or is there any specific cellular processes occurring at any of these time points

4. The authors clearly state - at two different places - that they avoid analyzing CDS regions by fear of false positive signals from the ribosomes. However, for most of their PPSs and structural analysis, the authors do analyze CDS regions and regions where ribosomes tend to occupy. Ribosome profiling experiments show enrichment of ribosomes at the CDS start and stop codons. Since most of the authors' analysis of the dynamics of RBP binding and RNA structure changes during erythroid differentiation was performed in Figure 2 and 3 and highlighting changes at the start and stop codons. One can wonder what part of those observations is merely due ribosome occupancy. The authors should address this major concern.

5. In their SNPs analysis falling in PPSs, the authors use bedtools shuffle to create their background dataset. To my understanding, this tool uses the whole chromosome to generate its background. Since the PPSs are calculated from signal limited to RNA transcript, this background should be limited to transcribe regions and not the whole chromosome where most of the sequence isn't transcribe at a significant level.

6. The same concern about background can be applied to their analysis of PPSs vs randomly generated mock overlapping with the PABPC1 CLIP experiments. I actually didn't find the information about how this background. That is said, I did find any file named Supplemental Experimental Procedures. Either this file wasn't available to me through the reviewing system or the authors forgot to provide this document. Anyhow, for this analysis it is crucial that the randomly generated mock comes from sequences that are in transcripts that are express at a similar level than those with PPS, ideally within the same transcript, but at different location not overlapping with any PPSs.

7. On page 8, the authors mentioned: "These results lead us to propose that PPSs, as potential interaction sites between RBPs and their target RNA, are functionally conserved throughout evolution and are also less prone to the effects of genetic drift in mice.". The fact that their PPSs are more conserved than their flanking sequences and that there are less SNPs within them (let see the new numbers with proper background) doesn't mean that they are functionally conserved. None of these analyses test for functionality. The authors should rephrase this statement.
8. It is unclear to me that the anti-correlation between PPS and RNA structure shouldn't be expected merely because of the PIP-seq protocol. Can the authors refer to any control (from previous literature or in this study) testing if the digestion of a protein with proteinase K leaves the RNA more open to be cleaved by ssRNase in the subsequent step. If that is the case, we would expect to have less structured RNA regions where there are more proteins bound. That would be the result of an artifact in the protocol rather than a biological observation.
9. The authors should mention how many genes are in there metaplot in Figure 2 and 3.
10. I didn't find the information about if the set of genes used to draw the metaplot in Figure 2 where the same for every time point. To do such a comparison, the authors need to analyze the same set of genes for each time point. That is particularly important considering the important sequence biases that this type of experiments contain.
11. The authors should explain their rationale of why they look at +/- 400 nt for Figure 2 and +/- 100 nt for Figure 3.
12. For their metaplot, in the method section under "Structure score profile analysis of mRNAs", the authors mention that they consider only mRNAs with ≥ 45 nt 5'UTRs and ≥ 140 nt 3'UTRs. It is unclear how the authors can build metaplot looking at region that exceed the length of certain mRNAs' UTRs (i.e. +/- 100 or 400 nt).
13. In the metaplots in Figure 2 and 3, what does the shadow of the lines mean?
14. The p-values in the tables in Figure 2 don't match those reported in the text. It is unclear if it is because they are calculating something else or if this is a mistake. There is a clear typo in the lower table for Start-100 Day 2 vs Day 0, there is a v after the p-value.
15. The authors should provide the sequencing depth of each PIP-seq samples. In these types of experiments, something certain features arise because of large different in coverage/sequencing depth.
16. There are clear differences in RNA structure and PPSs across time points that the authors don't mention and highlight. It would be nice to have an explanation or hypothesis about why PSSs goes down in the 5'UTR at Day 4 vs Day 0 and 2, or why PSSs in the 3'UTR is the highest at Day 4, lowest at Day 2 and in the middle at Day 0.
17. On page 11 the authors mention: "Thus, we observed a large-scale shift towards increasing overall secondary structure in protein-coding transcripts during the process of mammalian erythropoiesis." This overall continuous increase is only observed in CDS regions (regions that the authors said they should avoid analyzing because of the risk of false positive signal caused by ribosome occupancy). As mentioned above, this continuous trend isn't observed in 5' (Day 0 is

higher than 2) and 3'UTRs (Day 2 and 4 are pretty much the same). The authors should revisit this statement and discuss to full spectrum of their results.

18. The result section on page 13 titled "Identifying novel RBP-bound RNA motifs" is very long and disconnected overall. The authors should split it in logical unit.

19. For their motif enrichment in PPSs analysis, it is unclear if the authors report all the significant motifs in Figure 4 or if there were more than that (if yes they should be provided as supplementary data).

20. It is unclear what was the rationale to bin the different motifs in different panel on Figure 4 (A, B and C). The authors never discuss panel B and C, but discuss in large, maybe a bit too much, about the hypothetical roles of RBP recognizing motifs in panel A.

21. In the Figure legend of Figure 4, the authors mention that they limited their motif analysis to 400-500 nt downstream of the stop codon. Why? Why not analyze the full length 3'UTR of mRNAs?

22. On page 16 the authors state: "Our HOMER motif enrichment analysis of the PPSs detected several statistically enriched motifs in both biological replicates.". When the authors talk about several motifs, how many are they talking about? Is there more than the ones they report in Figure 4 and 5?

23. The whole paragraph starting with "Based upon differences in RNA secondary structure..." on page 16 is extremely misleading and hard to understand. It is unclear if the authors chose those 2 motifs (Fig. 5A-B) because there is a structural change or because they were found in both replicates or because they didn't have known interactors and were in interesting transcripts (the transcripts' name are not mentioned, if it is interesting they should probably be) or all of the previous. This whole paragraph should be revisited.

24. Figure legend 5, what does the shadow represents in the line plots of the structure score? How many sites are represented in each metaplot?

25. In Figure 5C, the authors chose to analyze further one specific motif that is enriched in PPSs that overlap RBM38 binding sites. It would be important to know how many PPSs overlapped with RBM38 binding sites and what are all the significant motifs they got from their HOMER analysis. It is mentioned that the motif in Fig. 5C is one of the identified sequences, but no information is provided about the others.

26. The authors mention on page 18: "Given the existence of dsRBPs, we included this motif on the basis of trying to identify RBPs with the potential of interacting with areas of increased structure.". While this is certainly interesting and often overlooked, the authors could find a better scientific narrative than "Given the existence of dsRBPs". Is there any rationale for the role off dsRBPs in regulation erythropoiesis or in relation with RBM38?

27. Still on page 18, the authors say that it is interesting that the RNA structure at motif 3 (Fig. 5C) increases in structure while they observe a global decrease in structure at PPSs (referred as anti-correlating). It would be interesting to perform a deeper analysis of the relationship between RNA structure and PPSs and that for different mRNA location. For example, out of all the PPSs (that pass a certain cutoff) in 3'UTRs how many increase or decrease in RNA structure? Same can be

done for 5'UTR and potentially for CDS. Different behavior could be interesting. We would expect more of the decrease in structure and it would reinforce the authors' statement about the anticorrelation RNA structure vs PPSs.

28. Same at previous point 25 for Fig. 5C, Fig. 5D's motif is said to be "One of the motifs of interest identified". Please provide more information about all the results of this motifs search focusing on erythropoiesis related transcripts.

29. I have serious concerns about the pull-down experiments to find interactors for the 4 motifs that they have selected. No controls were performed to validate the veracity of those data and the number of RBPs that they identify seems very large and unspecific. The authors should at the very least confirm the binding affinity of few of their candidate one or more of their motifs. This could be done using simple gel shift experiments.

30. The authors state on page 20: "Thus, these RNA sequence motifs interact with specific sets of proteins across mammalian red blood cell development.". I would caution the authors to use the word "specific" here. With 569 RBPs suggested to bind their sequences, it doesn't look very specific.

31. The 4 pages of explanation about why few of those 569 RBPs could be interesting in regard to post-transcriptional regulation during erythropoiesis feel very long and should be reduced in length and be more concise.

32. On page 24-25 the authors state: "It has been established that RBPs interact with RNAs most often through the 5' and 3' untranslated regions (i.e. 5' UTR and 3' UTR) to perform a variety of functions ranging from stabilizing the RNA to alternative splicing.". This is scientifically flawed. The authors should cite a reference to support their statement that RBPs interact with RNAs most often through the 5' and 3' UTRs, since it isn't a given. Clearly these regions are hubs for RBP binding, but here they add a quantification aspect to it that wouldn't qualify as well established. Especially since the authors refer as RBPs regulating alternative splicing that bind very often in introns and exons corresponding to CDSs.

Other minor points:

a) Repetition of the same sentence within the same paragraph in the introduction. Sentence #1: "Importantly, this process includes a significant dependence on post-transcriptional regulatory processes, especially during its terminal steps (An et al., 2014)." and Sentence #2: "Terminal steps in the process rely heavily on post-transcriptional regulatory processes (An et al., 2014)." These sentences are three sentences apart.

b) The authors wrongly cite Rouskin et al. 2014 in the introduction: "These techniques generally utilize structure-specific RNases (ssRNases and dsRNases) to provide site-specific evidence for a region being in single- or double-stranded configurations (Rouskin et al., 2014; Zheng et al., 2010)." Rouskin et al. used a chemical probing reagent, dimethyl sulfate, not a structure-specific RNases.

c) Figure 2, I would either label the lower panel B or just remove the label for panel A. No need for a label if there is only one panel.

d) The authors should split the result section "RNA secondary structure and RBP binding are anti-correlated". This is relevant for the first paragraph, but the second paragraph, starting with "We next interrogated whether we could detect ...", starts describing changes in PPSs and RNA structure across time points and should have its own new header to help the reader to follow.

e) On page 10 the authors mention "We found there was an initial significant (p -value $< 2.2 \times 10^{-16}$; Wilcoxon test) decrease in RBP binding around the stop codon of detectable protein-coding transcripts." The authors should clearly state with they are comparing and not only say an initial significant, if it is Day 0 vs Day 2 please state it clearly.

f) On page 15, there is still a track change left at "can bind to the 3' poly(A) tail".

g) Type on page 30 in result section "RBP bound sequence motif...", "Homer (...) as used" should be "was used".

h) Figure legend 5, the authors omitted "(C)".

i) Figure 5, please increase the font of the GO analysis.

Reviewer #3 (Comments to the Authors (Required)):

This manuscript reports a global overview of the dynamic changes in both RNA secondary structure and RBP occupancy for the erythroid transcriptome during terminal erythropoiesis. The authors used their PIP-seq (protein interaction profile sequencing) methods together with single strand- and double strand-specific nucleases to characterize regions protected by protein binding and/or by secondary structure. They also identify conserved motifs in protected regions that implicate a number of RBPs as candidate key post-transcriptional regulators. Interestingly, they also report that secondary structure and RBP occupancy are consistently anti-correlated throughout mRNAs during erythropoiesis.

Comments:

1. First, a general question. Did the authors consider whether any differences in profiles might be due to the rather extreme changes in transcript abundance that accompany terminal erythropoiesis? That is, if globin mRNAs predominate at the late time point, could differences in structure or RBP binding be due simply to a shift from a complex transcriptome to a globin-dominated transcriptome? Is it known what fraction of the transcriptome is globin mRNA at day 4 in MEL cells? This factor may not have been an issue in other cell contexts examined by these methods.

2. The distribution of PPSs among different RNA regions, as shown in Figure 1C, is subject to two large biases. First, as the authors mention earlier but do not seem to take into account in the reporting here, the CDS data includes ribosome binding sites as well as PPS. (This is also a problem for several of the later correlations.) Second, the intron sequence data should be normalized with respect to abundance of the sequences, that is, most introns are co-transcriptionally and presumably degraded, meaning that they will be present at a much lower molar concentration relative to the CDS and UTR regions. The normalization reported in the paper, i.e., adjusting for the number of bases annotated to each features, is inadequate in the case of introns to estimate the frequency of RBP binding.

3. The data in Figure 2 is interpreted to show anti-correlation of RNA secondary structure and RBP binding. It seems clear that this is true at the start and stop codons, but otherwise I think it's hard to validate in the coding regions without having a better handle on ribosome binding and its

contribution to the PPS signal.

4. In Figure 3, the text does not seem to accurately describe the figure pieces. I was confused, and wonder if the wrong version of the figure was uploaded? For example, on p. 10 the text says that "there was a highly significant (p -values $< 2 \times 10^{-16}$) decrease in the density of RBP binding around the start codon when comparing day 4 to both days 0 and 2 (Figure 3A)." But Fig 3A does not compare RBP density around the start codon at the different developmental times; it only shows data for day 0. Day 2 data is in Fig. 3B, while day 4 data is in Fig. 3C.

More importantly, when comparing the data across figure pieces, it's hard to be convinced that there is a widespread change in RBP binding density at the start and stop codons during terminal erythropoiesis. Regarding secondary structure, again I don't see the patterns described in the text. It is not obvious to me that secondary structure progressively increases in both regions around the start and stop codons. Or is this supposed to be shown in Figure 2?

5. Figure 5 shows new evidence for a possible dsRNA binding protein that binds to 3'UTR sequences overlapping or adjacent to RBM39 binding sites. This is very interesting preliminary data, but unfortunately the identity of the binding protein is as yet unknown.

6. On p. 19 the text refers to motifs 1-4. Are these motifs the same as the motifs shown in Figure 5? Please clarify, since these were not explicitly defined. Also, their relative abundance in introns should be corrected for the reduced occurrence of introns relative to CDS and UTR sequences due to splicing.

7. The last sections of the manuscript focused on the enriched motifs found in some of the 3'UTR regions, and mass spec analysis of candidate binding proteins. Overall this section describes a highly interesting approach that likely will reveal exciting new insights, but currently is still pretty preliminary. Figure 4 shows an impressive list of motifs that are highly enriched in the UTR sequences (although I missed something about how RNA structure influenced the motif selection, as implied in the figure legend).

Minor issues:

1. Fig. 5 legend is missing the label for part C.

Reviewer #1 (Comments to the Authors (Required)):

Shan et al present a manuscript describing the dynamics of the RNA-binding proteins occupancy and RNA structure remodeling in the course of mammalian erythropoiesis. Authors build on a method that was developed and used initially with plant extracts and use it in a mammalian cell line. Most of the claims are justified by the presented analysis, authors seemed to be knowledgeable with in silico and statistical analysis as far as I can judge, but did a very little homework while writing and compiling figures for this manuscript. Albeit there is a lot of work to be still done, should the authors invest the time in (mostly in silico) experimentation and very careful formatting of the manuscript to resolve the queries below, I would support the publication of this manuscript in LSA.

Major points

1. Authors describe how MEL cells can be used as a model for differentiation into the erythrocytes and extract the samples for PIP-seq from different stages of the differentiation process. Recognizing that the focus of authors lies in the large data side, they should nevertheless show quality controls for the cellular aspect of the study, specifically in the course of differentiation (e.g. cytopsin images with HE or benzidine staining, measurement of haemoglobin levels in the differentiated cells, cell size parameters etc.).

We have paired RNA-seq data in which we have measured the changes in RNA abundance across these three developmental time points. Using this data, we have found that the changes in RNA levels match well known changes during mammalian red blood cell development (e.g. increases in hemoglobin-encoding transcripts). This information has been added to a revised version of the manuscript in the Results section and as Supplemental Figure 1. It is also notable that the samples used in this current study are nearly identical to those used in our previously published study of PAPBC1 RNA binding that we published in *RNA* in 2016 (Kini, et al., *RNA* 2016 22: 61-74).

2. Authors excluded the CDS from the analysis due to the ribosomal footprints on the mRNAs. Yet, pre-initiation complexes including 40S subunits can leave footprints masking as PPS on the 5'UTR as well (2018 paper from Thomas Preiss lab). Can authors show an analysis that would exclude significant contribution of such scenario?

We have now excluded all PPSs in the 20 – 40 nucleotide range (comprising less than 1/6 of our total PPS data), which is the size range of the ribosomal footprint, and found that this changes none of our original findings. Therefore, we will note this in our revised manuscript, and use this slightly smaller set of PPSs given that this is a resource article and few of the PPSs seem to actually be the result of ribosomal footprints.

3. Some miRNAs can have high abundance; especially the miR-451 is massively expressed in the terminal erythropoiesis. Do authors control for dsRNA PIP sites masked as miRNAs target sites?

These sites have been excluded from our analyses in our revised manuscript.

4. Page 8 lane 3 refers to a SNP analysis in Figure S1C. There is a labelling mistake or the respective figure was not included in submitted manuscript, as the S1C is showing overlap of PIPs with CLIP for PABC1.

We have made this correction in the revised version of our manuscript.

5. On page 10-11 authors discuss increase in PPS and RNA structures on mRNAs in the course of erythroid differentiation. Yet, authors do not discuss here - or in any other part of the paper - the contribution of single mRNA, Hbb encoding for haemoglobin, to the signals in their seq datasets. In theory, pretty much all the translation in reticulocytes is devoted to this mRNA and authors should showcase this example in their PPS and RNA structure profiles, and evaluate the extent of this mRNA contribution to the aggregated results in the plots in Figures 2 and 3.

We have found that removing Hbb from the PPS density and RNA secondary structure analysis has no noticeable effect on the metagene profiles, which leads us to conclude that the influence of Hbb to the overall conclusions is minimal. These findings are shown in Supplemental Figure 7 and described in the revised manuscript.

6. Authors stress out how protein occupancy and RNA structure are anti-correlated in their analysis. Yet, in Figure 5 authors show examples of significantly enriched motif (I assume by analysis of PPS sites) and a high structural score. Though they devote one sentence to this, it will be important to analyse and interpret this more thoroughly.

We have expanded our discussion of these findings in the revised version of our manuscript. Specifically, we have added the following explanation to the revised manuscript.

“A further analysis of the PIP-seq data revealed that the motif occurs in a region where RNA secondary structure increased during mammalian red blood cell development, suggesting it as a potential binding sequence for an uncharacterized RBP that prefers to bind RNA in a paired conformation (double-stranded RBP (dsRBP)) (**Figure 6D**). Though the overall analysis of RNA secondary structure profiles and RBP binding densities suggests a general anti-correlation between these two features, there are RBPs that

preferentially bind to double-stranded RNAs and perform a variety of post-transcriptional regulatory actions [50]. Notably, our mRNA-seq data detected the significant increase in the levels of 11 mRNAs encoding dsRBPs when comparing day 4 RNA abundance to undifferentiated (day 0) MEL cells. Given the existence of dsRBPs in the mammalian genome and the upregulation of several of them throughout erythropoiesis, we included this motif on the basis of trying to identify RBPs with the potential for interacting with areas of increased structure.”

7. In the analysis on pages 18-19 authors selected a GO set with erythropoiesis, identified motifs in these mRNAs, then overlapped with PPS and then analysed which genes have such PPS and score with GO in erythropoiesis-related topics. This seems to me as "self-fulfilling prophecy" analysis. Authors should elaborate on what exactly was the aim and include a figure panel describing the analysis steps clearly.

The aim of this analysis was to identify enriched motifs in erythropoiesis related transcripts (the reverse GO analysis was done using a different algorithm to ensure the annotation wasn't outdated). We have added more in-depth descriptions denoting this analysis as suggested by the Reviewer in our revised manuscript, including a detailed Methods section to describe this approach.

8. The mass spectrometry data set seems to be of a questionable quality. No column in Supp Table 1 is devoted to the beads only control - therefore I assume that proteins coming up in this control were subtracted. In that case, the top 70 most enriched proteins for any motif of any developmental day contain essentially histones, tubulins, ATPase subunits or glycolytic enzymes, strongly suggesting that biochemical approach was not appropriate. Authors should compare their datasets to the "crapome" dataset that is publicly available and carefully re-analyse the hits.

Instead of a bead-only control, the study uses the other probe(s) as controls and requires that the protein of interest identified as potentially interacting with any motif show a >10-fold increase in binding to that probe vs. control (other) probes. This has further refined our protein identification. We have also validated our mass spec results through RIP-qPCR analyses demonstrating that identified RBP interact with the target mRNAs containing their motifs as predicted.

9. Motif 3 binds RBM38, but this protein is not in the MS dataset. It may be that the peptides do not fly well, albeit this is unlikely as RBM38 recurrently appears in RNA-binding proteomics datasets. In that case, how do the authors contemplate the motif 3 and RBM38 related results?

Motif 3 is not guaranteed to bind RBM38 as it is a motif found to neighbor the RBM38 binding motif. Thus, our hypothesis is that it's a binding motif for a protein that may interact/co-bind with RBM38, and thus the baits do not contain the RBM38 motif. Thus, it is not expected to identify RBM38 in our mass spec analysis as we show in our datasets. We have further clarified this point in our revised manuscript.

Minor points

1. First Results section titled "MEL cells as a model for red blood cell differentiation" does not provide any results in the form of figures or data. Unless controls from Major point 1 are included, authors should split and integrate this part into Introduction and Methods section.

We have made the suggested edits as described in our revised version of the manuscript.

2. Does the overlap of PIP sites and PABPC1 CLIP site overlap primarily in 3'UTR, as expected?

We have addressed this point in the text and in Figure S1E in the revised version of our manuscript.

3. Some Figures have unreadable axis or other labeling. It should be re-done or, in case needed, XLS files should be provided.

We have reformatted all figures to be more readable as suggested by the Reviewer.

4. Page 8 lane 3, provide what "SNP" abbreviation stands for.

This information has been added to the revised manuscript.

5. Figures 2 and 3 were swapped and Figure 2 lacks panel B annotation. This made it very hard to read and review.

These errors have been corrected in the revised manuscript.

6. Figure 1 panel D, labelling of X axis is cut at Day 0.

This error has been corrected in the revised manuscript.

7. Results in the section titled "Potential RBP-RNA interacting sites reveal a conservation of structure" don't show the conservation of RNA structure; the title is therefore misleading. In addition, there is no appreciable difference in the 0,2,4 days plots in Figure 1D, therefore I recommend to show an overlap of these.

We have made the changes as suggested by the Reviewer in our revised manuscript.

8. Page 8 bottom, authors repeat 3x times the argument of anti-correlation between PPS and RNA structure, when it only applies to translation start sites and stop codons. This section should be re-written accordingly.

We have made the changes as suggested by the Reviewer in our revised manuscript.

9. Page 11, last sentence, dot missing.

This error has been corrected in the revised manuscript.

10. When plotting Venn diagrams, circles should be scaled to the quantitative data they represent.

We have changed these plots as suggested by the Reviewer in our revised manuscript.

11. Figure 5 legend, "(C)" annotation missing.

This error has been corrected in the revised manuscript.

12. What is the sequence of the motifs (motif 1, 2, 3, 4) discussed in the second paragraph on the page 19? The ones in Figure 5? Then they should be labeled as such and their sequences should be included in the Supplementary Figure 3B-E.

This error has been corrected in the revised manuscript.

13. Lysis buffer composition for motif-beads missing.

This error has been corrected in the revised manuscript.

Reviewer #2 (Comments to the Authors (Required)):

In this manuscript, Shan and colleagues applied PIP-seq technique and analytic pipeline to monitor change in RNA-binding protein (RBP) interactions and RNA secondary structures during the last stages of erythropoiesis. The authors use murine erythroleukemia (MEL) cells, which are arrested at the proerythroblast stage of development, for which terminal erythroid differentiation can be initiated by dimethyl sulfoxide treatment. The authors performed PIP-seq experiment pre DMSO treatment and at 2- and 4-day post-treatment. PIP-seq results provide information on RBP interaction sites and RNA secondary structure in cell lysate following RBP crosslinking

in vivo. The authors extracted two metrics from their PIP-seq results: a protein protected sites (PPS) and a structure scores that provide information on RBP binding and RNA secondary structure, respectively. Next, they analyzed the distribution of PPS throughout the transcriptome and found an enrichment in UTRs and CDSs compared to introns. Moreover, the PPS sequences tend to be more conserved than their flanking sequences. A meta-analysis of PPS and Structure score surrounding the start and stop codon of protein coding transcripts reveals an anti-correlation between RBP binding and RNA structure and dynamics of RBP binding and RNA structure at those sites during erythropoiesis. Then, the authors identified motifs enriched in PPS regions found in UTRs and speculate on the potential function of RBPs known to bind these motifs during mammalian erythropoiesis. Next, they analyze the relationship between four of those motifs and changes in RNA structure in those regions. Finally, for the same set of four motifs, the authors attempted to identify the RBPs specific to those sequences. Using these motifs as baits, the authors pulled down proteins from cell lysates and analyze them mass spectrometry. This yielded a list of 569 proteins binding at least one motif and the authors end up speculating at length about the potential roles of few of these proteins during terminal erythroid differentiation.

In brief, the authors used the PIP-seq technique to analyze changes in RBP binding and RNA structure during terminal erythroid in a high throughput manner. They identified specific dynamics of RBP binding and RNA structure at the start and stop codon of protein coding transcripts. Finally, they tried to identify new regulators of gene expression during erythroid differentiation using their data. While a map of the dynamic landscape of RBP binding and RNA structure during terminal erythroid differentiation is of high interest and importance, the poor writing of the manuscript and very shallow, yet all over the place, analysis of their dataset shed serious doubts on the usefulness of their resource. Moreover, their conclusions are often poorly supported by their results or there is not enough information provided to attest the strength of their conclusions.

There is a list of major issues:

1. The poor writing of the manuscript makes it extremely hard to assess the quality of the data and the authors' conclusions. For instance, figures 2 and 3 seems to have been switched somehow. Certain paragraphs are just incomprehensible (see major point 23). There is a major lack of information when presenting the results. I will try to address several of these points along with my scientific arguments in my review.

We apologize for the poor writing of our manuscript. This has been remedied and we hope that the Reviewer finds our revised manuscript more suitable in its current form.

2. The authors mentioned that upon DMSO treatment, MEL cells start differentiation at a rate of approximately 30-100%. It is therefore primordial to analyze and report what proportion of cells undergo differentiation following their treatment. This control is essential to assess the quality and accuracy of their datasets.

We have added this information to the revised manuscript based on an mRNA-seq analysis the vast majority of cells (near 100%) are all going through the differentiation process induced by DMSO. It is also notable that the samples used in this current study are nearly identical to those used in our previously published study of PAPBC1 RNA binding that we published in *RNA* in 2016 (Kini, et al., *RNA* 2016 22: 61-74).

3. The authors never justified the choice of their time points. It is unclear why they chose 2- and 4-days post-treatment (in addition to the pre-treatment time point). It is important to know and mention what developmental stages these time points correspond to and justify how they are relevant in the process of terminal erythroid differentiation. For example, is there a drastic change in mRNA level or translation or is there any specific cellular processes occurring at any of these time points

We have added this information to the revised manuscript. Specifically, these time points are demonstrating large-scale and significant increases in *Hbb* levels and decreases in other known markers of erythropoiesis. These findings are added to the current manuscript. It is also notable that the samples used in this current study are nearly identical to those used in our previously published study of PAPBC1 RNA binding that we published in *RNA* in 2016 (Kini, et al., *RNA* 2016 22: 61-74).

4. The authors clearly state - at two different places - that they avoid analyzing CDS regions by fear of false positive signals from the ribosomes. However, for most of their PPSs and structural analysis, the authors do analyze CDS regions and regions where ribosomes tend to occupy. Ribosome profiling experiments show enrichment of ribosomes at the CDS start and stop codons. Since most of the authors' analysis of the dynamics of RBP binding and RNA structure changes during erythroid differentiation was performed in Figure 2 and 3 and highlighting changes at the start and stop codons. One can wonder what part of those observations is merely due ribosome occupancy. The authors should address this major concern.

As noted above, we have now excluded all PPSs in the 20 – 40 nucleotide range (comprising less than 1/6 of our total PPS data), which is the size range of the ribosomal footprint, and found that this changes none of our original findings. Therefore, we will note this in our revised manuscript, but will present the larger set of PPSs given that this is a resource article and few of the PPSs seem to actually be the result of ribosomal footprints.

5. In their SNPs analysis falling in PPSs, the authors use bedtools shuffle to create their background dataset. To my understanding, this tool uses the whole chromosome to generate its background. Since the PPSs are calculated from signal limited to RNA

transcript, this background should be limited to transcribe regions and not the whole chromosome where most of the sequence isn't transcribe at a significant level.

We have redone our background analysis by creating a shuffled background set with the same number, same strand, same chromosome, within annotated gene regions, as specified by the Reviewers. As demonstrated in new Figure S1E, the results are similar to our previous results, demonstrating the truly conserved nature of our identified PPS sequences.

6. The same concern about background can be applied to their analysis of PPSs vs randomly generated mock overlapping with the PABPC1 CLIP experiments. I actually didn't find the information about how this background. That is said, I did find any file named Supplemental Experimental Procedures. Either this file wasn't available to me through the reviewing system or the authors forgot to provide this document. Anyhow, for this analysis it is crucial that the randomly generated mock comes from sequences that are in transcripts that are express at a similar level than those with PPS, ideally within the same transcript, but at different location not overlapping with any PPSs.

We have redone our background analysis by creating a shuffled background set with the same number, same strand, same chromosome, within annotated gene regions, as specified by the Reviewers. Even using this background set as requested by the Reviewer our results are still as we previously described.

7. On page 8, the authors mentioned: "These results lead us to propose that PPSs, as potential interaction sites between RBPs and their target RNA, are functionally conserved throughout evolution and are also less prone to the effects of genetic drift in mice.". The fact that their PPSs are more conserved than their flanking sequences and that there are less SNPs within them (let see the new numbers with proper background) doesn't mean that they are functionally conserved. None of these analyses test for functionality. The authors should rephrase this statement.

We have made this change as requested by the Reviewer in our revised manuscript.

8. It is unclear to me that the anti-correlation between PPS and RNA structure shouldn't be expected merely because of the PIP-seq protocol. Can the authors refer to any control (from previous literature or in this study) testing if the digestion of a protein with proteinase K leaves the RNA more open to be cleaved by ssRNase in the subsequent step. If that is the case, we would expect to have less structured RNA regions where there are more proteins bound. That would be the result of an artifact in the protocol rather than a biological observation.

We have also published on regions that demonstrate positive correlation between these two features. See Gosai et al., *Mol. Cell* 2015 57: 376-388, Foley et al., *Dev. Cell* 2017 41: 204-220, and Kramer et al., *Plant Direct* 4: e00239. Thus, these findings are not merely a byproduct of our methodology.

9. The authors should mention how many genes are in there metaplot in Figure 2 and 3.

This information has been added to our revised manuscript.

10. I didn't find the information about if the set of genes used to draw the metaplot in Figure 2 where the same for every time point. To do such a comparison, the authors need to analyze the same set of genes for each time point. That is particularly important considering the important sequence biases that this type of experiments contain.

This analysis was done on the same set of transcripts across time points. This information has been added to the revised manuscript as suggested by the Reviewer.

11. The authors should explain their rationale of why they look at +/- 400 nt for Figure 2 and +/- 100 nt for Figure 3.

We have added this information to the revised manuscript.

12. For their metaplot, in the method section under "Structure score profile analysis of mRNAs", the authors mention that they consider only mRNAs with ≥ 45 nt 5'UTRs and ≥ 140 nt 3'UTRs. It is unclear how the authors can build metaplot looking at region that exceed the length of certain mRNAs' UTRs (i.e. +/- 100 or 400 nt).

We have revised these plots to only focus on the same set of transcripts across time points whose UTRs are equal to or longer than the specified UTR length for that analysis.

13. In the metaplots in Figure 2 and 3, what does the shadow of the lines mean?

The shading denotes the standard error of the mean. This information has been added to the revised manuscript.

14. The p-values in the tables in Figure 2 don't match those reported in the text. It is unclear if it is because they are calculating something else or if this is a mistake. There is a clear typo in the lower table for Start-100 Day 2 vs Day 0, there is a v after the p-value.

We have corrected these typos in our revised manuscript.

15. The authors should provide the sequencing depth of each PIP-seq samples. In these types of experiments, something certain features arise because of large different in coverage/sequencing depth.

This information has been added to our revised manuscript as requested by the Reviewer.

16. There are clear differences in RNA structure and PPSs across time points that the authors don't mention and highlight. It would be nice to have an explanation or hypothesis about why PSSs goes down in the 5'UTR at Day 4 vs Day 0 and 2, or why PSSs in the 3'UTR is the highest at Day 4, lowest at Day 2 and in the middle at Day 0.

We have added this discussion to the revised manuscript as requested by the Reviewer. We hypothesize that this is because of a decrease in binding of translation regulators (which is reflected in the decrease in 5' UTR binding), or maybe due to an increase in regulating the stability of transcripts that are still present toward the end of the differentiation process as transcription shuts down.

17. On page 11 the authors mention: "Thus, we observed a large-scale shift towards increasing overall secondary structure in protein-coding transcripts during the process of mammalian erythropoiesis." This overall continuous increase is only observed in CDS regions (regions that the authors said they should avoid analyzing because of the risk of false positive signal caused by ribosome occupancy). As mentioned above, this continuous trend isn't observed in 5' (Day 0 is higher than 2) and 3'UTRs (Day 2 and 4 are pretty much the same). The authors should revisit this statement and discuss to full spectrum of their results.

We have added a more in depth discussion of these results to the revised manuscript as requested by the Reviewer. This discussion can be read below.

"Overall, the structured 5' UTRs observed at day 4 could serve as a block for the binding of RBPs that could regulate the translation or other functionalities of those particular transcripts [37]. RBPs have been shown to bind to the 3' UTR to control mRNA stability and also translation in erythropoiesis [28], and the increase in PPS density in the region at day 4 could be as a result of the cell stabilizing the transcripts that are still present in the later stages of development as transcription is decreased."

18. The result section on page 13 titled "Identifying novel RBP-bound RNA motifs" is very long and disconnected overall. The authors should split it in logical unit.

We have revised this section as described by the Reviewer in our revised manuscript.

19. For their motif enrichment in PPSs analysis, it is unclear if the authors report all the significant motifs in Figure 4 or if there were more than that (if yes they should be provided as supplementary data).

We have added these results to our Supplemental Figures as requested by the Reviewer and have discussed these findings in more depth in our revised manuscript.

20. It is unclear what was the rationale to bin the different motifs in different panel on Figure 4 (A, B and C). The authors never discuss panel B and C, but discuss in large, maybe a bit too much, about the hypothetical roles of RBP recognizing motifs in panel A.

We have more balanced our discussion of the motifs presented in the Main Figures as suggested by the Reviewer in our revised manuscript.

21. In the Figure legend of Figure 4, the authors mention that they limited their motif analysis to 400-500 nt downstream of the stop codon. Why? Why not analyze the full length 3'UTR of mRNAs?

We have added an in depth description of our rationale to our revised manuscript as requested by the Reviewer.

22. On page 16 the authors state: "Our HOMER motif enrichment analysis of the PPSs detected several statistically enriched motifs in both biological replicates.". When the authors talk about several motifs, how many are they talking about? Is there more than the ones they report in Figure 4 and 5?

We have added this information into our Supplemental Figures and added more in depth discussion of these findings in our revised manuscript.

23. The whole paragraph starting with "Based upon differences in RNA secondary structure..." on page 16 is extremely misleading and hard to understand. It is unclear if the authors chose those 2 motifs (Fig. 5A-B) because there is a structural change or because they were found in both replicates or because they didn't have known interactors and were in interesting transcripts (the transcripts' name are not mentioned, if it is interesting they should probably be) or all of the previous. This whole paragraph should be revisited.

We have revised this paragraph in our revised manuscript as suggested by the Reviewer.

24. Figure legend 5, what does the shadow represents in the line plots of the structure score? How many sites are represented in each metaplot?

We have added this information into the Legend for Figure 5.

25. In Figure 5C, the authors chose to analyze further one specific motif that is enriched in PPSs that overlap RBM38 binding sites. It would be important to know how many PPSs overlapped with RBM38 binding sites and what are all the significant motifs they got from their HOMER analysis. It is mentioned that the motif in Fig. 5C is one of the identified sequences, but no information is provided about the others.

We have added this information and a more in depth discussion of these results as requested by the Reviewer to our revised manuscript.

26. The authors mention on page 18: "Given the existence of dsRBPs, we included this motif on the basis of trying to identify RBPs with the potential of interacting with areas of increased structure.". While this is certainly interesting and often overlooked, the authors could find a better scientific narrative than "Given the existence of dsRBPs". Is there any rationale for the role of dsRBPs in regulation erythropoiesis or in relation with RBM38?

We have further our description of the rationale behind focusing on this motif in our revised manuscript. Specifically, we found that the transcripts encoding several dsRNA binding proteins are upregulated in our paired mRNA-seq datasets for these differentiating MEL samples. Additionally, this motif is found in the 3'UTR of TAL1/SCL, which is a transcriptional regulator of erythropoiesis. Thus, we focused on identifying the protein(s) that bound to this motif in the context of erythropoiesis. This discussion has been added to our revised manuscript as requested by the Reviewer.

27. Still on page 18, the authors say that it is interesting that the RNA structure at motif 3 (Fig. 5C) increases in structure while they observe a global decrease in structure at PPSs (referred as anti-correlating). It would be interesting to perform a deeper analysis of the relationship between RNA structure and PPSs and that for different mRNA location. For example, out of all the PPSs (that pass a certain cutoff) in 3'UTRs how many increase or decrease in RNA structure? Same can be done for 5'UTR and potentially for CDS. Different behavior could be interesting. We would expect more of the decrease in structure and it would reinforce the authors' statement about the anticorrelation RNA structure vs PPSs.

We have added this analysis to our revised manuscript as requested by the Reviewer. We see changes in RNA secondary structure that range from decreasing (< 0) to increasing (> 0) when we compare induced cells with uninduced cells in two different subsets of PPSs. We compared the changes for PPSs found in the UTRs in both replicates at any time point (Supplemental Figure 8A) or in the UTRs in both replicates in induced cells (Supplemental Figure 8B) and don't see a clear decrease in structure. Descriptions of these findings have also been added to the revised manuscript.

28. Same at previous point 25 for Fig. 5C, Fig. 5D's motif is said to be "One of the motifs of interest identified". Please provide more information about all the results of this motifs search focusing on erythropoiesis related transcripts.

We have added this information to the revised manuscript as requested by the Reviewer.

29. I have serious concerns about the pull-down experiments to find interactors for the 4 motifs that they have selected. No controls were performed to validate the veracity of those data and the number of RBPs that they identify seems very large and unspecific. The authors should at the very least confirm the binding affinity of few of their candidate one or more of their motifs. This could be done using simple gel shift experiments.

To address this point, we have redone this analysis to focus on proteins enriched at least 10-fold in one probe as compared to the proteins found in the pull-downs done with all other probes. This has further refined our protein identification. We have also validated our mass spec results through RIP-qPCR analyses demonstrating that identified DCK1 interacts with the target mRNAs containing its motifs as predicted. The RIP-qPCR we feel is a more robust validation of our findings than gel shift assays.

30. The authors state on page 20: "Thus, these RNA sequence motifs interact with specific sets of proteins across mammalian red blood cell development." I would caution the authors to use the word "specific" here. With 569 RBPs suggested to bind their sequences, it doesn't look very specific.

We have revised our manuscript based on our more conservative analysis of 10-fold enrichment, and this new discussion has been added to our revised manuscript, thereby addressing this point by the Reviewer.

31. The 4 pages of explanation about why few of those 569 RBPs could be interesting in regard to post-transcriptional regulation during erythropoiesis feel very long and should be reduced in length and be more concise.

We have revised this discussion in our revised manuscript.

32. On page 24-25 the authors state: "It has been established that RBPs interact with RNAs most often through the 5' and 3' untranslated regions (i.e. 5' UTR and 3' UTR) to perform a variety of functions ranging from stabilizing the RNA to alternative splicing." This is scientifically flawed. The authors should cite a reference to support their statement that RBPs interact with RNAs most often through the 5' and 3' UTRs, since it isn't a given. Clearly these regions are hubs for RBP binding, but here they add a quantification aspect to it that wouldn't qualify as well established. Especially since the authors refer as RBPs regulating alternative splicing that bind very often in introns and exons corresponding to CDSs.

We have revised our manuscript as suggested by the Reviewer.

Other minor points:

a) Repetition of the same sentence within the same paragraph in the introduction. Sentence #1: "Importantly, this process includes a significant dependence on post-transcriptional regulatory processes, especially during its terminal steps (An et al.,

2014)." and Sentence #2: "Terminal steps in the process rely heavily on post-transcriptional regulatory processes (An et al., 2014)." These sentences are three sentences apart.

We have fixed this redundancy in our revised manuscript.

b) The authors wrongly cite Rouskin et al. 2014 in the introduction: "These techniques generally utilize structure-specific RNases (ssRNases and dsRNases) to provide site-specific evidence for a region being in single- or double- stranded configurations (Rouskin et al., 2014; Zheng et al., 2010)." Rouskin et al. used a chemical probing reagent, dimethyl sulfate, not a structure-specific RNases.

We have fixed this oversight in our revised manuscript.

c) Figure 2, I would either label the lower panel B or just remove the label for panel A. No need for a label if there is only one panel.

We have fixed this Figure labeling in our revised manuscript.

d) The authors should split the result section "RNA secondary structure and RBP binding are anti-correlated". This is relevant for the first paragraph, but the second paragraph, starting with "We next interrogated whether we could detect ...", starts describing changes in PPSs and RNA structure across time points and should have its own new header to help the reader to follow.

We have fixed this problem as suggested by the Reviewer in our revised manuscript.

e) On page 10 the authors mention "We found there was an initial significant (p -value $< 2.2 \times 10^{-16}$; Wilcoxon test) decrease in RBP binding around the stop codon of detectable protein-coding transcripts." The authors should clearly state with they are comparing and not only say an initial significant, if it is Day 0 vs Day 2 please state it clearly.

We have added this information to our revised manuscript.

f) On page 15, there is still a track change left at "can bind to the 3' poly(A) tail".

We have fixed this problem in our revised manuscript.

g) Type on page 30 in result section "RBP bound sequence motif...", "Homer (...) as used" should be "was used".

We have fixed this problem in our revised manuscript.

h) Figure legend 5, the authors omitted "(C)".

We have fixed this problem in our revised manuscript.

i) Figure 5, please increase the font of the GO analysis.

We have fixed this problem in our revised manuscript.

Reviewer #3 (Comments to the Authors (Required)):

This manuscript reports a global overview of the dynamic changes in both RNA secondary structure and RBP occupancy for the erythroid transcriptome during terminal erythropoiesis. The authors used their PIP-seq (protein interaction profile sequencing) methods together with single strand- and double strand-specific nucleases to characterize regions protected by protein binding and/or by secondary structure. They also identify conserved motifs in protected regions that implicate a number of RBPs as candidate key post-transcriptional regulators. Interestingly, they also report that secondary structure and RBP occupancy are consistently anti-correlated throughout mRNAs during erythropoiesis.

Comments:

1. First, a general question. Did the authors consider whether any differences in profiles might be due to the rather extreme changes in transcript abundance that accompany terminal erythropoiesis? That is, if globin mRNAs predominate at the late time point, could differences in structure or RBP binding be due simply to a shift from a complex transcriptome to a globin-dominated transcriptome? Is it known what fraction of the transcriptome is globin mRNA at day 4 in MEL cells? This factor may not have been an issue in other cell contexts examined by these methods.

To address this point, we have added information about globin expression in our samples based on paired RNA-seq results, and this information has been added to the revised manuscript. Additionally, we have also found that average structure score does not correlate with expression. Please see included figures addressing these points. Finally, we have also analyzed structure score and RBP binding patterns in the absence of globin transcripts (Supplemental Figure 7) and found no significant differences in these metrics with or without the inclusion of these mRNAs in our analyses. Thus, we feel confident that there is no effect of expression on structure results across these developmental time points. These points and findings are included in our revised manuscript.

2. The distribution of PPSs among different RNA regions, as shown in Figure 1C, is subject to two large biases. First, as the authors mention earlier but do not seem to take into account in the reporting here, the CDS data includes ribosome binding sites as well as PPS. (This is also a problem for several of the later correlations.) Second, the intron sequence data should be normalized with respect to abundance of the sequences, that is, most introns are co-transcriptionally and presumably degraded, meaning that they

will be present at a much lower molar concentration relative to the CDS and UTR regions. The normalization reported in the paper, i.e., adjusting for the number of bases annotated to each features, is inadequate in the case of introns to estimate the frequency of RBP binding.

We have addressed this point for the CDS by removing all PPSs that are 20 – 40 nt in length (the size of a ribosomal footprint). It is worth noting the results obtained are the same as with the inclusion of these PPSs. Thus, we are quite confident that the results are robust and reproducible. In regards to the point about the introns, in essence we are normalizing to RNA abundance values by calling PPSs in footprinting samples vs. structure samples. Thus, in PIP-seq data the same intron region would have the same mRNA abundance (i.e. expression) in both the footprinting and the structure library, which means that if we call a PPS in an intron region, the call has already taken mRNA abundance into consideration (i.e. the region is a PPS because the pileup of reads in the footprinting library was statistically greater than the pileup of reads in the structure library). What we are demonstrating with these figures in our manuscript is that – relative to the number of bases that are annotated as UTR/CDS/intron, we see less introns in intronic regions. We have further clarified this point in our revised manuscript by saying that we're seeing less intronic PPSs (because we don't enrich for pre-mRNAs in our library prep) as a result of the PIP-seq technique, and thus make our findings more clear for the readers of this manuscript.

3. The data in Figure 2 is interpreted to show anti-correlation of RNA secondary structure and RBP binding. It seems clear that this is true at the start and stop codons, but otherwise I think it's hard to validate in the coding regions without having a better handle on ribosome binding and its contribution to the PPS signal.

We have addressed this point by removing all PPSs that are 20 – 40 nt in length (the size of a ribosomal footprint). It is worth noting the results obtained are the same as with the inclusion of these PPSs. Thus, we are quite confident that the results are robust and reproducible.

4. In Figure 3, the text does not seem to accurately describe the figure pieces. I was confused, and wonder if the wrong version of the figure was uploaded? For example, on p. 10 the text says that "there was a highly significant (p -values $<2 \times 10^{-16}$) decrease in the density of RBP binding around the start codon when comparing day 4 to both days 0 and 2 (Figure 3A)." But Fig 3A does not compare RBP density around the start codon at the different developmental times; it only shows data for day 0. Day 2 data is in Fig. 3B, while day 4 data is in Fig. 3C.

More importantly, when comparing the data across figure pieces, it's hard to be convinced that there is a widespread change in RBP binding density at the start and stop codons during terminal erythropoiesis. Regarding secondary structure, again I don't see the patterns described in the text. It is not obvious to me that secondary structure

progressively increases in both regions around the start and stop codons. Or is this supposed to be shown in Figure 2?

We have fixed this oversight in our revised manuscript.

5. Figure 5 shows new evidence for a possible dsRNA binding protein that binds to 3'UTR sequences overlapping or adjacent to RBM39 binding sites. This is very interesting preliminary data, but unfortunately the identity of the binding protein is as yet unknown.

We have added additional discussion and more thorough analysis of these findings and this information is included in our revised manuscript.

6. On p. 19 the text refers to motifs 1-4. Are these motifs the same as the motifs shown in Figure 5? Please clarify, since these were not explicitly defined. Also, their relative abundance in introns should be corrected for the reduced occurrence of introns relative to CDS and UTR sequences due to splicing.

We have addressed this point for the CDS by removing all PPSs that are 20 – 40 nt in length (the size of a ribosomal footprint). It is worth noting the results obtained are the same as with the inclusion of these PPSs. Thus, we are quite confident that the results are robust and reproducible. In regards to the point about the introns being in less molar concentration, we have revised the text to note that the underrepresentation of PPSs in the intronic regions could potentially be attributed to the fact that our library does not enrich for pre-mRNAs and we would not expect to see an enrichment in PPSs in the introns as introns are typically removed by the splicing machinery.

7. The last sections of the manuscript focused on the enriched motifs found in some of the 3'UTR regions, and mass spec analysis of candidate binding proteins. Overall this section describes a highly interesting approach that likely will reveal exciting new insights, but currently is still pretty preliminary. Figure 4 shows an impressive list of motifs that are highly enriched in the UTR sequences (although I missed something about how RNA structure influenced the motif selection, as implied in the figure legend).

To address this point, we have redone this analysis to focus on proteins enriched >10-fold in one probe as compared to the proteins found in the pulldowns done with all other probes. This has further refined our protein identification. We have also validated our mass spec results through RIP-qPCR analyses demonstrating that identified RBPs interact with the target mRNAs containing their motifs as predicted. The RIP-qPCR we feel is a more robust validation of our findings than gel shift assays.

Minor issues:

1. Fig. 5 legend is missing the label for part C.

We have fixed this labeling problem in our revised manuscript.

February 7, 2021

Re: Life Science Alliance manuscript #LSA-2020-00659-TR-A

Dr. Brian D Gregory
University of Pennsylvania
Biology
433 S. University Ave.
Philadelphia, PA 19104

Dear Dr. Gregory,

Thank you for submitting your revised manuscript entitled "Dynamic Changes in RNA-Protein Interactions and RNA Secondary Structure in Mammalian Erythropoiesis" to Life Science Alliance. The manuscript has been seen by the original reviewers whose comments are appended below.

We were unable to secure comments from Rev # 1, however both Rev #2 and Rev #3 did look at the revised manuscript. As you will note from their comments both reviewers agree that the manuscript has been significantly improved, but also agree that there are still some remaining points that need to be addressed before the paper can be published in Life Science Alliance.

Our general policy is that papers are considered through only one revision cycle; however, given that the reviewers continue to be overall positive about the work in terms of its suitability for Life Science Alliance, we are open to one additional short round of revision.

Please submit the final revision within one month, along with a letter that includes a point by point response to the remaining reviewer comments. I am happy to discuss the timeline further, if needed.

- A letter addressing the reviewers' comments point by point.
- An editable version of the final text (.DOC or .DOCX) is needed for copyediting (no PDFs).
- High-resolution figure, supplementary figure and video files uploaded as individual files: See our detailed guidelines for preparing your production-ready images, <https://www.life-science-alliance.org/authors>
- Summary blurb (enter in submission system): A short text summarizing in a single sentence the study (max. 200 characters including spaces). This text is used in conjunction with the titles of

papers, hence should be informative and complementary to the title and running title. It should describe the context and significance of the findings for a general readership; it should be written in the present tense and refer to the work in the third person. Author names should not be mentioned.

B. MANUSCRIPT ORGANIZATION AND FORMATTING:

Sincerely,

Shachi Bhatt, Ph.D.

Executive Editor

Life Science Alliance

<https://www.lsjournal.org/>

Interested in an editorial career? EMBO Solutions is hiring a Scientific Editor to join the international Life Science Alliance team. Find out more here -

https://www.embo.org/documents/jobs/Vacancy_Notice_Scientific_editor_LSA.pdf

Reviewer #2 (Comments to the Authors (Required)):

In their revised version, Shan and colleagues have improved the writing of the paper, stream lined their rationale at few places and added few pieces of information. These improvements help the flow of the manuscript and lead to a better understanding of the results for the first half of the manuscript. However, the second half of the manuscript still needs significant redaction and stream lining of ideas to help the reader understand their rationales and results. This is also needed to be able to asset the strength of their conclusions and quality of the datasets. Moreover, few conclusions are overstated and not supported by the results (see below). These are major points that should be addressed:

1. The authors heavily rely on GO term and Mammalian phenotype prediction analyses to validate the results of their different datasets (both RNA structure and RBP binding). However, their background controls are inadequate. The signal from these techniques is directly linked to the expression level of individual transcripts. You can't detect RNA structure or RBP binding on non-expressed mRNA and, vice versa, you will have a better signal for highly expressed mRNAs due to better coverage. Therefore, in all their GO term and mammalian phenotype analyses, they should use as a background a list of genes that are expressed in their cell lines. For example, they can use their mRNA-seq datasets, use a cutoff to determine what genes are expressed at each time point and use these as backgrounds. Here, to my understanding of the default background in DAVID and MouseMine, the authors use all the genes found in the genome as background. By doing so, you will for sure get GO terms or phenotypes linked to erythropoiesis and related terms since it is the equivalent of doing the same type of analysis with mRNA-seq data. The authors should redo all their GO term and Mammalian phenotype analysis using expressed genes as a background.

2. Still related to backgrounds, the modification of the authors' background choice for the PPSs

analysis is better than previously (limited to annotated regions of pre-mRNAs versus anywhere in the chromosome), but it is not accurate enough. The nucleotide composition changes significantly across different pre-mRNA regions. The nucleotide composition of introns is different than the one from exons and UTRs different from CDS. When analyzing their PPSs versus a random background, the authors should take background regions in the same pre-mRNA regions than the PPS. For example, the background of a PPS in the 3'UTR should be a sequence of equal length in a 3'UTR. If the number of PPSs is low, they could choose X number of sequences (e.g. 10) per PPS.

3. To address the impact of ribosomes in their PPS analyses, the authors have excluded fragments in the 20-40 nt range. It is true that the ribosome footprint of a translating ribosome is ~28 nt in condition where there is a complete digestion of the RNA. In their PIP-seq experiments, it is unclear to me if the RNA digestion is high enough to fully digest the RNA and lead to ribosome-mediated ~28-nt fragments. It seems more likely to me that they are working in conditions where there is only partial digestion of the RNA and, therefore, the fragment impacted by the presence of ribosomes could be longer than ~28-nt. In my opinion, the authors can't use the removal of 20-40 nt fragments to remove the potential impact of ribosomes, and because I expect digestion to be incomplete (see point #4) they should rather assume that what they observe could come from the signal of ribosomes.

4. My rationale of why the signal originates from partially digested RNA is that the miRNeasy RNA isolation kit keeps fragment of >18-nt. If we think about the RNA structure part of the PIP-seq protocol, I think that fully double-stranded RNA or single-stranded RNA of >18-nt long are very rare in the transcriptome. Therefore, most of the fragments (especially as they increase in length) comes from partially digested RNAs. A similar rationale could be made for RBP. Most RBPs are likely to have a footprint smaller than 18-nt. The ribosome is a particularly large machinery composed of several RNAs and proteins and has a footprint of ~28-nt. On top of that, if the authors remove fragments with length between 20-40 nt, that means that most fragments are likely above 40-nt or concentrated between ~18 and 20-nt. To help the reader assess the authors' results and analyses. The authors should provide a fragment length distribution for each of their samples (RNA structure and PPSs).

5. Conclusion on page 11 at the end of the second paragraph, the authors wrote "We found no strong correlation between PPS coverage and RNA abundance (Figure 2F), suggesting that in general the total level of RNA-RBP interactions detected on mRNAs is independent of RNA abundance in mammalian erythropoiesis." This conclusion is misleading. I do agree with the authors that it doesn't seem like there is a strong bias in term of percentage of coverage and RNA levels for transcripts with at least 1 PPSs. However, I am convinced that if you compare the RNA level (from mRNA-seq experiment) of transcripts with 1 or more PPSs versus those without any PPS, the ones with 1 or more PPSs are found among the most highly expressed transcripts. As mentioned in major point #1, in these types of experiment you have a better signal from highly expressed genes. Especially with a sequencing depth of 36-58 million reads per library. The authors should rephrase their conclusion in this regard.

6. Conclusion on page 13 at the end of the first paragraph, the authors wrote "Taken together, our results demonstrate that the transcripts encoding known RBPs are differentially regulated in erythropoiesis and that we can use our PPSs to look for potential regions of RBP-RNA interactions for these known RBPs." To use "the transcripts encoding known RBPs" suggests that all mRNAs encoding known RBPs are differentially regulated in erythropoiesis. This is not what the authors show. They should replace the "the" by "certain" or "a subset" or equivalent.

7. Conclusion on page 14 at the end of the first paragraph, the authors wrote "In total, these findings reveal large-scale changes in RNA secondary structure during a mammalian cell developmental process that likely underlie important post-transcriptional regulatory processes important to mammalian erythropoiesis." This is an overstatement of the results. The authors don't provide evidence that RNA structure changes are likely underlying important post-transcriptional regulatory processes, nor that they are important for erythropoiesis. This conclusion should be toned down to reflect the results.

8. Page 14 second paragraph, the authors wrote "In the 400 nt window after the start codon". It is unclear and a bit hard to follow that the authors mention 400 nt window in the main text, but +/- 500 nt window flanking the start and stop codon in the Figure 3 legend and finally +/- 100 nt window for their p-value. This should be harmonized.

9. Results from Figure 5A and Supplemental Figure 8 really suggest that changes in RNA structure are independent of RBP binding activity. Since one of the main conclusions derived from the PIP-seq results (both RNA structure and RBP binding) is that there is an overall anti-correlation between RNA structure and PPSs as shown in metaplots Supplemental Figures 9A-C, the authors should really bring home the point that their results suggest that those changes in RNA structure are not driven by RBPs' binding activity. This should be mentioned in the abstract and conclusion. The authors do mention it very clearly in the Results and Discussion section at the end of page 16.

10. In regard to the anti-correlation between RNA structure and PPS, since there is a discrepancy between the interpretation drawn from the metaplot analysis (where anti-correlation is observed, Supplemental Figures 9A-C) and other analysis such as the boxplot analysis in Supplemental Figure 8), I was wondering if outliers could drive this anti-correlation in the metaplots. The authors plotted the average in their metaplot; therefore, each point can be dramatically affected by outlier values. One way to avoid this is to normalize values within each transcript so that each transcript accounts for the same total amount in the metaplot.

11. Conclusion on page 16 end of second paragraph, the authors wrote "Overall, our findings revealed that transcripts encoding proteins associated with hematopoietic processes and phenotypes are those marked by large changes in RNA secondary structure throughout mammalian red blood cell development." The authors overstate their results by writing "are those marked by large changes". The authors don't show that all genes changing in RNA structure encode for proteins associated with the hematopoietic process and phenotypes, nor the opposite. The authors should tone down this conclusion and rather refer to an enrichment. As mentioned in major point #1, proper background should be used in this analysis and the conclusion should be toned down.

12. Maybe I just didn't find this information on the journal's website, but it would be interesting if the authors can provide tables with the results of the motif searches (from Figure 6A, 6B-E and associated Supplemental Figures).

13. The rationale of choosing the Dido1 and Appl2 mRNAs as potential targets of DKC1 is poorly explained, which makes it look suspicious. The authors wrote "Specifically, we sought to validate some of the transcripts predicted to interact with this RBP based on sequence similarity to the motif with which we found it interacts (motif 2). We selected 2 transcripts (Dido1 and Appl2) that contained at least one matching motif and were annotated to exhibit abnormal erythrocyte phenotypes when mutated in mouse models [67]." What do the authors mean by "based on sequence similarity to the motif with which we found it interacts". What level of similarity? Do the

Dido1 and Appl2 mRNAs contain a motif 2 bound by DKC1? Do they have a PPS with the motif 2? The authors should explain better their rationale to make sure it doesn't look like they picked two transcripts with known phenotypes and something that looks like a DKC1 binding site in them.

14. On page 25 second paragraph starting at "We performed RNA immunoprecipitation (RIP) followed by quantitative reverse transcriptase PCR..." and for the remaining of the paragraph, this paragraph should be revisited since it is particularly hard to follow the way it is written.

Minor points:

1. In their analysis of PABPC1 CLIP sites in PPSs (at the end of page 8), why do the authors don't report the results for day 4. Only results from day 0 and day 2 are reported. As mentioned in major point #2 the authors should also use a proper background for this analysis.

2. This point is relevant for the authors, but also the journal, for future submissions. It would be very helpful if the figures number could be annotated either directly in the submission process or manually by the authors. Once printed and all over the place, figures with a bunch of motifs are very difficult to identify and put back in order.

3. Typo on page 19 "this this same are in day4,".

4. Potential typo on page 22 "motif 3 had 18 proteins bound by proteins in lysates from day 4 but not day 0,". I think it should be day 2 rather than day 0.

5. Typo on page 25 first paragraph "is noted to be important an important player".

6. Figure 2D sorting transcripts by % bound let's say at Day 0 would help interpreting the results.

7. Figure 2F, define VST. Maybe a better alternative would be scatter plots with x and y axes being VST and % covered.

8. Figure 4 add label to mentioned that one is Day 2 vs Day 0 and the other Day 4 vs Day 0.

9. Figure 5A add labels for each box plot (cluster 1, cluster 2, etc).

10. Figure 5A cluster 4 and 5 have different box style. It would be better to keep the same style throughout the manuscript and figure panel.

11. Figure 5B and 5C, figure legend should include all the clusters or at the very least keep the same color scheme between 5B and 5C.

Reviewer #3 (Comments to the Authors (Required)):

The authors report a ton of new work in this revised manuscript and attempted to address the main critiques, but the new work raises new questions. Some of the added work illustrates more clearly the potential value of this approach to identifying new candidate RBP regulatory proteins important in erythropoiesis.

Old issues for reviewer 3:

1. The question was raised as to whether changes in PPS and secondary structure profiles might be caused by a change in the transcript profile due to huge increase in globin transcripts as the cells differentiate. Here I will comment on the author's response (1.a. and 1.b.), and in 1.c. will pose the question in a more general form that is important to interpretation of the whole study.

1.a. The authors responded to the question by re-analyzing the data after filtering out "Hbb", and they state that the main conclusions remain unchanged. This is great as far as it goes, but it doesn't fully address the issue because Hbb is only one of the globin genes with increased expression in maturing cells.

1.b. Suppl Figure 7 shows data for secondary structure and PPSs in mouse Hbb mRNA, for the regions +/- 400nt flanking the start and stop codons. Since the Hbb-1 mRNA transcript is ~444nt from start codon to stop codon, there should be substantial overlap in the regions profiled starting from either end, but there doesn't appear to be overlap in the structure profiles in this figure. What exactly is depicted in the figure? Even if it is actually represents the pre-mRNA, there should be a small amount of overlap in the profiles.

1.c. Actually, the important question concerning population changes in secondary structure should be posed differently, because the issue is bigger than globin alone, given the huge changes in gene expression that occur during erythropoiesis. Can the data distinguish whether there is a broad change in secondary structure and RBP interactions within individual transcripts, vs a broad change in the abundance of transcripts with different intrinsic properties (but relatively little change in individual transcripts)?

It is quite possible that the authors' conclusions are correct as stated, and maybe the data is actually here and I missed it. Figure 4 certainly indicates that profile changes do occur within individual transcripts, so it appears the data is available to address my concern without too much trouble.

2.a. Regarding my query about whether ribosome binding sites are skewing interpretation of PPS data: the authors have reanalyzed the data by filtering out PPSs between 20-40nt in length, since ribosome footprints are said to be ~30nt, and they report that their main conclusions remain unchanged. This would be very helpful except for the fact that the Silverman et al. (2014) paper reported that the median PPS sizes for formaldehyde-cross-linked ss- and dsRNase treatments were 35-40 nucleotides. This raises the question as to what really has been filtered out. Is the median size of PPS in this new study also 35-40nt? If so, then a substantial portion of the desired PPSs are being lost as well, yet only 17% of footprints are reported to be lost. Please clarify.

2.b. I also asked about profiling PPS and secondary structure in introns. The authors reply that they are confident in their calling of intron PPSs, because in essence the data is normalized to RNA abundance values by calling PPSs in footprinting samples vs. structure samples. I agree that this validates PPS calls, but doesn't address the fact that many calls may be missed due to low abundance of intron sequences relative to CDS and UTR sequences in mature mRNA. Therefore I suspect the enrichment of PPSs in CDS and UTR in Figure 3, relative to the under-representation of intron sites, may be may be an artifact. There is a disclaimer in the text to acknowledge under-representation of introns, but unless I'm missing something it seems counterproductive to show the figure while pointing out it's deficiency.

New issues:

3. New Figure 2 reports that PPSs appearing only in differentiated cells (i.e., not day 0) are enriched

in transcripts associated with phenotype terms related to abnormal hematopoiesis, and GO terms related to erythroid development. Enrichment studies across erythroid differentiation can be confounded by the tremendous changes in gene expression. This is a common problem and raises the issue as to what background set of genes should be used. I've been advised by reviewers in the past that one should use genes expressed in erythroblasts as the background set.

4. The last new sections devoted to identifying potential post-transcriptional regulators of erythropoiesis are interesting, if somewhat preliminary. It is quite useful as a guide to demonstrate how motifs enriched in PPSs can be used as affinity probes to pull down RBPs that bind, and then validate predictions by using antibody to the RBP (e.g., DKC1) to confirm via RIP-qPCR that it does bind the predicted target transcripts.

5. New data in Fig. 2 provides information on percentage of RNAs covered by PPSs. It looks much lower than what I would have expected, since we don't think RNAs are "naked" in the cell. Does this mean that, useful as these methods are, they don't capture all RBP binding?

6. The absence of figure labels makes it quite inconvenient to associate specific figures with the text and with the figure legends. With this confusion in mind, Figure legend 8 doesn't seem to be associated with any text or any Figure. Moreover, the text refers to a Suppl Figure 20E that I didn't see in the provided documents. I think perhaps the relevant figure is Suppl Figure 19B ... but again it is complicated by the lack of Figure numbers.

Minor issues

1. Suppl Fig. 1B: why are some of the gene names listed multiple times? It appears that the transcript labeling needs substantial editing.

2. Several places in the text refer to the "Methods" section, but that section is actually labeled as "Experimental Procedures". (It makes a difference if one uses 'methods' as the search term to find that section quickly on the computer).

3. Suppl Fig. 6: In part A: TFRC, a transferrin receptor, is listed in the figure as an RBP. Part B: why give percentages to hundredths if everything is in whole numbers? Actually, it doesn't seem plausible that all results are exact whole numbers unless the number of PPSs being assessed is exactly 100.

4. p.14: Please explain what is meant by the statement that the "increase in RNA secondary structure is likely to result in RNAs acquiring a more energetically favorable state (more paired) during these later stages of developmental." Why is this more favorable? (And please note that 'developmental' is the wrong form of the word here.)

5. p.15. As noted earlier, globin transcripts are the products of multiple genes, not only Hbb.

6. Typos noted in the following sentences (just a few that I happened to notice - need to do spellcheck):

p.3 polyadenylation is misspelled

p.26: "When we examined our mRNA-seq data, we found that Appl2 shows a continual and significant increase in RNA abundance through bout MEL development while Dido1 ..."

p.23: "This enrichment for proteins involved in alternatively splicing in our mass spectrometry data highlights the potential of alternative splicing as a key post-transcriptional regulation mechanism in

mammalian erythropoiesis.

7. For readers unfamiliar with the VST term (including this reviewer), please explain what the Y axis values represent in Fig. 2 and some of the Supplemental Figures. Also, is it a linear scale?

Reviewer #2 (Comments to the Authors (Required)):

In their revised version, Shan and colleagues have improved the writing of the paper, streamlined their rationale at few places and added few pieces of information. These improvements help the flow of the manuscript and lead to a better understanding of the results for the first half of the manuscript. However, the second half of the manuscript still needs significant redaction and stream lining of ideas to help the reader understand their rationales and results. This is also needed to be able to assert the strength of their conclusions and quality of the datasets. Moreover, few conclusions are overstated and not supported by the results (see below). These are major points that should be addressed:

1. The authors heavily rely on GO term and Mammalian phenotype prediction analyses to validate the results of their different datasets (both RNA structure and RBP binding). However, their background controls are inadequate. The signal from these techniques is directly linked to the expression level of individual transcripts. You can't detect RNA structure or RBP binding on non-expressed mRNA and, vice versa, you will have a better signal for highly expressed mRNAs due to better coverage. Therefore, in all their GO term and mammalian phenotype analyses, they should use as a background a list of genes that are expressed in their cell lines. For example, they can use their mRNA-seq datasets, use a cutoff to determine what genes are expressed at each time point and use these as backgrounds. Here, to my understanding of the default background in DAVID and MouseMine, the authors use all the genes found in the genome as background. By doing so, you will for sure get GO terms or phenotypes linked to erythropoiesis and related terms since it is the equivalent of doing the same type of analysis with mRNA-seq data. The authors should redo all their GO term and Mammalian phenotype analysis using expressed genes as a background.

We have redone the GO analysis as requested, using a background that is comprised of transcripts with >1 TPM in at least one timepoint. However, Supplemental Figure 1C and Supplemental Figure 16A have been kept as is and use the default background. In the former, we are trying to establish that the MEL cell system is an appropriate model for erythropoiesis and do so by demonstrating that, out of all the possible mRNAs that could be expressed in a mouse cell, the MEL cells are enriched for erythropoiesis-relevant terms. In the latter, we are confirming that genes we have chosen to use for motif enrichment are valid, again by demonstrating that out of all the possible mRNAs we could have chosen in any mouse cell, we have chosen to look at those that are relevant to erythropoiesis.

2. Still related to backgrounds, the modification of the authors' background choice for the PPSs analysis is better than previously (limited to annotated regions of pre-mRNAs versus anywhere in the chromosome), but it is not accurate enough. The nucleotide composition changes significantly across different pre-mRNA regions. The nucleotide composition of introns is different than the one from exons and UTRs different from CDS. When analyzing their PPSs versus a random background, the authors should take background regions in the same pre-mRNA regions than the PPS. For example, the background of a PPS in the 3'UTR should be a sequence of equal length in a 3'UTR. If the number of PPSs is low, they could choose X number of sequences (e.g. 10) per PPS.

We have analyzed the set of PPSs that were used as “background” for Supplemental Figures 5B and 5D and matched the distribution of PPSs in the 3’ UTR, CDS, 5’ UTR, intron, and/or intergenic regions as closely as possible. The breakdown of the background events is depicted in the plot below for the information of the Reviewers (but not included in the manuscript figures).

3. To address the impact of ribosomes in their PPS analyses, the authors have excluded fragments in the 20-40 nt range. It is true that the ribosome footprint of a translating ribosome is ~28 nt in condition where there is a complete digestion of the RNA. In their PIP-seq experiments, it is unclear to me if the RNA digestion is high enough to fully digest the RNA and lead to ribosome-mediated ~28-nt fragments. It seems more likely to me that they are working in conditions where there is only partial digestion of the RNA and, therefore, the fragment impacted by the presence of ribosomes could be longer than ~28-nt. In my opinion, the authors can't use the removal of 20-40 nt fragments to remove the potential impact of ribosomes, and because I expect digestion to be incomplete (see point #4) they should rather assume that what they observe could come from the signal of ribosomes.

In an abundance of caution, and in response to previous Reviewer comments, we excluded the PPSs between 20 – 40 nts. Our re-analysis of the data with the excluded PPSs showed that none of the major conclusions changed and the manuscript was submitted with the re-

analysis. However, as requested by the Reviewer, the full list of PPSs will be released in the associated GEO record with the publication.

4. My rationale of why the signal originates from partially digested RNA is that the miRNeasy RNA isolation kit keeps fragment of >18-nt. If we think about the RNA structure part of the PIP-seq protocol, I think that fully double-stranded RNA or single-stranded RNA of >18-nt long are very rare in the transcriptome. Therefore, most of the fragments (especially as they increase in length) comes from partially digested RNAs. A similar rational could be made for RBP. Most RBPs are likely to have a footprint smaller than 18-nt. The

ribosome is a particularly large machinery composed of several RNAs and proteins and has a footprint of ~28-nt. On top of that, if the authors remove fragments with length between 20-40 nt, that means that most fragments are likely above 40-nt or concentrated between ~18 and 20-nt. To help the reader assess the authors' results and analyses. The authors should provide a fragment length distribution for each of their samples (RNA structure and PPSs).

We have provided a distribution of the length of the identified PPSs as Figure 1A, showing that a large portion of the PPSs we considered did not fall between 20 – 40 nt. We call PPSs with the full read set and then remove PPSs that fall between 20 – 40 nts. This allows us to detect regions of interaction that are >40 nts. To address the reviewer's comment, we have also plotted the distribution of the read length for all of our PIP-seq libraries below.

Replicate 1

Replicate 2

5. Conclusion on page 11 at the end of the second paragraph, the authors wrote "We found no strong correlation between PPS coverage and RNA abundance (Figure 2F), suggesting that in general the total level of RNA-RBP interactions detected on mRNAs is independent of RNA abundance in mammalian erythropoiesis." This conclusion is misleading. I do agree with the authors that it doesn't seem like there is a strong bias in term of percentage of coverage and RNA levels for transcripts with at least 1 PPSs. However, I am convinced that if you compare the RNA level (from mRNA-seq experiment) of transcripts with 1 or more PPSs versus those without any PPS, the ones with 1 or more PPSs are found among the most highly expressed transcripts. As mentioned in major point #1, in these types of experiment you have a better signal from highly expressed genes. Especially with a sequencing depth of 36-58 million reads per library. The authors should rephrase their conclusion in this regard.

We have separated the transcripts into groups depending on how much of the transcript is covered by a PPS and plotted the distribution of $\log_{10}(\text{TPM})$ for each group (Figure 2F). The results show that while there seems to be an upward trend in the median $\log_{10}(\text{TPM})$ values as you increase how much the transcript is covered, the range of TPM values in each group makes it hard to draw the conclusion that there is a strong correlation between RNA abundance and PPS coverage.

6. Conclusion on page 13 at the end of the first paragraph, the authors wrote "Taken together, our results demonstrate that the transcripts encoding known RBPs are differentially regulated in erythropoiesis and that we can use our PPSs to look for potential regions of RBP-RNA interactions for these known RBPs." To use "the transcripts encoding known RBPs" suggests that all mRNAs encoding known RBPs are differentially regulated in erythropoiesis. This is not what the authors show. They should replace the "the" by "certain" or "a subset" or equivalent.

This statement has been replaced with "our results demonstrate that the transcripts encoding several of the known RBPs are differentially regulated in erythropoiesis" as recommended.

7. Conclusion on page 14 at the end of the first paragraph, the authors wrote "In total, these findings reveal large-scale changes in RNA secondary structure during a mammalian cell developmental process that likely underlie important post-transcriptional regulatory processes important to mammalian erythropoiesis." This is an overstatement of the results. The authors don't provide evidence that RNA structure changes are likely underlying important post-transcriptional regulatory processes, nor that they are important for erythropoiesis. This conclusion should be tone down to reflect the results.

We have adjusted the conclusion as suggested.

8. Page 14 second paragraph, the authors wrote "In the 400 nt window after the start codon". It is unclear and a bit hard to follow that the authors mention 400 nt window in the main text, but +/- 500 nt window flanking the start and stop codon in the Figure 3 legend and finally +/- 100 nt window for their p-value. This should be harmonized.

The figure and the appropriate legend have been remade as suggested.

9. Results from Figure 5A and Supplemental Figure 8 really suggest that changes in RNA structure are independent of RBP binding activity. Since one of the main conclusion derived

from the PIP-seq results (both RNA structure and RBP binding) is that there is an overall anti-correlation between RNA structure and PPSs as shown in metaplots Supplemental Figures 9A-C, the authors should really bring home the point that their results suggest that those changes in RNA structure are not driven by RBPs' binding activity. This should be mentioned in the abstract and conclusion. The authors do mention it very clearly in the Results and Discussion section at the end of page 16.

We have added statements about these findings to the Abstract and Conclusions section as suggested by the Reviewer.

10. In regard to the anti-correlation between RNA structure and PPS, since there is a discrepancy between the interpretation drawn from the metaplot analysis (where anti-correlation is observed, Supplemental Figures 9A-C) and other analysis such as the boxplot analysis in Supplemental Figure 8), I was wondering if outliers could drive this anti-correlation in the metaplots. The authors plotted the average in their metaplot; therefore, each point can be dramatically affected by outlier values. One way to avoid this is to normalize values within each transcript so that each transcript account for the same total amount in the metaplot.

Supplemental Figures 9A-C plot the average PPS density and RNA secondary structure around the start and the stop codon, showing the anti-correlation when we look at the average values from all the transcripts. In Supplemental Figure 8, the boxplots show that the change in RNA secondary structure is not influenced by whether the PPS is found within the 3' or the 5' UTR. We would like to note that in Supplemental Figures 9A-C, the shaded regions indicate the SEM, which should account for outliers, which means specifically if we had extreme outliers, the shaded region should be much larger.

11. Conclusion on page 16 end of second paragraph, the authors wrote "Overall, our findings revealed that transcripts encoding proteins associated with hematopoietic processes and phenotypes are those marked by large changes in RNA secondary structure throughout mammalian red blood cell development." The authors overstate their results by writing "are those marked by large changes". The authors don't show that all genes changing in RNA structure encode for proteins associated with the hematopoietic process and phenotypes, nor the opposite. The authors should tone down this conclusion and rather referred to an enrichment. As mentioned in major point #1, proper background should be used in this analysis and the conclusion should be tone down.

The conclusion has been adjusted as suggested.

12. Maybe I just didn't find this information on the journal's website, but it would be interesting if the authors can provide tables with the results of the motif searches (from Figure 6A, 6B-E and associated Supplemental Figures).

We have provided the HOMER text outputs, with the position weigh matrices for all the motifs discovered in the analysis for Figures 6 and the Supplemental Figures, in an Excel notebook as Supplemental Table 2.

13. The rationale of choosing the Dido1 and Appl2 mRNAs as potential targets of DKC1 is poorly explained, which makes it look suspicious. The authors wrote "Specifically, we sought to validate some of the transcripts predicted to interact with this RBP based on sequence similarity to the motif with which we found it interacts (motif 2). We selected 2 transcripts (Dido1 and

Appl2) that contained at least one matching motif and were annotated to exhibit abnormal erythrocyte phenotypes when mutated in mouse models [67]." What do the authors mean by "based on sequence similarity to the motif with which we found it interacts". What level of similarity? Do the *Dido1* and *Appl2* mRNAs contain a motif 2 bound by DKC1? Do they have a PPS with the motif 2? The authors should explain better their rationale to make sure it doesn't look like they picked two transcripts with known phenotypes and something that looks like a DKC1 binding site in them.

This was specifically the point of the experiment that we did and presented in the manuscript. Specifically, we picked two transcripts that were involved in erythropoiesis (since that was the process of interest) and had a PPS which contained the motif that could be a DKC1 binding site.

DKC1 hasn't been shown to bind to either of these transcripts before our analysis, so we were not validating known interactions. Rather, our data showed that *Dido1* and *Appl2* each had a PPS that contained a motif with which we had pulled down DKC1 in the mass spectrometry. We focused on these two transcripts, as opposed to the other ones because their mouse models showed abnormal erythrocyte phenotypes, suggesting they might be involved in erythropoiesis. Once again, the entire point of this experiment was to take a transcript involved in erythropoiesis with a motif that we suspect is the binding site of an RBP and then validate that the hypothesized RBP does indeed bind to the transcript.

14. On page 25 second paragraph starting at "We performed RNA immunoprecipitation (RIP) followed by quantitative reverse transcriptase PCR..." and for the remaining of the paragraph, this paragraph should be revisited since it is particularly hard to follow the way it is written.

We have rewritten the section to better explain this and hopefully clear up these confusions. We hope the re-write explains things better.

Minor points:

1. In their analysis of PABPC1 CLIP sites in PPSs (at the end of page 8), why do the authors don't report the results for day 4. Only results from day 0 and day 2 are reported. As mentioned in major point #2 the authors should also use a proper background for this analysis.

These was no data reported for day 4 because the PABPC1 CLIP was done in a different study, which did not include day 4 as a relevant timepoint. Therefore, we cannot compare our day 4 PIP-seq data with CLIP-seq data.

2. This point is relevant for the authors, but also the journal, for future submissions. It would be very helpful if the figures number could be annotated either directly in the submission process or manually by the authors. Once printed and all over the place, figures with a bunch of motifs are very difficult to identify and put back in order.

3. Typo on page 19 "this this same are in day4,".

This was fixed as requested.

4. Potential typo on page 22 "motif 3 had 18 proteins bound by proteins in lysates from day 4 but not day 0,". I think it should be day 2 rather than day 0.

Motif 3 was only evaluated in day 4 and day 0 cells (no day 2 cells were used) so day 0 is correct.

5. Typo on page 25 first paragraph "is noted to be important an important player".

This was fixed as requested.

6. Figure 2D sorting transcripts by % bound let's say at Day 0 would help interpreting the results.

The figure is sorted by the default k-means clustering that is part of the heatmap analysis; the lines depicting the clustering were removed for legibility and because they were not part of future analyses.

7. Figure 2F, define VST. Maybe a better alternative would be scatter plots with x and y axes being VST and % covered.

VST in the plots have been changed to the better known TPM metric.

8. Figure 4 add label to mentioned that one is Day 2 vs Day 0 and the other Day 4 vs Day 0.

This was done as requested.

9. Figure 5A add labels for each box plot (cluster 1, cluster 2, etc).

This was done as requested.

10. Figure 5A cluster 4 and 5 have different box style. It would be better to keep the same style throughout the manuscript and figure panel.

They are all the same type of plots (box plot with notch where the median is), it's just harder to see the notch in the box for the other clusters. The notch indicates the 95% CI around the median and in the case of clusters 1 and 2, that interval is small, which is why it's hard to see.

11. Figure 5B and 5C, figure legend should include all the clusters or at the very least keep the same color scheme between 5B and 5C.

Some of the clusters had no significant terms, which is why they were not included in figure 5B.

Reviewer #3 (Comments to the Authors (Required)):

The authors report a ton of new work in this revised manuscript and attempted to address the main critiques, but the new work raises new questions. Some of the added work illustrates more clearly the potential value of this approach to identifying new candidate RBP regulatory proteins important in erythropoiesis.

Old issues for reviewer 3:

1. The question was raised as to whether changes in PPS and secondary structure profiles might be caused by a change in the transcript profile due to huge increase in globin transcripts as the cells differentiate. Here I will comment on the author's response (1.a. and 1.b.), and in 1.c. will pose the question in a more general form that is important to interpretation of the whole study.

1.a. The authors responded to the question by re-analyzing the data after filtering out "Hbb", and they state that the main conclusions remain unchanged. This is great as far as it goes, but it doesn't fully address the issue because Hbb is only one of the globin genes with increased expression in maturing cells.

The full list of transcripts that were filtered out in the revised document include: Hbb-y, Hbb-bh1, Hbb-bh2, and Hbb-bt. In the revised analysis, we have further excluded Hba-a2, Hba-a1 and Hba-x and the conclusions remain unchanged.

1.b. Suppl Figure 7 shows data for secondary structure and PPSs in mouse Hbb mRNA, for the regions +/- 400nt flanking the start and stop codons. Since the Hbb-1 mRNA transcript is ~444nt from start codon to stop codon, there should be substantial overlap in the regions profiled starting from either end, but there doesn't appear to be overlap in the structure profiles in this figure. What exactly is depicted in the figure? Even if it is actually represents the pre-mRNA, there should be a small amount of overlap in the profiles.

In order to also address point 1a, we have redone the analysis to show the metaplots of PPS density and RNA secondary structure that now include Hbb-y, Hbb-bh1, Hbb-bh2, Hbb-bt, Hba-a2, Hba-a1 and Hba-x.

1.c. Actually, the important question concerning population changes in secondary structure should be posed differently, because the issue is bigger than globin alone, given the huge changes in gene expression that occur during erythropoiesis. Can the data distinguish whether there is a broad change in secondary structure and RBP interactions within individual transcripts, vs a broad change in the abundance of transcripts with different intrinsic properties (but relatively little change in individual transcripts)?

It is quite possible that the authors' conclusions are correct as stated, and maybe the data is actually here and I missed it. Figure 4 certainly indicates that profile changes do occur within individual transcripts, so it appears the data is available to address my concern without too much trouble.

The full dataset of structure scores and PPSs will be released in the relevant GEO record, so it is possible for interested parties to go in and query their favorite gene on a transcript-by-transcript basis. Since enriched RBP binding sites are determined by comparing the footprint vs. control library created from the same sample, the changes in abundance between time points should have little effect on whether a region is identified as a PPS or not (i.e. the only difference that matters is whether we see an over-abundance in reads in the footprinting library vs. the control library). The same principle also applies to the RNA structure score, which is calculated as a ratio of ssRNA vs. dsRNA reads taken from the same sample. Therefore, changes in RNA abundance between time points does not impact whether the algorithm assigns a higher/lower RNA structure score to a region between different time points.

2.a. Regarding my query about whether ribosome binding sites are skewing interpretation of PPS data: the authors have reanalyzed the data by filtering out PPSs between 20-40nt in length, since ribosome footprints are said to be ~30nt, and they report that their main conclusions remain unchanged. This would be very helpful except for the fact that the Silverman et al. (2014) paper reported that the median PPS sizes for formaldehyde-cross-linked ss- and dsRNase treatments were 35-40 nucleotides. This raises the question as to what really has been filtered out. Is the median size of PPS in this new study also 35-40nt? If so, then a substantial portion of the desired PPSs are being lost as well, yet only 17% of footprints are reported to be lost. Please clarify. **We have plotted the distribution of the PPS sizes for the full dataset (prior to any filtering) in Figure 1A and indicated the PPSs that were moved by blue dotted lines. Our average PPS size was greater 35 – 40 nt, which means that with our filtering step, we have included a majority of the PPSs.**

2.b. I also asked about profiling PPS and secondary structure in introns. The authors reply that they are confident in their calling of intron PPSs, because in essence the data is normalized to RNA abundance values by calling PPSs in footprinting samples vs. structure samples. I agree that this validates PPS calls, but doesn't address the fact that many calls may be missed due to low abundance of intron sequences relative to CDS and UTR sequences in mature mRNA. Therefore I suspect the enrichment of PPSs in CDS and UTR in Figure 3, relative to the under-representation of intron sites, may be may be an artifact. There is a disclaimer in the text to acknowledge under-representation of introns, but unless I'm missing something it seems counterproductive to show the figure while pointing out it's deficiency.

Our original intent with the figure was to show that, despite having a lot of PPSs in the intronic regions, we actually see an under-representation of intronic PPSs once we normalized the data to how much of the genome is made up of introns. In other words, the large number of intronic PPSs we observe is likely a result of the number of bases that are intronic. But in terms of likelihood, it is actually far likelier for an RBP to bind to an exon or in the UTRs than it is for an RBP to bind in an intron space.

However, we agree with the reviewer's comment that figure is counterproductive and, in response, have removed Figure 1C. In its stead, we have added a density plot showing the distribution of PPS sizes as to validate our claim that we have retained a large portion of PPSs even with filtering out those that are 20 – 40 nt in size.

New issues:

3. New Figure 2 reports that PPSs appearing only in differentiated cells (i.e., not day 0) are enriched in transcripts associated with phenotype terms related to abnormal hematopoiesis, and GO terms related to erythroid development. Enrichment studies across erythroid differentiation can be confounded by the tremendous changes in gene expression. This is a common problem and raises the issue as to what background set of genes should be used. I've been advised by reviewers in the past that one should use genes expressed in erythroblasts as the background set. **The GO analysis has been redone with the appropriate background (only transcripts with >1 FPM in our datasets), except for Supplemental Figure 1C and Supplemental Figure 16A as the primary goal for these two figures were to demonstrate the relevance of**

the MEL model for erythropoiesis (Supplemental Figure 1C) and the relevance of the transcripts selected for motif enrichment analysis (Supplemental Figure 16A) (see also response to comment from Reviewer 2 above). In both cases, the intent is to demonstrate that out of all the possible mRNAs from a mouse, the transcripts selected were enriched for erythropoiesis relevant terms.

4. The last new sections devoted to identifying potential post-transcriptional regulators of erythropoiesis are interesting, if somewhat preliminary. It is quite useful as a guide to demonstrate how motifs enriched in PPSs can be used as affinity probes to pull down RBPs that bind, and then validate predictions by using antibody to the RBP (e.g., DKC1) to confirm via RIP-qPCR that it does bind the predicted target transcripts.

We thank the Reviewer for these very positive comments about our new findings.

5. New data in Fig. 2 provides information on percentage of RNAs covered by PPSs. It looks much lower than what I would have expected, since we don't think RNAs are "naked" in the cell. Does this mean that, useful as these methods are, they don't capture all RBP binding?

PIP-seq is not intended to capture all RBP binding; it is subject to influences such as tissue, developmental time point, as well as technical biases as we use formaldehyde crossing to fix the RBP-RNA interactions. Furthermore, as PIP-seq analysis does require for there to be a significant enrichment for reads in footprinting vs. control sample, weak interactions could be discarded as a false negative. PIP-seq's strengths lies in its ability to simultaneously probe RNA-secondary structure and RBP-RNA interactions, as well as to capture a wide set of interactions that could then be used for motif discovery.

6. The absence of figure labels makes it quite inconvenient to associate specific figures with the text and with the figure legends. With this confusion in mind, Figure legend 8 doesn't seem to be associated with any text or any Figure. Moreover, the text refers to a Suppl Figure 20E that I didn't see in the provided documents. I think perhaps the relevant figure is Suppl Figure 19B ... but again it is complicated by the lack of Figure numbers.

We apologize for the inconvenience that our mis-labeling has caused. These issues are now taken care of in this revised version. Figures are now labeled by their number in the file name and the legends are within the body of the text.

Furthermore, Supplemental Figure 20E has been changed to Supplemental Figure 19B. Figure 8 legend has been removed.

Minor issues

1. Suppl Fig. 1B: why are some of the gene names listed multiple times? It appears that the transcript labeling needs substantial editing.

They were labeled multiple times because multiple transcripts were annotated to correspond to the same gene and the RNA abundance patterns of those transcripts were different enough that we didn't want to average them and lose the signal. The updated figure now displays their average TPM.

2. Several places in the text refer to the "Methods" section, but that section is actually labeled as

"Experimental Procedures". (It makes a difference if one uses 'methods' as the search term to find that section quickly on the computer).

We have changed Methods to Experimental Procedures as requested.

3. Suppl Fig. 6: In part A: TFRC, a transferrin receptor, is listed in the figure as an RBP. Part B: why give percentages to hundredths if everything is in whole numbers? Actually, it doesn't seem plausible that all results are exact whole numbers unless the number of PPSs being assessed is exactly 100.

TFRC according to the annotation found in UCSC (https://genome.ucsc.edu/cgi-bin/hgGene?hgg_gene=ENSMUST00000023486.14&hgg_prot=uc007vza.3&hgg_chrom=chr16&hgg_start=32608919&hgg_end=32632794&hgg_type=knownGene&db=mm10&hgsid=1004909463_qS3HUoSh3qWsJnVY7KPR8HWHcFYh#go) shows “double-stranded RNA binding” as one of its molecular functions. We don't claim that these are RBPs, just that these have been annotated to have RNA binding capabilities (i.e. “transcripts with RNA binding and erythropoiesis GO annotations”).

The legend should have clarified that we rounded to the nearest whole number for ease of displaying the results. However, we have since updated the heatmap showing the correct decimal points as requested.

4. p.14: Please explain what is meant by the statement that the "increase in RNA secondary structure is likely to result in RNAs acquiring a more energetically favorable state (more paired) during these later stages of developmental." Why is this more favorable? (And please note that 'developmental' is the wrong form of the word here.)

In general, base paired nucleotides (double-stranded RNA regions) are considered to be more stable, and thus this state is considered to be more energetically favored.

We also changed developmental to development as requested.

5. p.15. As noted earlier, globin transcripts are the products of multiple genes, not only Hbb. **For the relevant figures, we have redone the analysis to include more globin transcripts (as specified) as suggested by this Reviewer.**

6. Typos noted in the following sentences (just a few that I happened to notice - need to do spellcheck):

p.3 polyadenylation is misspelled

We have fixed this typo as directed.

p26: "When we examined our mRNA-seq data, we found that Appl2 shows a continual and significant increase in RNA abundance through out MEL development while Dido1 ..."

We have fixed this typo as directed.

p.23: "This enrichment for proteins involved in alternatively splicing in our mass spectrometry data highlights the potential of alternative splicing as a key post-transcriptional regulation mechanism in mammalian erythropoiesis.

We have fixed this typo as directed.

7. For readers unfamiliar with the VST term (including this reviewer), please explain what the Y axis values represent in Fig. 2 and some of the Supplemental Figures. Also, is it a linear scale?

We have changed all analyses that include VST (which is a normalized mRNA read count metric) to the more prevalent and more easily understood TPM metric.

May 21, 2021

RE: Life Science Alliance Manuscript #LSA-2020-00659-TRR

Dr. Brian D Gregory
University of Pennsylvania
Biology
433 S. University Ave.
Philadelphia, PA 19104

Dear Dr. Gregory,

Thank you for submitting your revised manuscript entitled "Dynamic Changes in RNA-Protein Interactions and RNA Secondary Structure in Mammalian Erythropoiesis". We would be happy to publish your paper in Life Science Alliance pending minor revisions in response to the reviewer's comments (appended at the end of this email) and final revisions necessary to meet our formatting guidelines.

As you will note below, the reviewer has expressed a concern about using genes expressed at >1 TPM (transcripts per million) at at least one timepoint as control (pt 1) but they have also mentioned that they are not an expert in determining whether this is an appropriate background to use. We shared the reviewer's report with our academic board expert, who has assured us that your method, adjusting background set for GO analysis to include only transcripts expressed at 1 TPM in at least one stage, is appropriate.

- We do encourage you to address the concern about over-simplification mentioned by the reviewer in pt 1 with text changes.
- Reviewers' pts 2-4 and minor concerns are mainly requests for clarifications and text changes, which should be addressed in the revision as well

In the interest of speeding up the time in publication, we also suggest you attend to the following formatting requests in the revised manuscript:

- please consult our manuscript preparation guidelines <https://www.life-science-alliance.org/manuscript-prep> and make sure your manuscript sections are in the correct order
- please separate the Results and Discussion section into two - 1. Results 2. Discussion, as per our formatting requirements
- please add Author Contributions for all Authors to your main manuscript text
- please upload your main and supplementary figures as single files
- please upload your main manuscript text as an editable doc file
- please check your figure callouts in your main manuscript text: please add callouts for Figures 5C, 7D, S7A, B; S8A, B to your main manuscript text
- Graphs in Figure 3 match with graphs in Figure S6- if the same graphs were used in both images, please clarify that in the figure legends

A. FINAL FILES:

B. MANUSCRIPT ORGANIZATION AND FORMATTING:

Sincerely,

Shachi Bhatt, Ph.D.
Executive Editor
Life Science Alliance
<http://www.lsajournal.org>
Tweet @SciBhatt @LSAJournal

Reviewer #3 (Comments to the Authors (Required)):

The authors have improved the manuscript in response to previous critiques. Overall I think the methods could have great potential to discover new patterns of structural change and RBP binding in transcripts during terminal differentiation. However, I am still not entirely convinced that the statistical associations are fully justified. It is a difficult problem when vast changes in gene expression are occurring in these cells.

1. One important finding, if verified, is whether 'transcripts exhibiting larger increases in RNA secondary structure around the start codon preferentially encode proteins associated with hematopoietic processes and phenotypes' (paraphrased from p. 16). The authors have responded to the previous critiques by the adjusting background set for GO analysis to include only transcripts expressed at 1 TPM in at least one stage. Compared with the last revision, this change resulted in dramatic differences in the $-\log_{10}(\text{FDR})$ scores for various phenotypes associated with PPSs (Figure 2) or with structure changes (Figure 4). Some of the log scores changed by many orders of magnitude, becoming less significant and emphasizing the great importance of how the background set is defined. It really is a tricky problem when gene expression patterns change so much during differentiation! As one example, "abnormal erythropoiesis" appears to have a $-\log_{10}(\text{FDR})$ score of about 7 in the previous version of the paper, but only about 2 in the current manuscript (estimated from the graphs in Figure 4). Many of the log scores for PPS are also much reduced.

Unfortunately, I don't have sufficient expertise to judge whether 1 TPM in at least one stage is the best background. This definition would seemingly include some (many?) transcripts that are not expressed in the other stages. A slightly more restrictive measure, e.g., requiring expression in all stages, might eliminate the statistical significance of many structure-associated or PPS-associated phenotypes. Unfortunately, this could potentially weaken an important conclusion of the manuscript.

Even if the analysis is judged to be OK, it's an over-simplification to say that transcripts with the largest increase in RNA secondary structure are associated with hematopoietic phenotypes - notice that cluster 4 (Figure 5) encompasses a relatively small group of transcripts with the largest change in structure, but no association with phenotypes.

2. It's also not entirely clear how the 1 TPM measure is applied. In Figure 4 and related figures, are the measures of secondary structure change applied only to transcripts that are expressed in all three stages (day 0, 2, 4)? Previous studies have shown that a huge number of genes are strongly down-regulated during mouse terminal erythropoiesis. If a transcript isn't expressed at day 4, is it excluded from analysis or scored as a transcript with no secondary structure (or no PPS)? Sorry if I don't understand, please clarify.

3. Figure 5: The text says "normalized mRNA abundance increased upon terminal differentiation regardless of whether the RNA secondary structure increased or decreased in comparison to the structure in undifferentiated cells." This would be surprising given previous data showing that a large number of genes decrease in expression during terminal differentiation, at least when primary cells are analyzed. In fact, expression values shown in the log (TPM) plots do not seem to show increased expression. (In contrast, the "normalized mRNA abundance" values shown in the previous version of this manuscript did indicate increased abundance in differentiated cells).

4. The revised abstract raises an issue that I had over-looked before regarding correlation between secondary structure and RBP binding. The abstract now says the authors "identify dynamic patterns of RNA secondary structure and RBP binding that are consistently anti-correlated during erythropoiesis and likely independent of one another". First, as a general rule, I don't think anti-correlation implies independence. Moreover, the statement is oversimplified given that the text contains a number of seemingly inconsistent statements regarding correlation of these processes in the UTRs or across the entire transcripts.

For example:

Page 14: "data revealed an increase in RNA secondary structure in the 5'UTR of day 4 cells in comparison to uninduced cells and, in the same region, we observed a decrease in PPS density, suggesting that the increase in RNA secondary structure and decrease in RBP-RNA interaction are related".

page 16: "changes in RNA secondary structure did not appear to correlate with changes in mRNA abundance or in PPS coverage" and: "This lack of correlation among RNA secondary structure conformation, RBP-RNA interaction, and mRNA abundance suggests that these parameters do not have a cause-and-effect relationship and appear to be largely independent of each other when interrogated on a global scale."

P 17: "we observe an anti-correlation of mRNA secondary structure and RBP binding events around the translation start and stop codons"

p. 20: "the overall analysis of RNA secondary structure profiles and RBP binding densities suggests a general anti-correlation between these two features"

5. p 15: "we do observe an anti-correlation between RBP-RNA interaction sites and RNA secondary structure on a global scale when we examine the entire transcriptome instead of focusing on specific sites." This last statement implies that entire transcripts are being analyzed, but the relevant figure, S9, only shows specific regions around the start and stop codons. Please clarify.

Minor issues and typos.

1. I don't know whether this is the fault of the authors or the journal, but in the merged file the figures still lack figure numbers. This makes review a lot more difficult because one needs to manually label the figures!

2. Fig. 3: is it reasonable for all P values to be the same? For both secondary structure scores and PPS calculations?
3. Fig S7: what are the pink and blue shaded regions in the bottom PPS density profiles for hemoglobin transcripts?
4. P 10: "depending on the functionality the RBP" - something is missing.
5. p. 5: "throughout the three times points of differentiation". Need to delete the "s" in the word times.
6. p. 14: "the structured 5' UTRs observed at day 4 could serve impede the binding of RBPs" - need to add the word "to" after impede.
7. P 16: "abnormal definite hematopoiesis" - should be "definitive"

Editor and Reviewer Responses:

In the interest of speeding up the time in publication, we also suggest you attend to the following formatting requests in the revised manuscript:

-please consult our manuscript preparation guidelines <https://www.life-science-alliance.org/manuscript-prep> and make sure your manuscript sections are in the correct order

-please separate the Results and Discussion section into two - 1. Results 2. Discussion, as per our formatting requirements

We have adjusted the format as requested.

-please add Author Contributions for all Authors to your main manuscript text

We have added author contributions as requested

-please upload your main and supplementary figures as single files

-please upload your main manuscript text as an editable doc file

-please check your figure callouts in your main manuscript text: please add callouts for Figures 5C, 7D, S7A, B; S8A, B to your main manuscript text

We have added callouts to Figures 5C and Figure 7D to the manuscript. As requested, we have split the callouts to Supplemental Figures 7 and 8 into separate calls for S7A, S7B, S8A, and S8B.

-Graphs in Figure 3 match with graphs in Figure S6- if the same graphs were used in both images, please clarify that in the figure legends

The figures in Figure 3 and Figure S7 match to the naked eye but are of different sets of transcripts (Figure S7 is on a smaller subset because we removed and plotted the hemoglobin genes separately as requested by the Reviewers, which had no effect on the overall patterns we observed in Figure 3). The figure legends for Figure 3 and Figure S7 clarify the difference between the two.

To upload the final version of your manuscript, please log in to your account: <https://lsa.msubmit.net/cgi-bin/main.plex>

A. FINAL FILES:

Using PIP-seq, a high-throughput sequencing approach, on a model of erythropoiesis, identifies dynamic changes of RBP-RNA interactions and RNA secondary structure during this developmental process (173 characters)

B. MANUSCRIPT ORGANIZATION AND FORMATTING:

****Reviews, decision letters, and point-by-point responses associated with peer-review at Life Science Alliance will be published online, alongside the manuscript. If you do want to opt out of having the reviewer reports and your point-by-point responses displayed,**

please let us know immediately.**

Sincerely,

Shachi Bhatt, Ph.D.
Executive Editor
Life Science Alliance
<http://www.lsjournal.org>
Tweet @SciBhatt @LSAJournal

Reviewer #3 (Comments to the Authors (Required)):

The authors have improved the manuscript in response to previous critiques. Overall I think the methods could have great potential to discover new patterns of structural change and RBP binding in transcripts during terminal differentiation. However, I am still not entirely convinced that the statistical associations are fully justified. It is a difficult problem when vast changes in gene expression are occurring in these cells.

1. One important finding, if verified, is whether 'transcripts exhibiting larger increases in RNA secondary structure around the start codon preferentially encode proteins associated with hematopoietic processes and phenotypes' (paraphrased from p. 16). The authors have responded to the previous critiques by the adjusting background set for GO analysis to include only transcripts expressed at 1 TPM in at least one stage. Compared with the last revision, this change resulted in dramatic differences in the $-\log_{10}(\text{FDR})$ scores for various phenotypes associated with PPSs (Figure 2) or with structure changes (Figure 4). Some of the log scores changed by many orders of magnitude, becoming less significant and emphasizing the great importance of how the background set is defined. It really is a tricky problem when gene expression patterns change so much during differentiation! As one example, "abnormal erythropoiesis" appears to have a $-\log_{10}(\text{FDR})$ score of about 7 in the previous version of the paper, but only about 2 in the current manuscript (estimated from the graphs in Figure 4). Many of the log scores for PPS are also much reduced.

Unfortunately, I don't have sufficient expertise to judge whether 1 TPM in at least one stage is the best background. This definition would seemingly include some (many?) transcripts that are not expressed in the other stages. A slightly more restrictive measure, e.g., requiring expression in all stages, might eliminate the statistical significance of many structure-associated or PPS-associated phenotypes. Unfortunately,

this could potentially weaken an important conclusion of the manuscript.

Even if the analysis is judged to be OK, it's an over-simplification to say that transcripts with the largest increase in RNA secondary structure are associated with hematopoietic phenotypes - notice that cluster 4 (Figure 5) encompasses a relatively small group of transcripts with the largest change in structure, but no association with phenotypes.

Figure 4 contains GO enrichment analysis performed on a larger set of transcripts (i.e. the top 10% increasing and top 10% decreasing transcripts) while Figure 5 is GO enrichment analysis performed on clusters of genes separated through hierarchical clustering. As a result, cluster 4 in Figure 5 only encompasses 69 transcripts – which is a very small subset of what is presented in Figure 4.

Therefore, we have modified the conclusion in the manuscript relevant to Figure 4 to be more precise.

2. It's also not entirely clear how the 1 TPM measure is applied. In Figure 4 and related figures, are the measures of secondary structure change applied only to transcripts that are expressed in all three stages (day 0, 2, 4)? Previous studies have shown that a huge number of genes are strongly down-regulated during mouse terminal erythropoiesis. If a transcript isn't expressed at day 4, is it excluded from analysis or scored as a transcript with no secondary structure (or no PPS)? Sorry if I don't understand, please clarify.

We have clarified this point in the methodology section of the paper as requested.

3. Figure 5: The text says "normalized mRNA abundance increased upon terminal differentiation regardless of whether the RNA secondary structure increased or decreased in comparison to the structure in undifferentiated cells." This would be surprising given previous data showing that a large number of genes decrease in expression during terminal differentiation, at least when primary cells are analyzed. In fact, expression values shown in the log (TPM) plots do not seem to show increased expression. (In contrast, the "normalized mRNA abundance" values shown in the previous version of this manuscript did indicate increased abundance in differentiated cells).

We have revised the statement as requested.

4. The revised abstract raises an issue that I had over-looked before regarding correlation between secondary structure and RBP binding. The abstract now says the authors "identify dynamic patterns of RNA secondary structure and RBP binding that are consistently anti-correlated during erythropoiesis and likely independent of one another". First, as a general rule, I don't think anti-correlation implies independence. Moreover, the statement is oversimplified given that the text contains a number of seemingly inconsistent statements regarding correlation of these processes in the UTRs or across the entire transcripts.

For example:

Page 14: "data revealed an increase in RNA secondary structure in the 5'UTR of day 4 cells in comparison to uninduced cells and, in the same region, we observed a decrease in PPS density, suggesting that the increase in RNA secondary structure and decrease in RBP-RNA interaction are related".

page 16: "changes in RNA secondary structure did not appear to correlate with changes in mRNA abundance or in PPS coverage" and: "This lack of correlation among RNA secondary structure conformation, RBP-RNA interaction, and mRNA abundance suggests that these parameters do not have a cause-and-effect relationship and appear to be largely independent of each other when interrogated on a global scale."

P 17: "we observe an anti-correlation of mRNA secondary structure and RBP binding events around the translation start and stop codons"

p. 20: "the overall analysis of RNA secondary structure profiles and RBP binding densities suggests a general anti-correlation between these two features"

5. p 15: "we do observe an anti-correlation between RBP-RNA interaction sites and RNA secondary structure on a global scale when we examine the entire transcriptome instead of focusing on specific sites." This last statement implies that entire transcripts are being analyzed, but the relevant figure, S9, only shows specific regions around the start and stop codons. Please clarify.

We observe the anti-correlation (Supplemental Figure S9) when we look at the average structure score and average PPS density for the entire transcriptome (i.e. all the transcripts) instead of focusing on subsets of transcripts that fulfill certain criteria and/or specific sites. This suggests that, globally, RNA secondary structure and PPS density are anti-correlated but individual transcripts can be exceptions to that observation. We have clarified this difference in the manuscript.

Minor issues and typos.

1. I don't know whether this is the fault of the authors or the journal, but in the merged file the figures still lack figure numbers. This makes review a lot more difficult because one needs to manually label the figures!

2. Fig. 3: is it reasonable for all P values to be the same? For both secondary structure scores and PPS calculations?

The results were derived from a statistical test and because of the p-value being small, the algorithm is only able to approximate that they are all less than 2.2×10^{-16} , but not a more specific value.

3. Fig S7: what are the pink and blue shaded regions in the bottom PPS density profiles for hemoglobin transcripts?

We have clarified that the shaded regions indicate the standard error of the mean (SEM) in the figure legends.

4. P 10: "depending on the functionality the RBP" - something is missing.

We have edited the sentence as requested.

5. p. 5: "throughout the three times points of differentiation". Need to delete the "s" in the word times.

We have made the edit as requested.

6. p. 14: "the structured 5' UTRs observed at day 4 could serve impede the binding of RBPs" - need to add the word "to" after impede.

We have edited the sentence as requested.

7. P 16: "abnormal definite hematopoiesis" - should be "definitive"

We have made the edit as requested.

July 7, 2021

RE: Life Science Alliance Manuscript #LSA-2020-00659-TRRR

Dr. Brian D Gregory
University of Pennsylvania
Biology
433 S. University Ave.
Philadelphia, PA 19104

Dear Dr. Gregory,

Thank you for submitting your Resource entitled "Dynamic Changes in RNA-Protein Interactions and RNA Secondary Structure in Mammalian Erythropoiesis". It is a pleasure to let you know that your manuscript is now accepted for publication in Life Science Alliance. Congratulations on this interesting work.

LSA now encourages authors to provide a 30-60 second video where the study is briefly explained. We will use these videos on social media to promote the published paper and the presenting author. Corresponding or first-authors are welcome to submit the video. Please submit only one video per manuscript. The video can be emailed to contact@life-science-alliance.org

*****IMPORTANT:** If you will be unreachable at any time, please provide us with the email address of an alternate author. Failure to respond to routine queries may lead to unavoidable delays in publication.*******

DISTRIBUTION OF MATERIALS:

Again, congratulations on a very nice paper. I hope you found the review process to be constructive and are pleased with how the manuscript was handled editorially. We look forward to future exciting submissions from your lab.

Sincerely,
